# PCSK9 stimulates Syk, PKCδ, and NF-κB, leading to atherosclerosis progression independently of LDL receptor

Dasom Shin [1,2,3,7], Soungchan Kim[1,2,3,7], Hwan Lee[1,2,3,7], Hyun-Chae Lee[1,2], Jaewon Lee [1,2], Hyun-woo Park[1,2,4], Mina Fukai[1,2,3], EunByule Choi[1,2,3], Subin Choi[1,2,3], Bon-Jun Koo[1,2,3], Ji-Hoon Yu[1,2,3], Gyurae No[1,2,3], Sungyoon Cho[1,2,4], Chan Woo Kim[5], Dohyun Han[2], Hyun-Duk Jang [1,2,8] ✉ & Hyo-Soo Kim [1,3,4,6,8] ✉

Proprotein convertase subtilisin/kexin type-9 (PCSK9) binds to and degrades low-density lipoprotein (LDL) receptor, leading to increase of LDL cholesterol in blood. Its blockers have emerged as promising therapeutics for cardiovascular diseases. Here we show that PCSK9 itself directly induces inflammation and aggravates atherosclerosis independently of the LDL receptor. PCSK9 exacerbates atherosclerosis in LDL receptor knockout mice. Adenylyl cyclase-associated protein 1 (CAP1) is the main binding partner of PCSK9 and indispensable for the inflammatory action of PCSK9, including induction of cytokines, Toll like receptor 4, and scavenger receptors, enhancing the uptake of oxidized LDL. We find spleen tyrosine kinase (Syk) and protein kinase C delta (PKCδ) to be the key mediators of inflammation after PCSK9-CAP1 binding. In human peripheral blood mononuclear cells, serum PCSK9 levels are positively correlated with Syk, PKCδ, and p65 phosphorylation. The CAP1-fragment crystallizable region (CAP1-Fc) mitigates PCSK9-mediated inflammatory signal transduction more than the PCSK9 blocking antibody evolocumab does.

Proprotein convertase subtilisin/kexin type-9 (PCSK9), highly expressed in adult hepatocytes, increases low-density lipoprotein cholesterol (LDL-C) levels by promoting the degradation of the LDL receptor (LDLR)[1,2]. Excessive LDL-C in the blood infiltrates the sub-endothelial layer and is oxidized when exposed to reactive oxygen species produced by macrophages, endothelial cells (ECs), and vascular smooth muscle cells (SMCs), resulting in foam cell formation, vascular inflammation, and atherosclerosis[3]. Inhibitors of PCSK9 reduce LDL-C levels and improve cardiovascular outcomes[4,5]. In addition, PCSK9 inhibits the LDLR-mediated clearance of pathogenic lipids and exacerbates the innate immune response, ultimately leading to poor outcomes of sepsis[6].

Pcsk9-knockout mice display a decreased inflammatory response to lipopolysaccharide (LPS) and pharmacological inhibition of PCSK9 improves survival and inflammation in murine poly-microbial peritonitis[6]. Furthermore, Pcsk9 loss-of-function genetic variants in septic shock patients are associated with improved survival, whereas gain-of-function mutants show decreased survivability[6].

PCSK9 produced in hepatocytes is expressed in human atherosclerotic plaques, especially on macrophages that reside in the plaques, leading to an increased local concentration of PCSK9[3]. Local inflammation in the vessel wall is the major cause of atherosclerotic plaque formation. However, it is unclear whether PCSK9 can trigger

[1]Center of CBT (Cell and BioTherapy), Seoul National University Hospital, Seoul, Republic of Korea. [2]Biomedical Research Institute, Seoul National University Hospital, Seoul, Republic of Korea. [3]Department of Molecular Medicine and Biopharmaceutical Sciences, Seoul National University, Seoul, Republic of Korea. [4]Program in Stem Cell Biology, Seoul National University College of Medicine, Seoul, Republic of Korea. [5]Department of Preclinical Trial, Laboratory Animal Center, Osong Medical Innovation Foundation (KBIO), Cheongju, Chungbuk, Republic of Korea. [6]Cardiovascular Center & Department of Internal Medicine, Seoul National University Hospital, Seoul, Republic of Korea. [7]These authors contributed equally: Dasom Shin, Soungchan Kim, Hwan Lee. [8]These authors jointly supervised this work: Hyun-Duk Jang, Hyo-Soo Kim. ✉e-mail: 65765@snuh.org; hyosoo@snu.ac.kr

inflammation in the blood vessel as a direct signal modulator of monocytes or ECs independently of LDLR.

Previously, we showed that adenylyl cyclase-associated protein 1 (CAP1), the receptor for human resistin, binds to PCSK9[7]. This interaction is a prerequisite for PCSK9-mediated lysosomal degradation of LDLR, which increases plasma LDL-C levels. Because Resistin binding to CAP1 leads to nuclear factor (NF)-κB activation via the cyclic adenosine 3′,5′ monophosphate (cAMP)/protein kinase A (PKA) pathway[8], we speculated that PCSK9 binding to CAP1 may switch on pro-inflammatory signaling and further aggravate atherosclerosis independently of LDLR.

In this study, our aim was to ascertain whether PCSK9 possesses the capacity to induce inflammation directly, exacerbating atherosclerosis, independently of alterations in lipid profiles. Moreover, we sought to elucidate the mechanism underlying PCSK9 secretion from macrophages and its direct induction of inflammation within atherosclerotic plaques, focusing on *Ldlr*−/− mice and *Cap1* hetero-knockout mice as experimental models. Additionally, we also identified CAP1 binding partners and elucidated the downstream signaling pathways involved. Our findings strongly suggest that CAP1 serves as the pivotal receptor through which PCSK9 initiates an inflammatory cascade.

## Results

### PCSK9 directly activated pro-inflammatory genes independently of LDLR in vitro and in vivo

To analyze PCSK9-mediated inflammation, we first assessed the time course of p65 Ser-276 phosphorylation, which was used as the marker for NF-κB activation in response to the CAP1 ligand, resistin[8]. Recombinant human PCSK9 (rhPCSK9) stimulation (0, 50, 200, 2000 ng/mL) induced p65 phosphorylation in human monocytes and ECs in a dose-dependent manner (Fig. 1a). Next, we performed a reporter assay with a luciferase reporter plasmid driven by tandem NF-κB binding sites in HEK293T cells to investigate whether rhPCSK9 directly switched on NF-κB-mediated transcription. NF-κB transcriptional activity in monocytes was induced by PCSK9 (*P* = 0.012, *P* < 0.001, respectively) and by resistin or tumor necrosis factor-α (TNF-α) at 12 h (Fig. 1b). PCSK9 treatment increased the mRNA levels of pro-inflammatory cytokines and adhesion molecules, including *TNF-α*, interleukin *(IL)*−*1β*, *IL-6*, *ITGA4*, *ITGB1* in a dose-dependent manner but it did not increase *IL-10* transcription (Fig. 1c). PCSK9 also induced *TNF-α*, *IL-1β*, *IL-6* and C-reactive protein (*CRP*) expression in hepatocytes, the main producers of PCSK9 (Fig. 1d). PCSK9 treatment in human ECs led to an increase in the mRNA levels of pro-inflammatory cytokines such as *TNF-α*, *IL-1β*, *IL-6* and induction of adhesion molecules, including *VCAM1*, *ICAM1*, and *SELE* (Fig. 1e). The very late antigen-4 (VLA-4)[9,10] adhesion molecules, consisting of integrin-α4 and integrin-β1 and expressed on the immune cell surface, promote the inflammatory response. PCSK9 treatment not only increased the protein levels of integrin-α4 and integrin-β1 (Fig. 1f) in monocytes but also enhanced VLA-4 activation by 16.0% compared with only 2.7% in the vehicle group (Fig. 1g). The protein levels of vascular cell adhesion molecule 1 (VCAM-1) and intercellular adhesion molecule 1 (ICAM-1) in ECs also increased with PCSK9 treatment in a dose-dependent manner (Fig. 1f).

To rule out the involvement of LDLR on PCSK9-induced inflammation, we examined the effect of rhPCSK9 on bone marrow-derived macrophages (BMDMs) from *Ldlr*−/− mice. PCSK9 treatment activated NF-κB in BMDMs from wild-type (WT; BL6 mice) and *Ldlr*−/− mice (Fig. 1h) and induced pro-inflammatory cytokines, such as *TNF-α*, *IL-1β*, and *IL-6*, implying that PCSK9-mediated inflammation was independent of LDLR (Fig. 1i).

To confirm this direct inflammatory action of PCSK9 in vivo, we used an *Ldlr*−/− mouse model of atherosclerosis by partial ligation of the carotid artery (Fig. 2a). The plaque area of the right carotid artery exposed to disturbed blood flow (because of partial ligation, D-flow) was significantly broader in the adenovirus (AdV)-PCSK9 injection

group (44.4%) than the control AdV-CTRL group (22.5%) (*P* = 0.001, Fig. 2b). Furthermore, in cross-sectional analysis, the AdV-PCSK9 injection group showed greater plaque volume and arterial thickness than the control AdV-CTRL group (Fig. 2c). Systemic administration of AdV-PCSK9 in *Ldlr*−/− mice substantially increased the area of fibrosis (Masson's trichrome stain) and lipid accumulation (Oil red O) in the arteries exposed to D-flow (Fig. 2d, e). Furthermore, AdV-PCSK9 increased the number of terminal deoxynucleotidyl transferase dUTP nick end labeling (TUNEL)-positive cells in the arteries exposed to D-flow in *Ldlr*−/− mice (Fig. 2f). The overall PCSK9 expression was greater in the arteries injected with AdV-PCSK9 in *Ldlr*−/− mice than in those injected with AdV-CTRL. This difference further increased in the ligated carotid artery exposed to D-flow (*P* = 0.067; Fig. 2g). Additionally, compared with AdV-CTRL, AdV-PCSK9 administration significantly increased the infiltration of macrophages in the arteries exposed to D-flow (*P* = 0.011; Fig. 2g). The expression of inflammatory cytokines (*TNF-α*, *IL-1β*, and *IL-6*) in the carotid artery was significantly higher in AdV-PCSK9-injected mice than in AdV-CTRL-injected mice (Fig. 2h). These observations in the *Ldlr*−/− mice suggested that PCSK9-induced NF-κB-mediated inflammation and atherosclerosis directly and not via LDLR.

We examined the main source of PCSK9 after systemic administration of AdV-PCSK9 in the *Ldlr*−/− mice. As AdV mainly infects the liver[11], we observed that the liver was the main source of PCSK9 after AdV-PCSK9 administration, exhibiting overexpression of adhesion molecules (integrin-α4, integrin-β1, VCAM-1, and ICAM-1) compared with AdV-CTRL (Supplementary Fig. 1a). In contrast, bone marrow cells, such as BMDMs, were not effectively infected with AdV-PCSK9 after systemic administration (Supplementary Fig. 1b).

### CAP1 is required for PCSK9-mediated inflammation

In our previous report[7], we demonstrated the direct binding of PCSK9 and CAP1 in hepatocytes. We evaluated this phenomenon in monocytes and found that PCSK9 treatment changed CAP1 localization to the membrane in monocytes (*P* < 0.001, *P* < 0.001, *P* < 0.001, respectively) (Fig. 3a). Through immunofluorescence, colocalization of PCSK9 and CAP1 was observed mainly in the cell membrane, and this colocalization further increased 60 min after rhPCSK9 treatment in THP-1 cells (Fig. 3b). Toll-like receptor 4 (TLR4) and lectin-type oxidized LDLR1 (LOX1) are reported to mediate PCSK9-induced inflammation[12,13]. A direct binding assay using the BLItz system[7] showed high binding (nm) between PCSK9 and CAP1 ($K_d$ = 0.032 μM) compared with binding between PCSK9 and TLR4 ($K_d$ = 0.037 μM) or binding between PCSK9 and LOX1 ($K_d$ = 2.833 μM) (Fig. 3c). These results indicate that CAP1 is the major receptor of PCSK9, TLR4 is the minor receptor, and LOX1 is not a receptor.

To determine whether PCSK9 directly binds to CAP1, we used THP-1 lysates along with CAP1 and PCSK9 antibodies for immunoprecipitation (Fig. 3d). As expected, the PCSK9-CAP1 interaction was not perturbed in monocytes even with TLR4 or LOX1 knockdown (Fig. 3e). In contrast, the interaction of PCSK9 with TLR4 was perturbed when CAP1 was knocked down (Fig. 3f). Furthermore, we investigated the colocalization of PCSK9 and CAP1 in the context of atherosclerosis in an *Ldlr*−/− mouse model using partial carotid ligation. Immunostaining of arterial tissues revealed that the colocalization of PCSK9 and CAP1 increased in AdV-PCSK9-treated mice and was further enhanced under D-flow conditions by partial ligation (Fig. 3g). These observations suggest that CAP1 is a crucial factor that mediates the inflammatory action of PCSK9 in monocytes.

Treatment with rhPCSK9 (2 μg/mL) activated NF-κB in TLR4-deficient monocytes but not in CAP1-deficient monocytes (Fig. 4a). Furthermore, rhPCSK9 activated pro-inflammatory cytokines (TNF-α, IL-1β, and IL-6) in TLR4-deficient monocytes but not in CAP1-deficient monocytes (Fig. 4b). These findings implied that CAP1 is the major mediator of PCSK9-induced inflammation. Next, we performed a NF-

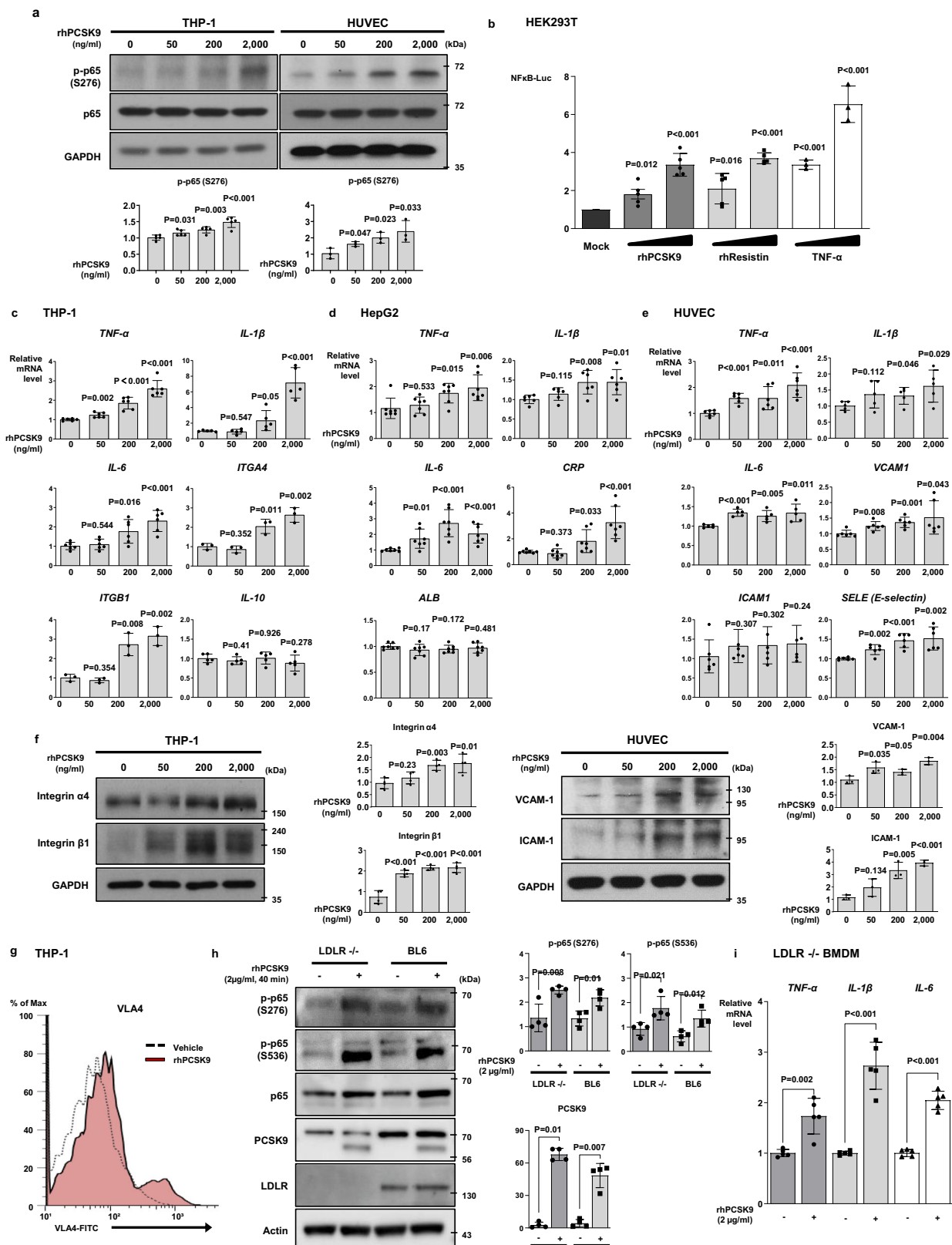

κB reporter assay to determine whether CAP1 directly affects NF-κB transcriptional activity after PCSK9 stimulation. rhPCSK9 treatment induced NF-κB luciferase activity in *CAP1*^(+/+) HEK293T cells (*P* < 0.001), which was significantly reduced in *CAP1*^(−/−) HEK293T cells (*P* < 0.001; Fig. 4c). The mRNA levels of pro-inflammatory cytokines (*TNF-α*, *IL-1β*, and *IL-6*) and adhesion molecules (*ITGA4* and *ITGB1*) in control

monocytes increased significantly after PCSK9 treatment but were significantly reduced in CAP1-knocked down monocytes (Fig. 4d).

We further investigated whether PCSK9 enhanced monocyte-EC adhesion and trans-endothelial migration. PCSK9-induced VLA-4 expression in monocytes was significantly attenuated in CAP1-deficient monocytes (Fig. 4e). Next, we performed fluorescence-activated cell

**Fig. 1 | PCSK9 directly switched on pro-inflammatory genes and increased the expression of adhesion molecules. a** Immunoblot analysis demonstrated that proprotein convertase subtilisin/kexin type-9 (PCSK9) activated and phosphorylated nuclear factor (NF)-κB p65 in THP-1 ($N = 5$) and human umbilical vein endothelial cells (HUVECs, $N = 3$) in a dose-dependent manner (0, 50, 200, and 2000 ng/mL). **b** Luciferase assay demonstrating that the NF-κB gene promoter was significantly activated ($P < 0.001$) after 12-h treatment with PCSK9 (200, 2000 ng/mL, $N = 5$), resistin (10, 50 ng/mL, $N = 5$) or TNF-α (10, 20 ng/mL, $N = 3$) in HEK293T cells. **c** qPCR analysis demonstrating that PCSK9 significantly increased the expression of cytokines, including TNF-α ($N = 6$), IL-1β ($N = 5$), IL-6 ($N = 6$) and IL-10 ($N = 5$), and integrin-α4 ($N = 3$) and -β1($N = 3$) in a dose-dependent manner (0, 50, 200, and 2000 ng/mL) in monocytes. **d** qPCR analysis demonstrating that PCSK9 significantly increased the expression of cytokines, including TNF-α ($N = 7$), IL-1β ($N = 6$), and IL-6 ($N = 7$) and C-reactive protein (CRP, $N = 7$) in a dose-dependent manner (0, 50, 200, and 2000 ng/mL), while albumin ($N = 7$) was not increased in hepatocytes. **e** qPCR analysis demonstrated that PCSK9 significantly increased the expression of pro-inflammatory cytokines, including TNF-α ($N = 6$), IL-1β ($N = 5$), and IL-6 ($N = 5$), as well as adhesion molecules (VCAM-1 ($N = 6$), ICAM-1 ($N = 6$), E-selectin [SELE, $N = 6$]) in HUVECs. **f** Immunoblot analysis demonstrating that PCSK9 increased the protein levels of integrin-α4 and -β1 in THP-1 cells ($N = 4$) and VCAM-1 and ICAM-1 in HUVECs ($N = 3$) in a dose-dependent manner (0, 50, 200, and 2000 ng/mL). **g** Fluorescence-activated cell sorting analysis reveals a 16% increase in VLA-4 activation on monocyte surfaces with PCSK9 treatment, compared to 2.7% in the vehicle group. **h** Immunoblot analysis demonstrating that PCSK9 treatment (2 μg/mL) for 40 min activated and phosphorylated NF-κB in BMDMs from $Ldlr^{-/-}$ mice and BL6 control mice ($N = 4$). **i** qPCR analyses revealed PCSK9-induced cytokines (TNF-α, IL-1β, and IL-6) even in $Ldlr^{-/-}$ BMDMs ($N = 5$) after 24-h treatment (2 μg/mL). The differences between the groups were compared using the unpaired $t$-test (two-tailed). All experiments were independently performed, and data are presented as mean values ± SD.

sorting (FACS) analysis to examine VLA-4 activation. PCSK9 treatment significantly increased VLA-4 surface expression in monocytes treated with control siRNA (from 2.3% to 17.4%); however, the same effect was not observed in CAP1-knocked down cells (change from 3.2% to 6.5%) (Fig. 4f). In ECs, rhPCSK9 treatment increased the protein levels of adhesion molecules VCAM-1 and ICAM-1, whereas this was not observed in CAP1-deficient ECs (Fig. 4g). Finally, PCSK9 enhanced the interaction between monocytes and ECs, which was significantly reduced in monocytes treated with CAP1 siRNA (Fig. 4h).

rhPCSK9 treatment enhanced oxidized LDL (ox-LDL) uptake in monocytes, leading to foam cell formation, which is another important function of monocytes in atherosclerosis; however, ox-LDL uptake was reduced in CAP1-deficient monocytes (Fig. 5a). Concomitantly, rhPCSK9 treatment significantly induced the expression of scavenger receptors (OLR1, CD36, and scavenger receptor 1 [SRA1]) in control monocytes, which was blocked in CAP1-deficient monocytes (Fig. 5b, c). Bone marrow was extracted from WT and CAP1-heterozygous knockout mice ($Cap1^{+/-}$ mice) and the cells were differentiated into BMDMs using monocyte-colony stimulating factor (M-CSF; 50 ng/mL). After ox-LDL treatment (20 μg/mL), lipid aggregation and transformation into lipid-laden macrophages increased significantly in a time-dependent manner in $Cap1^{+/+}$ BMDMs (P < 0.001), whereas lipid aggregation and transformation were inhibited in $Cap1^{+/-}$ BMDMs ($P = 0.554$) (Fig. 5d). However, the action of LPS, such as the induction of NF-κB and downstream signaling pathways was preserved in CAP1-deficient monocytes (Supplementary Fig. 2).

To investigate whether the pro-inflammatory environment induced PCSK9 expression, we analyzed the transcription levels of *PCSK9* secreted from monocytes upon exposure to various inflammatory stimuli. PCSK9 expression was significantly induced by pro-inflammatory cytokines (TNF-α, IL-1β, IL-6; $P < 0.001$, $P < 0.001$, $P < 0.001$, respectively) and LPS ($P < 0.001$; Fig. 5e). In addition, treatment with rhPCSK9 and another CAP1 ligand, resistin, showed a significant increase in *PCSK9* expression ($P < 0.001$ and $P < 0.001$, respectively), suggesting that PCSK9 is induced not only in the pro-inflammatory environment but also due to the autocrine effect of PCSK9 (Fig. 5e). Induction of PCSK9 secretion after treatment with rhPCSK9 was significantly attenuated in $Cap1^{+/-}$ BMDMs compared with that in BMDMs from the $Cap1^{+/+}$ mice, suggesting that the positive feedback loop where 'PCSK9 inducing PCSK9' was dependent on CAP1 (Fig. 5f). The mechanism underlying the positive feedback loop of PCSK9 involved the induction of the upstream regulator sterol regulatory element binding protein-2 (SREBP-2) by PCSK9, which was also blocked in CAP1-deficient monocytes (Fig. 5g).

### PCSK9-CAP1 interaction induced inflammation: the role of Syk and PKCδ

To identify the downstream molecules of the PCSK9-CAP1 interaction, we performed a pull-down assay and detected potential binding proteins to human CAP1 whose carboxyl-terminal end was conjugated with the Fc region of mouse immunoglobulin (denoted as CAP1-mFc). The CAP1-mFc fusion was expressed and purified to homogeneity after transfecting CAP1-mFc DNA into human Expi293F cells. The THP-1 lysate was incubated with either the mFc or CAP1-mFc proteins and pulled down with mFc-specific beads for quantitative proteomic analysis using liquid chromatography-tandem mass spectrometry (LC-MS/MS)[14]. CAP1-binding proteins were identified based on the Significance Analysis of INTeractome (SAINT) algorithm (Fig. 6a). Among the 2,103 proteins that were pulled down by CAP1-mFc, 464 proteins with SAINT AvgP > 0.6 were sorted and analyzed using the Database for Annotation, Visualization, and Integrated Discovery (DAVID) with an EASE score threshold of <0.1 for Gene Ontology (GO) term analysis. Following GO enrichment analysis, the enriched terms (level 1 of GO terms) in each of three categories (biological process, cellular component, and molecular function) were presented in Fig. 6b. PCSK9, whose CRD domain is structurally similar to the resistin trimer, activates the LDLR degradation pathway by binding to CAP1[7]. We also established that CAP1 serves as the receptor for resistin, leading to inflammation in humans[8]. Consequently, we focused on the 'Immune system process (GO.0002376)' category and ranked the SAINT AvgP in descending order, obtaining a list of nine proteins with AvgP = 1. Among these, Syk and PKCδ were selected as potential binding candidates for CAP1 because of their kinase activities (Fig. 6c). The MS/MS spectrum was used to identify a peptide with the sequence LIATTAHEK (part of Syk) and TGVA-GEDMQDNSGTYGK (part of PKCδ) (Fig. 6d, e). A direct binding assay using the BLItz system showed an effective increase in binding between Syk and CAP1 ($K_d = 5.957$ nM) and between PKCδ and CAP1 ($K_d = 322$ nM; Fig. 6f). In addition, the interaction between CAP1 and Syk or CAP1 and PKCδ was confirmed via immunoprecipitation using THP-1 lysates (Fig. 6g).

rhPCSK9 treatment of monocytes led to the significant phosphorylation of Syk and PKCδ at ~20 min and phosphorylation of AKT from 40–60 min (Fig. 7a). As the binding of resistin with CAP1 stimulates adenylyl cyclase[8] to produce cAMP and activate NF-κB and as PCSK9 also binds to CAP1[7], we examined whether PCSK9 can activate adenylyl cyclase, PKA, and then NF-κB. PCSK9 significantly increased cAMP levels in THP-1 cells (Fig. 7b). Interestingly, cAMP induction was exclusively blocked by rottlerin (PKCδ inhibitor), but not by R406 (Syk inhibitor) or H892HCl (PKA inhibitor) (Fig. 7c). Interestingly, CAP1 knockdown significantly attenuated Syk and PKCδ phosphorylation, whereas the absence of TLR4 did not affect PCSK9-induced Syk ($P = 0.003$) and PKCδ ($P = 0.004$) activation in monocytes (Fig. 7d). Consistently, in $Ldlr^{-/-}$ BMDMs, rhPCSK9 induced the phosphorylation of Syk and PKCδ (Fig. 7e). In the $Ldlr^{-/-}$ mouse arteries, PCSK9 expression was greater in mice that received AdV-PCSK9 than in those that received AdV-CTRL ($P = 0.028$). Elevated PCSK9 levels in the arterial tissue were associated with a significant increase in the

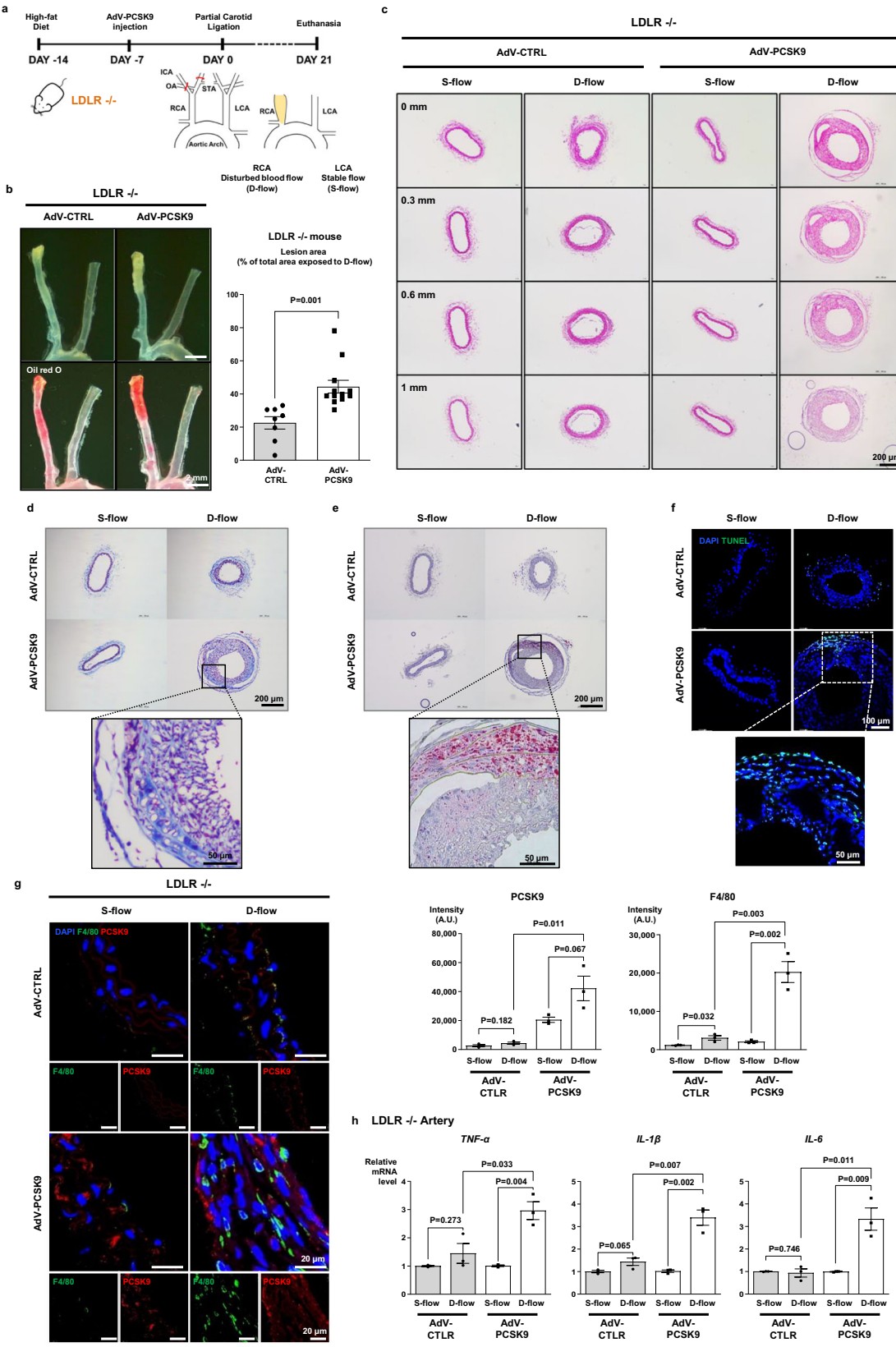

phosphorylation of Syk and PKCδ in the neointimal tissue, and this effect was further enhanced by partial ligation ($P = 0.001$, $P = 0.021$, respectively; Fig. 7f). These observations confirmed that PCSK9-induced inflammation was triggered by its binding to CAP1, followed by downstream activation of Syk and PKCδ, and that this mechanism was independent of LDLR.

## CAP1 deficiency attenuated PCSK9-induced inflammation and atherosclerosis in CAP1-heterozygous knockout mice

We investigated whether CAP1 was required to induce PCSK9-mediated atherosclerosis using a model of partial ligation of the carotid artery to induce D-flow at the distal end[15] under high-fat diet conditions in WT and $Cap1^{+/-}$ mice with or without tail vein injection of

**Fig. 2 | PCSK9 directly activated the pro-inflammatory genes independently of LDLR in vitro and in vivo. a** Experimental scheme demonstrating that PCSK9 aggravated atherosclerosis in *Ldlr⁻/⁻* mice. AdV-PCSK9 at $1 \times 10^{11}$ infectious units/ mouse was intravenously administered to mice under a high-fat diet. Partial ligation at the distal end exposed the right common carotid artery (RCA) to disturbed blood flow (D-flow), accelerating atherosclerosis. The left common carotid artery (LCA) remained under stable flow (S-flow) (AdV-CTRL *N* = 8; AdV-PCSK9 *N* = 12). **b** Oil red O staining of whole carotid arteries shows atherosclerotic plaque in the partial ligation-induced carotid atherosclerosis in *Ldlr⁻/⁻* mice. The lesion area was significantly broader in AdV-PCSK9-treated *Ldlr⁻/⁻* mice (44.4% of total RCA) than in AdV-control mice (CTRL) (22.5%). The scale bar represents 2 mm. **c**–**g** Carotid artery cross-section staining shows atherosclerotic plaque development in partial ligation-induced atherosclerosis in *Ldlr⁻/⁻* mice. Enlarged atherosclerotic plaques were observed in the arteries of AdV-PCSK9-treated mice under D-flow compared with those of AdV-CTRL, indicating the significant impact of PCSK9 on atherosclerosis. **c** Hematoxylin and eosin staining of serial sections from the aortic root at 0.3, 0.6, and 1 mm. The scale bar represents 200 μm. **d** Masson's trichrome staining. The scale bar represents 200 μm, and scale bars of magnified fields represent 50 μm. **e** Oil red O staining. The scale bar represents 200 μm, and scale bars of magnified fields represent 50 μm. **f** Immunofluorescence images stained with TUNEL (green). The scale bar represents 100 μm, and scale bars of magnified fields represent 50 μm. **g** Immunofluorescence images stained with F4/80 (green) and PCSK9 (red) in *Ldlr⁻/⁻*, demonstrating significantly elevated PCSK9 expression in AdV-PCSK9-injected mice and increased F4/80 expression under D-flow. Each scale bar represents 20 μm (*N* = 3). **h** qPCR analysis of the carotid artery from *Ldlr⁻/⁻* mice revealed significantly higher expression of inflammatory cytokines (TNF-α, IL-1β, and IL-6) in AdV-PCSK9 compared with AdV-CTRL (*N* = 3). The differences between the groups were compared using the unpaired *t*-test (two-tailed). All experiments are independently performed and all data are presented as mean values ± SEM.

AdV-PCSK9[8] (Fig. 8a). AdV-PCSK9 injection significantly increased the plasma PCSK9 level ($P < 0.001$, $P = 0.002$, respectively) and atherosclerotic plaque area ($P < 0.001$, $P < 0.001$, respectively), as observed using Oil red O staining in WT mice. However, in *Cap1⁺/⁻* mice[8], AdV-PCSK9 injection significantly increased the plasma PCSK9 level but showed lesser aggravation of atherosclerosis than that in WT mice. The plaque area was smaller in *Cap1⁺/⁻* mice than in WT mice (Fig. 8b, c). Furthermore, we analyzed the expression of pro-inflammatory cytokines in the atherosclerotic artery exposed to D-flow and S-flow in WT and *Cap1⁺/⁻* mice which received AdV-PCSK9 injection. Expression of *TNF-α*, *IL-1β*, and *IL-6* in the atherosclerotic arteries was significantly lower in *Cap1⁺/⁻* mice than in WT mice (Fig. 8d). In the cross-sectional histological analysis, compared with the AdV-CTRL injection group, AdV-PCSK9 injection led to an increase in plaque volume and arterial thickness, but these effects were attenuated in *Cap1⁺/⁻* mice (Fig. 8e). Additionally, atherosclerosis plaques with fibrosis area and lipid accumulation increased in WT mice injected with AdV-PCSK9 (Fig. 8f) and the number of apoptotic cells increased in arteries exposed to D-flow in WT mice which received AdV-PCSK9 injection (Fig. 8g). Remarkably, in *Cap1⁺/⁻* mice, all of the aforementioned effects were completely prevented, even in the arteries exposed to D-flow (Fig. 8f, g).

To elucidate the underlying mechanisms, we performed analyses on ligated arteries exposed to D-flow. AdV-PCSK9 injection led to a significant increase in PCSK9 levels in the neointima, with infiltrated macrophages showing a significant colocalization with PCSK9 ($P < 0.001$, $P < 0.001$, respectively). In contrast, in *Cap1⁺/⁻* mice, both PCSK9 levels and infiltrated macrophages were markedly reduced ($P < 0.001$, $P < 0.001$, respectively) (Fig. 9a). Furthermore, elevated serum levels of PCSK9 led to a significant increase in Syk and PKCδ phosphorylation in vivo ($P = 0.047$, $P < 0.001$, respectively), both of which were markedly reduced in *Cap1⁺/⁻* mice ($P = 0.028$, $P = 0.005$, respectively) (Fig. 9b). Next, to elucidate the cellular localization of CAP1 and PCSK9 within the atherosclerotic plaque, we conducted immunofluorescence staining of infiltrated macrophages, ECs, and SMCs, which constitute atherosclerotic plaques. We observed that PCSK9 and CAP1 were expressed in atherosclerotic plaques and confirmed their colocalization on macrophages (Fig. 9c), ECs (Fig. 9d), and SMCs (Fig. 9e). Notably, SMCs within the atherosclerotic milieu exhibited characteristics akin to immunocyte-like cells, consistent with previous research[16,17]. PCSK9 expression was identified in both SMA⁺F4/80⁺ and SMA⁺F4/80⁻ cell populations, emphasizing its relevance across these cell types (Fig. 9f).

As summarized in Fig. 9g, PCSK9 binds to CAP1, leading to the activation of Syk and PKCδ and induction of inflammatory gene cascades, TLR4, and scavenger receptors on mono-macrophages and adhesion molecules on ECs. These actions of PCSK9 were dependent on CAP1 and thus attenuated by CAP1 depletion. In vivo, systemic injection of AdV-PCSK9 significantly induced atherosclerosis of the carotid artery exposed to D-flow, which was prevented in *Cap⁺/⁻* mice. These results suggested that the PCSK9-CAP1-PKCδ/Syk pathway may be a viable target for developing new therapeutics for dyslipidemia, atherosclerotic cardiovascular diseases, and inflammation-based diseases.

## Serum PCSK9 levels in coronary artery disease (CAD) patients correlate with Syk, PKC, and NF-κB phosphorylation in peripheral blood mononuclear cells (PBMCs), and CAP1-hFc could block PCSK9-mediated inflammatory signals

To investigate the potential association between human serum PCSK9 levels and its effect on inflammatory signaling pathways, blood samples were collected from individuals diagnosed with CAD and healthy donors. CAD patients were all under strict statin treatment. Pearson's correlation and simple linear regression analyses were used to examine the relationship between PCSK9 concentration in the serum and phosphorylation of Syk, PKCδ, p65(S276), and p65(s536) in PBMCs. Interestingly, serum PCSK9 concentration showed a positive correlation with the quantified phosphorylation of Syk ($P = 0.03$, $R = 0.498$), PKCδ ($P = 0.049$, $R = 0.458$), p65 (S276, $P = 0.168$, $R = 0.33$), and p65 (S536, $P = 0.022$, $R = 0.52$) in matched PBMCs (Fig. 10a and Supplementary Fig. 3). In Fig. 10a, the data from CAD patients (indicated by black dots) exhibited a predominant distribution on the upper-right side of the slope, because of their higher PCSK9 concentration (409.4 ng/mL) and higher phosphorylation of signaling proteins compared with healthy donors whose distribution was on the lower-left side (indicated by pink dots, 290.1 ng/mL). Interestingly, such a higher level of PCSK9 in patients with CAD than healthy volunteers was well contrasted with lower cholesterol and lipid parameters in CAD patients who were under treatment with statins with or without fibrates (Fig. 10b). The action mechanism of statin is to block cholesterol synthesis in hepatocytes, leading to the increased synthesis of LDLR through SREBP2, which also induces PCSK9. Thus, CAD patients under statin treatment showed low cholesterol and high PCSK9 levels in their serum.

Subsequently, we conducted experiments to ascertain whether the PCSK9-induced direct inflammatory response was attenuated by evolocumab, a well-known antibody that inhibits the binding of PCSK9 and LDLR. We compared evolocumab with our new therapeutic competitive inhibitor against binding of PCSK9 and CAP1, CAP1-hFc, a fusion of human CAP1 with the Fc region of human immunoglobulin. We performed a competitive binding assay using evolocumab and CAP1-hFc (Fig. 10c). Our findings demonstrated that treatment with CAP1-hFc at concentrations of 10 μM and 50 μM notably disrupted the interaction between PCSK9 and CAP1, with a significant effect observed at 50 μM ($P = 0.643$ and $P = 0.011$, respectively). In contrast, treatment with evolocumab at concentrations of 10 μM and 50 μM exhibited the opposite effect by enhancing this interaction ($P = 0.435$ and $P = 0.162$, respectively) (Fig. 10c). The binding of PCSK9

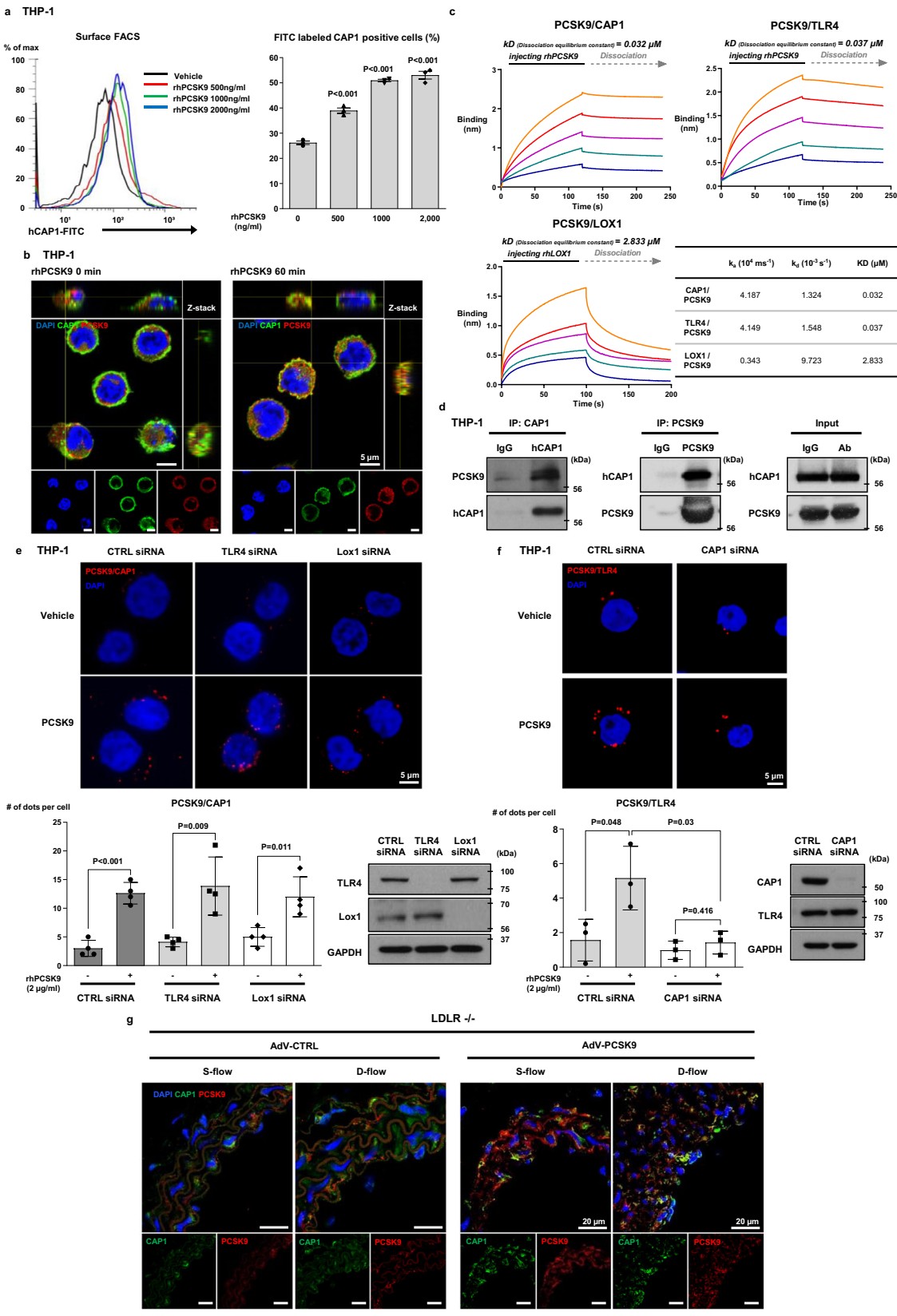

and CAP1 was not blocked by evolocumab but was blocked by CAP1-hFc.

In monocytes, rhPCSK9 treatment induced significant phosphorylation of p65 (S276 and S536), which was more significantly blocked by CAP1-hFc ($P = 0.037$ and $0.002$, respectively) than those with evolocumab ($P = 0.534$ and $P = 0.257$, respectively) (Fig. 10d). PCSK9 significantly induced the phosphorylation of Syk, PKCδ, and p65 (S276 and S536) ($P = 0.007$, $P = 0.012$, $P = 0.025$, and $P = 0.003$, respectively) in human PBMC-derived macrophages, which was more effectively and significantly attenuated by CAP1-hFc ($P = 0.045$, $P = 0.006$, $P = 0.004$, and $P = 0.003$, respectively) than by evolocumab ($P = 0.096$, $P = 0.062$, $P = 0.034$, and $P = 0.013$, respectively) (Fig. 10e).

**Fig. 3 | CAP1 is a receptor of PCSK9. a** Fluorescence-activated cell sorting analysis of THP-1 cells demonstrating that treatment of PCSK9 for 1 h changed CAP1 localization to the membrane surface of monocytes in a dose-dependent manner (0, 500, 1000, and 2000 ng/mL) ($N=3$). **b** Immunofluorescent staining of THP-1 cells with CAP1 (green) and PCSK9 (red) demonstrating their colocalization (left panel) mainly to the membrane surface of monocytes (yellow). The colocalization between PCSK9 and CAP1 further increased 60 min after treatment with 2 µg/mL rhPCSK9 (right panel). Colocalization analysis within the membrane was performed using orthogonal views from different planes of confocal microscope images. Scale bar represents 5 µm. **c** A direct binding assay using the BLItz system showed that the binding affinity to PCSK9 was strongest for CAP1 (0.032 µM), intermediate for TLR4 (0.037 µM), and weakest for LOX1 (2.833 µM). **d** Immunoprecipitation analysis of THP-1 cells demonstrating the interaction between PCSK9 and CAP1 in monocytes. **e** Duolink Proximity Ligation Assay for detecting the interaction between PCSK9 and CAP1 after treatment with CTRL siRNA, TLR4 siRNA, or LOX1 siRNA. The interaction between PCSK9 and CAP1 was quantified by counting the red dots. The Proximity Ligation Assay showed that the interaction between PCSK9 and CAP1 was not affected by the presence or absence of TLR4 or LOX1 ($N=4$). The scale bar represents 5 µm. **f** Duolink Proximity Ligation Assay for detecting the interaction between PCSK9 and TLR4 with CTRL or CAP1 siRNA. The extent of the interaction between PCSK9 and TLR4 was quantified by counting the red dots ($N=3$). The scale bar represents 5 µm. **g** Immunofluorescence staining of CAP1 (green) and PCSK9 (red) in the arteries of AdV-CTRL or AdV-PCSK9-treated *Ldlr*$^{-/-}$ mice under S-flow and D-flow. CAP1 and PCSK9 were colocalized in all groups. CAP1 and PCSK9 expression increased in the AdV-PCSK9 injection group compared with that in the AdV-CTRL group under D-flow. The scale bar represents 20 µm. The differences between the groups were compared using the unpaired *t*-test (two-tailed). All experiments are independently performed and all data are presented as mean values ± SD.

In conclusion, CAP1 is a binding partner of PCSK9, which mediates not only caveolae-dependent endocytosis and lysosomal degradation of LDLR, but also recruits PKCδ and Syk and modulates PCSK9-mediated inflammatory signal transduction independently of LDLR. CAP1-hFc inhibits the binding of CAP1 and PCSK9, which blocks both LDLR degradation and the inflammatory signaling pathway. In contrast, PCSK9 inhibitory antibody evolocumab can only block the LDLR degradation pathway because it cannot block the binding of PCSK9 and CAP1.

## Discussion

Atherosclerosis is a complex disease characterized by LDL-C deposition, inflammation, and macrophage infiltration[18]. Binding of PCSK9 to LDLR initiates LDLR degradation, leading to an increase in LDL-C levels, progression of atherosclerosis, and cardiovascular events[1]. It is thought that PCSK9 increases LDL-C and has an independent pro-inflammatory effect. In this study, we provided evidence that PCSK9 directly induces inflammatory genes in monocytes and atherosclerosis independently of LDLR in *Ldlr*$^{-/-}$ mice (Figs. 1 and 2). In addition to NF-κB activation and induction of pro-inflammatory cytokines, including TNF-α, IL-1β, and IL-6, PCSK9 directly induced the expression of adhesion molecules such as VLA-4 on monocytes, as well as VCAM-1 and ICAM-1 on ECs, leading to the enhanced interaction of these two cell types, which guides the initial steps in atherosclerosis development. This inflammatory action of PCSK9 was maintained in the monocytes of *Ldlr*$^{-/-}$ mice and systemic administration of AdV-PCSK9 significantly increased atherosclerotic plaque formation. Thus, PCSK9 directly induces inflammation independently of LDLR.

Resistin, a peptide hormone rich in cysteine, initiates diverse pathways within cells to prompt vascular inflammation, the buildup of lipids, and heightened susceptibility of plaques. This positions resistin as a possible biomarker and treatment target for atherosclerosis[19]. We have previously reported that CAP1, the receptor for human resistin, leads to NF-κB activation via the cAMP/PKA pathway[8]. Interestingly, *CAP1* mRNA expression is reported to be not only significantly increased in CAD patients[20,21], but also positively correlated with the carotid intima-media thickness in patients with end-stage renal disease[21]. We observed that CAP1 directly binds to the C-terminal cysteine-rich domain of PCSK9, which is structurally similar to the globular C-terminal of the resistin trimer[7]. Therefore, we speculated that PCSK9 binding to CAP1 may switch on pro-inflammatory signaling and further aggravate atherosclerosis independently of LDLR. Here, we confirmed that CAP1, unlike LDLR, is required for the pro-inflammatory action of PCSK9 (Figs. 3, 4, and 5). We again demonstrated that PCSK9 directly binds to CAP1 in monocytes and that CAP1 deletion can block the induction of inflammatory genes by PCSK9. Previous studies have suggested the possible involvement of TLR4 and LOX1 in

PCSK9-mediated inflammation[12,13] or the upregulation of TLR4 and scavenger receptors (LOX1, CD36, SRA1) by PCSK9, ultimately resulting in the activation of NF-κB and inflammatory cytokines and induction of ox-LDL uptake[22,23]. To verify this, we first determined the main binding partner of PCSK9. A direct binding assay using the BLItz system showed that the binding affinity to PCSK9 was strongest for CAP1, intermediate for TLR4, and weakest for LOX1, which excluded LOX1 as the receptor for PCSK9. In the proximity ligation assay, the interaction between PCSK9 and CAP1 was robust and was not affected by TLR4 or LOX1, whereas the interaction between PCSK9 and TLR4 was dependent on CAP1 and decreased after CAP1 depletion. We then compared CAP1 and TLR4 in terms of PCSK9-induced inflammatory signaling, which was significantly blocked in CAP1-deficient monocytes, but remained unchanged in TLR4-deficient cells (Fig. 4a). Furthermore, PCSK9-mediated induction of TLR4 and scavenger receptors (LOX1, CD36, SRA1) was also dependent on CAP1 and attenuated in CAP1-deficient monocytes (Fig. 5b, c). Finally, the PCSK9-mediated enhancement of ox-LDL uptake was also dependent on CAP1. Therefore, CAP1 is the key mediator for all the previously reported actions of PCSK9, such as the induction of TLR4, scavenger receptors (LOX1, CD36, SRA1), and inflammatory cytokines in monocytes, and ox-LDL uptake by macrophages.

Next, we investigated the downstream mechanism of interaction between PCSK9 and CAP1 using MS/MS analysis and identified PKCδ and Syk as downstream molecules (Figs. 6, 7). PCSK9 and CAP1 binding recruited and phosphorylated PKCδ and Syk. PKCδ is a serine- and threonine-specific protein kinase activated by diacylglycerol level or calcium ions and is involved in cancers and cardiovascular diseases. Therefore, PKC inhibitors reduce inflammatory diseases by suppressing LPS-stimulated pro-inflammatory cytokine production. Syk, a non-receptor tyrosine kinase, was initially known to play a crucial role in adaptive immune responses. Recently, Syk has also been shown to play a critical role in TLR4-mediated inflammation[24]. Syk binds to TLR4 and triggers various downstream cascades[25]. In primary human monocytes, PKCδ and Syk are involved in dectin-1-mediated phagocytosis[26]. In this study, CAP1, Syk, and PKCδ bound to each other, and PCSK9 treatment led to phosphorylation of Syk and PKC, followed by AKT phosphorylation. During inflammation, Syk is activated via integrin signal transduction[24], resulting in increased leukocyte adhesion to inflamed ECs. We observed that PCSK9 treatment increased VLA-4 activation in monocytes by 17.4%.

Using heterozygous *Cap1* knockout mice, we showed that CAP1 could be a therapeutic target that can prevent PCSK9-induced vascular inflammation and atherosclerosis (Figs. 8, 9). In high-fat diet-fed WT mice, systemic injection of AdV-PCSK9-induced extensive atherosclerosis with high inflammatory gene expression and macrophage infiltration in the right carotid arteries exposed to D-flow. In contrast, PCSK9-induced atherosclerosis was significantly prevented in heterozygous *Cap1* knockout mice, suggesting that CAP1 may be a viable target

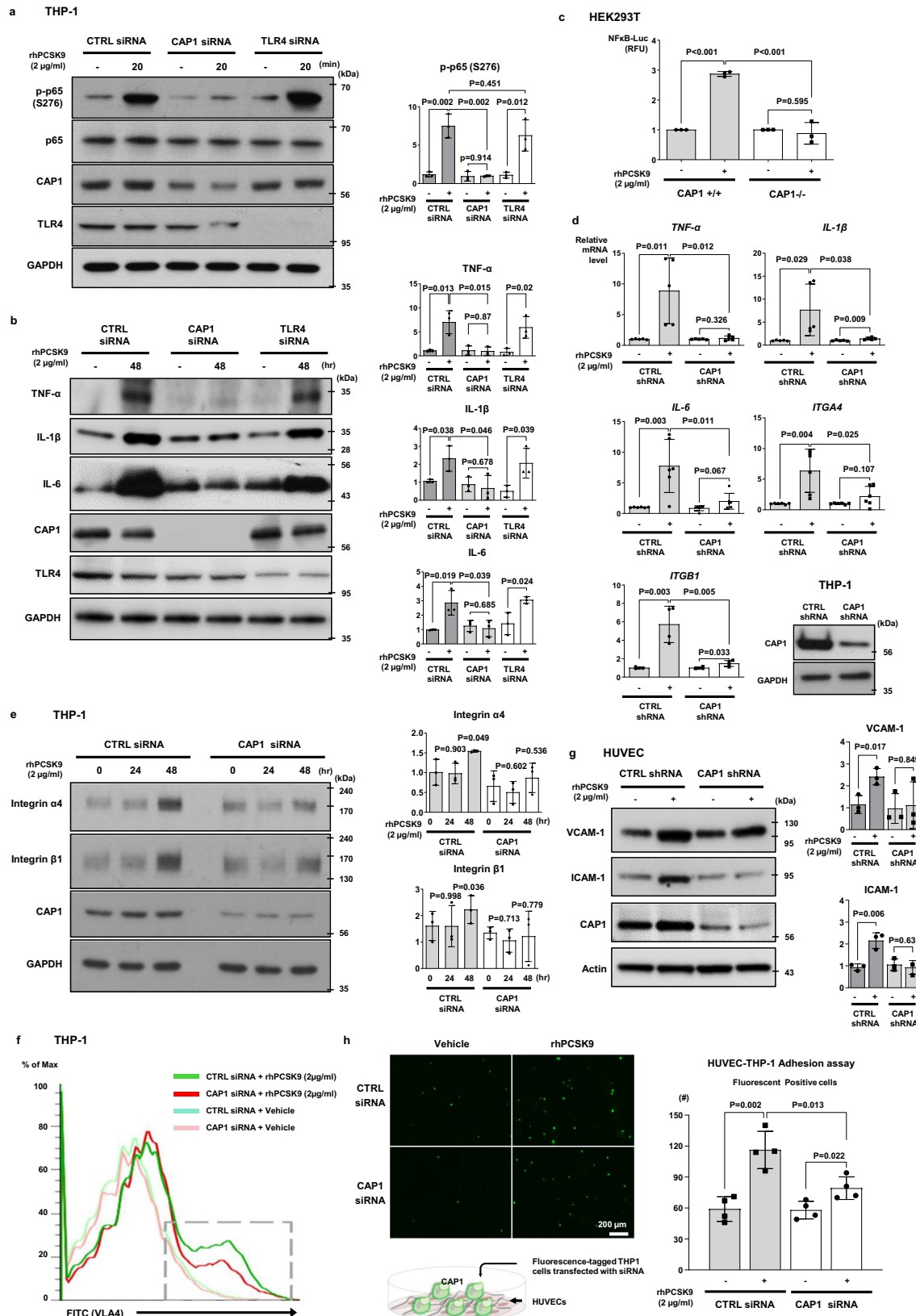

for controlling atherosclerosis. Although mainly produced in hepatocytes, PCSK9 is also expressed in human atherosclerotic plaques, especially in ECs and vascular SMCs[3]. In our study, PCSK9 treatment induced SREBP-2 activation, followed by PCSK9 expression via autocrine signaling (positive feedback loop), which was dependent on CAP1 and thus was impaired in CAP1-deficient cells. In addition, plaque macrophages express inflammatory cytokines that can enhance PCSK9 expression in ECs, VSMCs, and macrophages via SREBP-2, which enhances PCSK9 transcription[27]. Although SMCs with macrophage-like characteristics contribute to the development of atherosclerosis, their contribution is lower than that of macrophages, and the levels of inflammatory factors in these cells are also inferior to those in monocyte-derived

**Fig. 4 | CAP1 was required for PCSK9-mediated inflammation. a** Immunoblot demonstrating rhPCSK9 treatment (2 μg/mL for 20 min) resulted in NF-κB p65 phosphorylation, which was blocked in CAP1-deficient THP-1 cells, but not in TLR4-deficient THP-1 cells (N = 3). **b** rhPCSK9 treatment (2 μg/mL, 48 h) elevated TNF-α, IL-1β, and IL-6 protein levels in THP-1 cells treated with CTRL and TLR4 siRNA, but not in CAP1 siRNA (N = 3). **c** Luciferase assay demonstrating the activity of NF-κB gene promoter in *CAP1*+/+ and *CAP1*−/− 293 T cells. The activation of NF-κB signaling induced by rhPCSK9 treatment (2 μg/mL) was attenuated in *CAP1*−/− 293 T cells (N = 3). **d** qPCR demonstrating rhPCSK9-induced pro-inflammatory cytokines in THP-1 cells transfected with CTRL or CAP1 shRNA. PCSK9 treatment increased *TNF-α* (N = 5), *IL-1β* (N = 5), *IL-6* (N = 6), integrin-α4 (N = 6), and -β1 (N = 4) in CTRL group, not in CAP1-deficient cells. **e** Immunoblot to analyze the protein levels of rhPCSK9-induced adhesion molecules in THP-1 cells. PCSK9 treatment induced the integrin-α4 and -β1 expression in a time-dependent manner (0, 24, and 48 h) in THP-1 cells with CTRL siRNA, but not in with CAP1 siRNA (N = 3). **f** Fluorescence-activated cell sorting demonstrating VLA-4 activation in THP-1 cells with CTRL or CAP1 siRNA. THP-1 cells were transfected with CTRL or CAP1 siRNA and left untreated or treated with rhPCSK9 (2 μg/mL). PCSK9 treatment increased VLA-4 expression in THP-1 cells, which was reduced in CAP1-deficient THP-1 cells. **g** VCAM-1 and ICAM-1 expression increased after rhPCSK9 treatment in HUVECs transfected with CTRL shRNA, but not in with CAP1 shRNA (N = 3). **h** Cell adhesion assay of THP-1 and HUVECs with rhPCSK9 treatment demonstrating that adhesion to HUVECs was enhanced by PCSK9 in THP-1 cells with CTRL siRNA, which was blocked in CAP1-deficient THP-1 cells. Representative images of fluorescently labeled adherent THP-1 cells (upper-left panel), and fluorescence-positive cells were counted to quantify cell adhesion (upper-right panel) (N = 4). The schematic figure for adhesion assay (bottom panel). The scale bar represents 200 μm. The differences between the groups were compared using the unpaired *t*-test (two-tailed) or one-way analysis of variance. All experiments are independently performed, and data are presented as mean values ± SD.

macrophages[28]. Thus, based on the positive feedback loop between PCSK9 and cytokines involving CAP1 and SREBP-2, we expect PCSK9 to play a significant role in inflammatory vascular diseases.

Monoclonal inhibitory antibodies against PCSK9 reduced plasma LDL-C levels and improved clinical outcomes in patients with atherosclerotic cardiovascular disease. Here, we identified a pivotal mechanism for how PCSK9 directly induces inflammation and aggravates atherosclerosis independently of the changes in lipid levels. Septic shock was improved in PCSK9-knockout mice, as indicated by the decrease in inflammatory cytokine production and other physiological responses to LPS[6]. Furthermore, PCSK9 expression in the liver can be positively linked to TNF-α and interferon-γ levels, and other possible inflammation-linked pathways[29]. In contrast, antibodies such as alirocumab or evolocumab to block binding of LDLR and PCSK9 did not prevent death from LPS-induced endotoxemia in mice[30]. In addition, these inhibitory antibodies against PCSK9 could not reduce inflammatory markers such as high-sensitivity CRP, IL-6, IL-1B, and others in a human cohort[31–34]. This discrepancy, that PCSK9 is associated with inflammation whereas evolocumab or alirocumab does not reduce inflammation[35,36], could be explained by the model of triple complex, "LDLR-PCSK9-CAP1" (Fig. 10f). The catalytic domain of PCSK9 binds to the epidermal growth factor (EGF) precursor homology domain of LDLR[37], whereas the cysteine-rich domain of PCSK9 binds to the SH3 domain of CAP1 (Fig. 10f). The triple complex is internalized through the caveosome because CAP1 binds to caveolin-1, resulting in the lysosomal degradation of LDLR[7]. Evolocumab only inhibits the binding of PCSK9 and LDLR, leading to salvage of LDLR and a lipid-lowering effect, while it cannot block the binding of PCSK9 and CAP1, and thus the inflammatory signal still turns on (Fig. 10c, d). However, CAP1-hFc, a fusion of human CAP1 with the Fc region of human immunoglobulin, can competitively inhibit the binding of PCSK9 and CAP1, preventing the internalization and degradation of LDLR-PCSK9 complex and more importantly, leading to the blocking of the inflammatory signal (Fig. 10c). These data demonstrate that CAP1 is required for not only the LDLR degradation[7] but also for PCSK9-mediated inflammation.

We observed that PCSK9 binds to CAP1 and then activates Syk and PKCδ, leading to the inflammatory cascade. Here, we found that serum PCSK9 concentration was positively correlated with Syk, PKCδ, and p65 phosphorylation in human PBMCs (Fig. 10a). Furthermore, the average serum PCSK9 concentration was significantly higher in CAD patients than that in healthy individuals, whereas LDL-C was lower in CAD patients than healthy individuals because of the strict statin treatment, implying the presence of inflammatory residual risk even after statin therapy in patients with CAD (Fig. 10b). CAP1-hFc attenuated PCSK9-induced phosphorylation of Syk, PKCδ, and p65 more effectively than evolocumab did in human PBMC-derived macrophages, suggesting CAP1-hFc as a promising therapeutic agent for PCSK9-mediated inflammatory diseases or attenuating residual risk

even after statin therapy (Fig. 10e). Our findings will extend the clinical significance of PCSK9 from the control of LDL-C levels to the modulation of inflammatory signaling, provoking new analyses of the published landmark trials using evolocumab or alirocumab as well as designing of new clinical trials.

In conclusion, we clarified how PCSK9 induces inflammation and atherosclerosis, either directly or independently of LDLR. PCSK9-induced NF-κB and inflammatory genes in monocytes and induced atherosclerosis in *Ldlr*−/− mice. As summarized in Fig. 9g, PCSK9 binds to CAP1, leading to the activation of Syk and PKCδ and induction of inflammatory gene cascades, TLR4, and scavenger receptors on mono-macrophages and adhesion molecules on ECs. These functions of PCSK9 were dependent on CAP1 and thus attenuated by CAP1 depletion. Systemic injection of AdV-PCSK9 significantly induced atherosclerosis of the carotid artery exposed to D-flow in vivo, which was prevented in *Cap*+/− mice. Finally, the human serum level of PCSK9 correlated well with the degree of phosphorylation of these signaling proteins in PBMCs. These results suggest that the PCSK9-CAP1-Syk/PKCδ pathway may be a viable target for developing new therapeutics for dyslipidemia, atherosclerotic cardiovascular, and inflammation-based diseases.

## Methods

### Antibodies and chemicals

The primary antibodies used in this study were as follows: anti-NF-κB p65 (Santa Cruz Biotechnology, Santa Cruz, CA, USA; sc-109; WB 1:1000), anti-human CAP1 (Santa Cruz Biotechnology; sc-100917 WB 1:2000, IF 1:100), anti-human/mouse CAP1 (Santa Cruz Biotechnology; sc-134637; WB 1:2000 IF 1:100), anti-ICAM-1 (Santa Cruz Biotechnology; sc-8439; WB 1:1000), anti-ICAM-1 (Santa Cruz Biotechnology; sc-7891; WB 1:1000), anti-GAPDH (Sigma-Aldrich, St. Louis, MO, USA; G9545; WB 1:20,000), anti-VCAM-1 (Santa Cruz Biotechnology; sc-1504; WB 1:1000), anti-PCSK9 (Cell Signaling Technology, Danvers, MA, USA; #85813; WB 1:1000, IF 1:50), anti-p-SYK (Cell Signaling Technology; #2701S; WB:1000, IF 1:50), anti-p-PKCδ (Cell Signaling Technology; #2055S; WB 1:1000, IF 1:50, Abcam, Cambridge, UK; #109539; WB 1:1000), anti-p-AKT (Cell Signaling Technology; #4060S; WB 1:1000), anti-SYK (Cell Signaling Technology; #13198S; WB 1:1000), anti-PKCδ (Cell Signaling Technology; #9616S; WB 1:1000), anti-AKT (Cell Signaling Technology; #2920S; WB 1:1000), anti-F4/80 (Cell Signaling Technology; #30325; IF 1:100), and anti-αSMA (Sigma-Aldrich, St. Louis, MO, USA; #A2547; IF 1:200). For the secondary antibody, anti-mouse IgG horseradish peroxidase (HRP; Thermo Fisher Scientific [formerly called Invitrogen], Waltham, MA, USA; #31430; WB 1:5000), anti-goat IgG HRP (Invitrogen; #31403; WB 1:3000), or anti-rabbit IgG HRP (Invitrogen; #32460; WB 1:5000), Donkey anti-Mouse IgG Antibody, Alexa Fluor™ 488 (Invitrogen; A21202; IF 1:200 - 500), Donkey anti-Mouse IgG Antibody, Alexa Fluor™ 555 (Invitrogen; A31570; IF 1:200 - 500), Donkey anti-

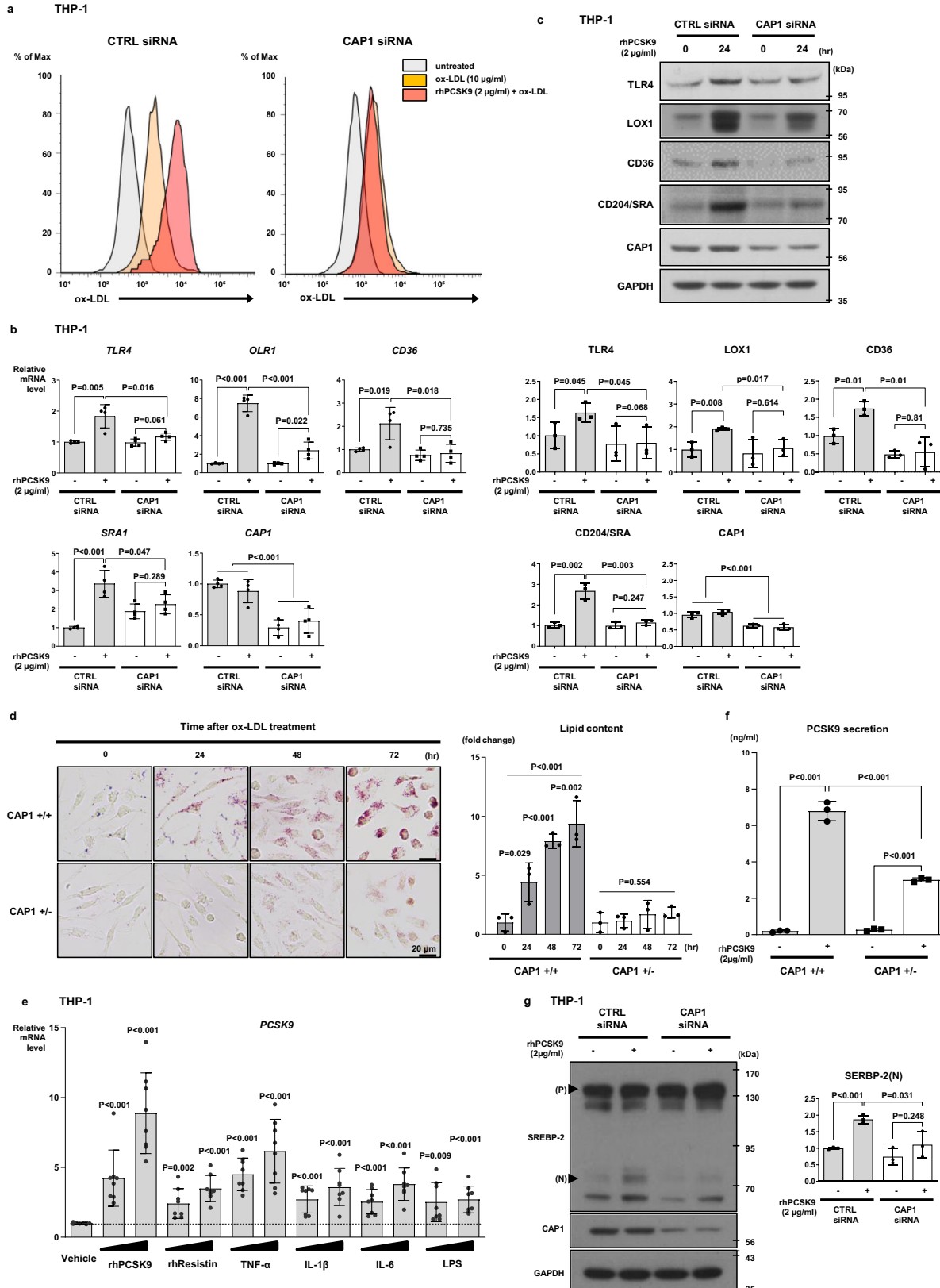

Rabbit IgG Antibody, Alexa Fluor™ 488 (Invitrogen; A21206; IF 1:200 ~ 500), Donkey anti-Rabbit IgG Antibody, Alexa Fluor™ 555 (Invitrogen; A31572; IF 1:200 ~ 500), Donkey anti-Goat IgG Antibody, Alexa Fluor™ 633 (Invitrogen; A21082; IF 1:200 ~ 500) was used. All uncropped and unprocessed scans of immunoblots are provided in Source Data file.

## Cell culture

THP-1 cells (American Type Culture Collection; TIB-202) were cultured in high-glucose RPMI medium (WellGene, Daegu, Republic of Korea) supplemented with 10% FBS (Gibco, Waltham, MA, USA) and 1X antibiotics-antimycotics (Gibco) at 37 °C in a 5% $CO_2$ incubator. Pooled human umbilical vein ECs (HUVECs; Lonza; C2519A, Basel, Switzerland)

**Fig. 5 | PCSK9 regulated the expression of scavenger receptors and ox-LDL uptake via CAP1. a** For the ox-LDL assay, THP-1 cells were transfected with CTRL or CAP1 siRNA and then treated with or without rhPCSK9. ox-LDL uptake decreased in CAP1-deficient THP-1 cells in response to PCSK9 (2 μg/mL). **b** qPCR analysis demonstrating the mRNA levels of several scavenger receptors in response to PCSK9 treatment in THP-1 cells transfected with CTRL or CAP1 siRNA. The mRNA levels of *LOX1*, *CD36*, *SRA1*, and *TLR4* increased significantly with rhPCSK9 (2 μg/mL) treatment in monocytes with CAP1, but not in THP-1 cells with CAP1 knockdown (*N* = 4). **c** Immunoblot analysis demonstrating the protein levels of several scavenger receptors in response to PCSK9 treatment in THP-1 cells transfected with CTRL or CAP1 siRNA. Consistently, the protein levels of scavenger receptors were induced by PCSK9 treatment in control THP-1 cells, but not in CAP1-deficient THP-1 cells (*N* = 3). **d** Oil Red O staining of BMDMs from *Cap1*^+/+^ and *Cap1*^+/−^ mice after differentiation into macrophages, after treatment with ox-LDL for 0, 24, 48, and 72 h. Ox-LDL treatment induced lipid formation and accumulation in *Cap1*^+/+^ BMDMs (upper panel) because of ox-LDL uptake, which was significantly inhibited in CAP1-deficient BMDMs. The scale bar represents 20 μm (*N* = 3). **e** *PCSK9* mRNA expression induced by pro-inflammatory cytokines was analyzed by qPCR. *PCSK9* mRNA levels increased in response to various inflammatory stimuli in THP-1 cells (*N* = 8). **f** hPCSK9 levels in media were determined using ELISA in *Cap1*^+/+^ and *Cap1*^+/−^ BMDMs. Positive feedback loop of PCSK9-induced PCSK9 secretion was attenuated in *Cap1*^+/−^ BMDMs (*N* = 3). **g** Immunoblot analysis of signal activation induced by PCSK9 in THP-1 cells transfected with CTRL or CAP1 siRNA demonstrating that PCSK9-induced SREBP-2 expression was attenuated when CAP1 was knocked down (*N* = 3). (P) and (N) denote the precursor and nuclear active forms of SREBP-2, respectively. The differences between the groups were compared using the unpaired *t*-test (two-tailed) or one-way analysis of variance. All experiments are independently performed and all data are presented as mean values ± SD.

were cultured using EGM-Plus SingleQuot Kit (Lonza; CC-4542) and reconstituted with 500 mL of EBM-Plus Basal Medium (CC-5036) to obtain the EGM-Plus growth medium; the medium contained 1.0 mL of bovine brain extract, 25.0 mL of L-glutamine, 0.5 mL each of ascorbic acid, hydrocortisone, rhEGF, heparin, and Gentamicin sulfate-Amphotericin, and 10% FBS (Gibco; #16000). The HUVECs used in this study were obtained from passages 6 to 7.

## NF-κB luciferase reporter assay
HEK293T cells ($2.4 \times 10^4$, American Type Culture Collection; CRL-3216) were seeded onto a clear-bottom 96-well plate (Corning, NY, USA) and incubated overnight. Subsequently, the cells were subjected to transfection with 10 ng of the pNF-κB luciferase plasmid (pGL3-basic) and 2 ng of pRL-TK (E2241) plasmid (both sourced from Promega, Madison, WI, USA) using Lipofectamine LTX (Thermo Fisher Scientific) in accordance with the manufacturer's instructions. Following a 24-h incubation, the cells were treated with PCSK9 (200, 2000 ng/mL), resistin (10, 50 ng/mL), or TNF-α (10, 20 ng/mL) for 12 h. Cell harvesting and subsequent analysis were performed using the Dual-Glo Luciferase Reporter System (Promega; E2920) as per the manufacturer's protocol. Luminescence measurements were taken using a fluorescence detector (GloMax Discover Microplate Reader, Promega). The data were obtained from three independent transfections and represented as the –fold increase in luciferase activities (mean ± SD) relative to the control.

## RNA interference and transfection
The shRNA constructs that contain target genes used in this study were cloned into *Hpa*I and *Xho*I sites of the pLL3.7 lentiviral vector (Addgene, MA, USA; #11795). The sequences targeted by constructs were as follows: human CAP1; 5′-AGATGTGGATAAGAAGCAT-3′. siCAP1 was synthesized by Bioneer (Daejeon, Republic of Korea) with the *CAP1* target sequence 5′-AAACCGAGTCCTCAAAGAGTA-3′. *TLR4* and *LOX1* siRNA oligos were obtained from Dharmacon ON-TARGETplus SMARTpool siRNA (Lafayette, CO, USA), and each consisted of a mixture of four sequences. For the target knockdown, siRNA oligos were transfected using Lipofectamine RNAiMax Transfection Reagent (Invitrogen Life Technologies) according to the manufacturer's instructions. Transfection time required for optimum CAP1 or TLR4 or TLR4 knockdown is presented in Supplementary Fig. 4.

## Adenoviral transduction
Human *PCSK9* was cloned in an adenoviral shuttling vector (pAdTrack CMV)[11] containing green fluorescence protein and finally transferred to the adenoviral genome (pAdEasy-1) via homologous recombination. The recombinant adenoviral particles were purified from HEK293A cells (American Type Culture Collection; CRL-1573) via ultracentrifugation in a cesium chloride density gradient (100,000 × g, overnight, 4 °C), followed by dialysis (dialysis buffer: 2 M MgCl₂, 3% sucrose, and 1 M Tris [pH 8.9], 5% glycerol; dialysis cassette, Thermo Fisher Scientific). The final multiplicity of infection of adenoviral PCSK9 used in this study was 100.

## Animals and carotid ligation model
All animal experiments were performed with approval from the Institutional Animal Care and Use Committee (IACUC, 17-0181-C1A0) of the Clinical Research Institute of Seoul National University Hospital, Republic of Korea. Mice were housed in a specific-pathogen-free (SPF) facility with controlled environmental conditions. This facility maintained a temperature range of 20–26 °C, with humidity levels maintained within the range of 30–70%, and adhered to a standard 12-h light/dark cycle. The mice were provided with *ad libitum* access to standard rodent chow and clean water, and their cages were equipped with suitable bedding, nesting materials, and environmental enrichment items. Routine cage cleaning and sanitation protocols were implemented to ensure a hygienic environment, and noise levels were minimized to reduce stress levels among the animals. Mice were euthanized by carbon dioxide inhalation. Age-matched (8-week-old) *Cap1*^+/+^ mice, their *Cap1*^+/−^ littermates, and *Ldlr*^−/−^ male mice were used for the carotid ligation model. Mice were fed a high-fat diet and AdV-PCSK9 ($1 \times 10^{11}$ infectious units/mouse) was administered via tail vein injection 1 week before the ligation surgery. Mice were anesthetized with Alfaxan (20 mg/kg) and Rompun (10 mg/kg) via intramuscular injection. The epilated area was disinfected with betadine, and a ventral midline incision (4–5 mm) was made in the neck. The left carotid artery (LCA) was exposed by blunt dissection. All three branches of the right common carotid artery (RCA) (external carotid artery, internal carotid artery, and occipital artery) were partially ligated with a 6–0 suture, resulting in substantial flow reduction in the RCA. After surgery, mice were allowed to recover in a recovery chamber with a heating pad underneath to maintain body temperature. Three weeks after the ligation surgery, gross plaque imaging, immunohistochemistry, plasma PCSK9 measurement, and lipid profile analysis were performed to assess whether the carotid ligation model was successfully established[14]. PCSK9 level in the plasma was measured using an ELISA kit (R&D Systems, Minneapolis, MN, USA; DPC900). Lesion area quantification was performed using ImageJ software plugin ITCN (National Institutes of Health, Bethesda, MD, USA). Initially, the entire RCA area with the formed lesion was measured, followed by the application of a color thresholding technique to isolate and quantify the specific area occupied by the plaque within the lesion. Subsequently, the ratio of plaque area to the total RCA area was calculated to determine the percentage of lesion coverage. The obtained data was graphically presented using Prism 6 software (GraphPad Software, San Diego, CA, USA).

## BLItz assays
The BLItz assay consisted of five steps: initial base line (30 s), loading (120 s, hPCSK9, 500 nM), base line (30 s), association (120 s, hCAP1, TLR4, LOX1, Syk, PKCδ), and dissociation (120 s). PCSK9-His or CAP1-mFc was immobilized on an anti-Penta-HIS, anti-mouse IgG Fc(AMC)

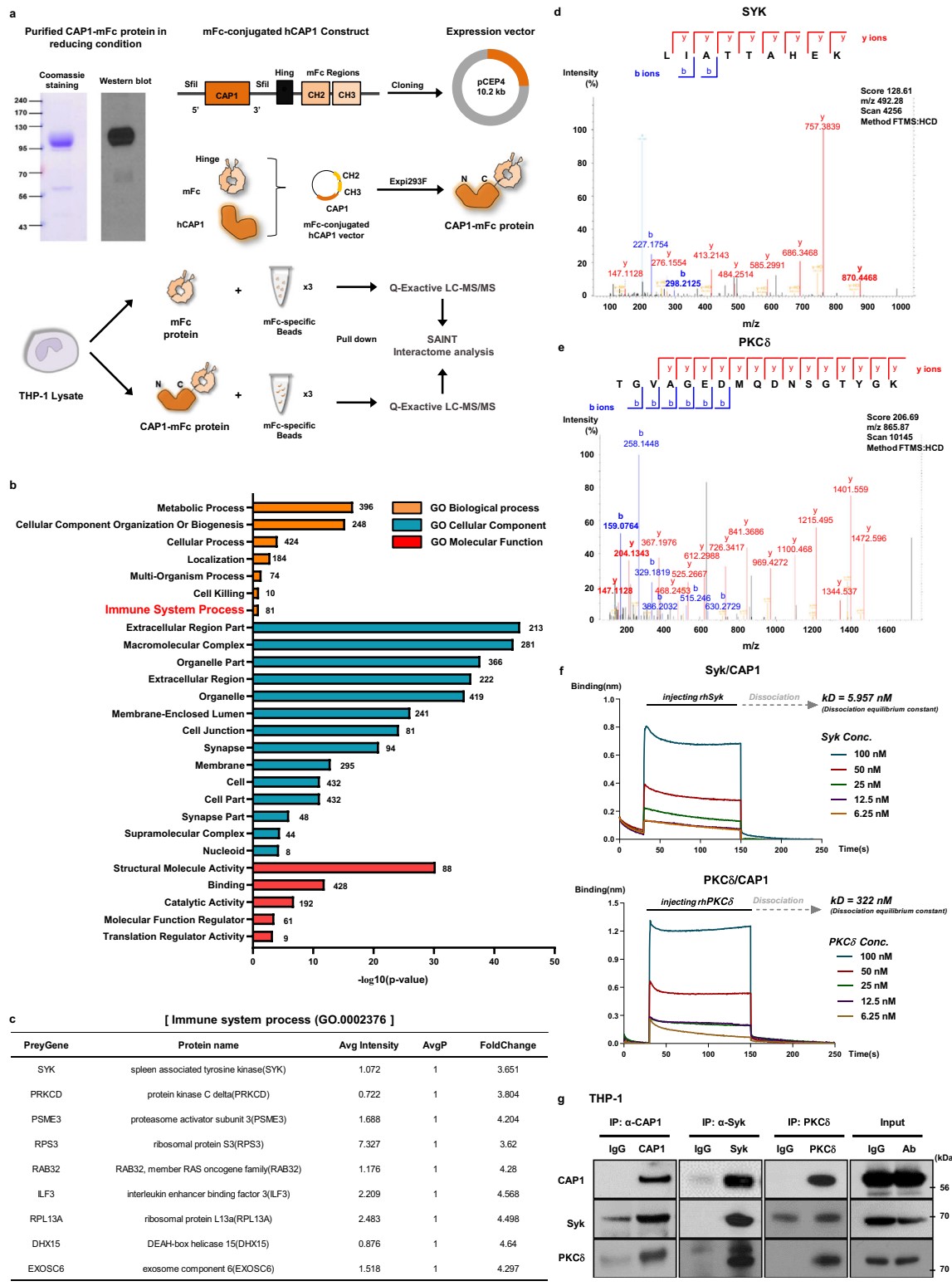

**Fig. 6 | Syk and PKCδ were identified as binding partners of CAP1.**
**a** Experimental scheme to generate an mFc-conjugated hCAP1 construct for the pull-down assay. Coomassie brilliant blue staining and immunoblotting to confirm the expression and homogeneity of hCAP1-mFc. Scheme of proteomic analysis for CAP1-binding proteins using liquid chromatography-tandem mass spectrometry (LC-MS/MS). The THP-1 lysate was incubated with either mFc or CAP1-mFc proteins and pulled down with mFc-specific beads. **b** Database for Annotation, Visualization, and Integrated Discovery (DAVID) functional enrichment analysis was performed on differentially bound proteins. The graph displays the top-ranked significantly enriched Gene Ontology (GO) terms in biological process, cellular component, and molecular function. All the adjusted statistically significant values of the terms were transformed to their negative 10-base logarithm. The analysis was conducted using an EASE score threshold of <0.1. **c** List of nine proteins that bound to CAP1 with the most significant values of SAINT Avg*P* = 1 in the immune system process (GO.0002376). MS/MS spectral data for Syk (**d**) and PKCδ (**e**). **f** The BLItz system showed that binding affinity between CAP1 with Syk or PKCδ increased in a dose-dependent manner. The equilibrium dissociation constant for Syk or PKCδ from CAP1 was 5.957 nM or 322 nM, respectively. **g** Immunoprecipitation of endogenous CAP1, Syk, and PKCδ from THP-1 cell lysate, demonstrating that these proteins bind to each other.

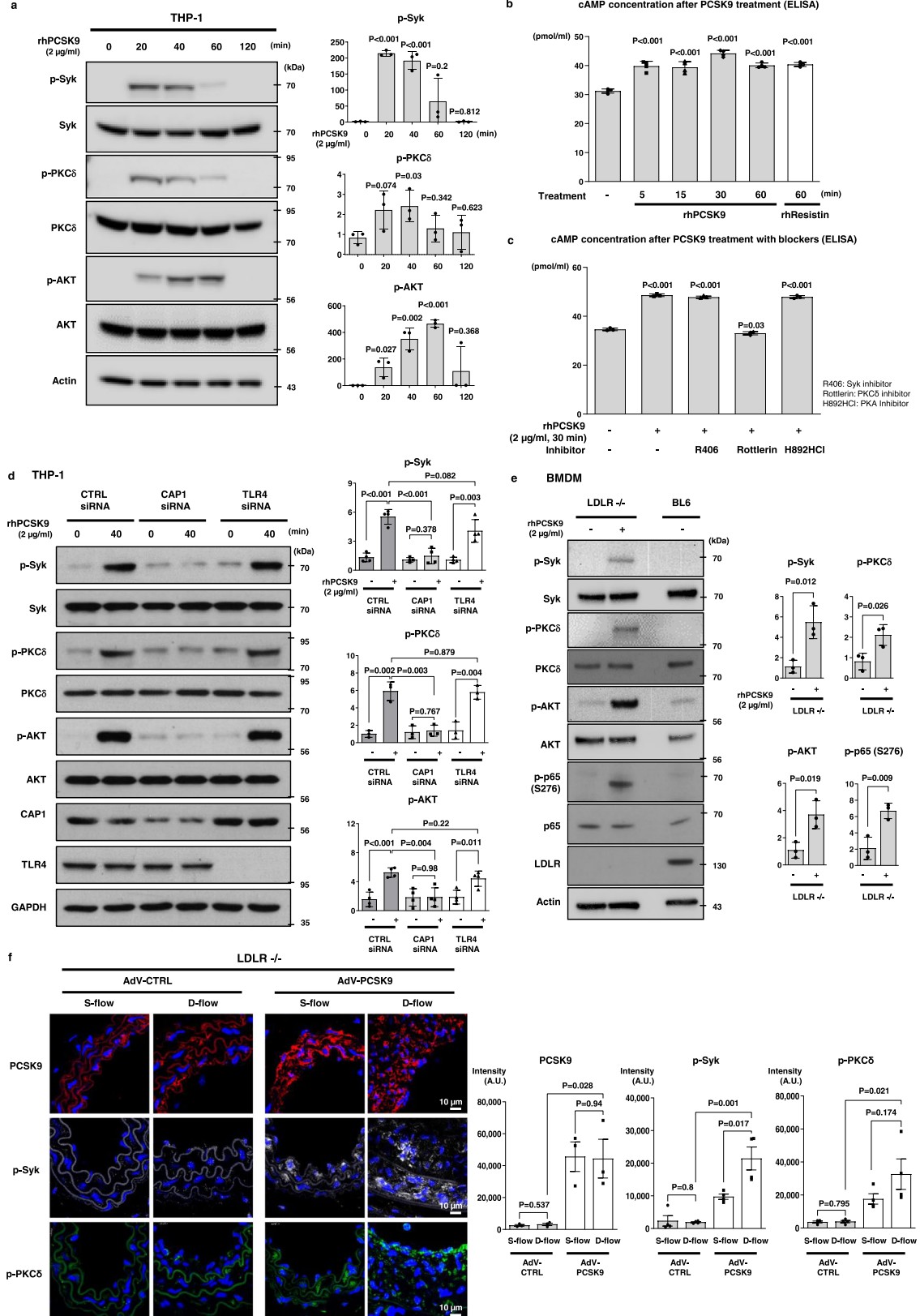

sensor. The sensorgrams were fit globally to a 1:1 binding model using the BLItz Pro software (ForteBio, Menlo Park, CA, USA), from which the dissociation equilibrium constant ($K_D$) and association ($K_a$) and dissociation ($K_d$) rate constants were calculated. The recombinant proteins used in this study were as follows: rhCAP1 (Abnova, Taipei, China; H00010487-P01), rhTLR4 (Abcam; ab159717), rhLOX1 (Sino Biological,

Beijing, China; 10585-H07H), rhSyk (Abcam; ab60886), and rhPKCδ (Abcam; ab60844).

**Proximity ligation in situ assay (PLA)**

For PLA, THP-1 cells were incubated with primary antibodies against PCSK9 (Cell Signaling Technology; #85813 S), CAP1 (Santa Cruz

**Fig. 7 | PCSK9 phosphorylated Syk and PKCδ, and their phosphorylation was dependent on CAP1, not on LDLR. a** Immunoblot analysis of signal activation of Syk, PKCδ, AKT, and NF-κB p65 after rhPCSK9 (2 μg/mL) treatment in a time-dependent manner (0, 20, 40, 60, and 120 min) in the THP-1 cell line. Syk and PKCδ were phosphorylated after 20 min of rhPCSK9 (2 μg/mL) treatment. AKT phosphorylation started at 40 min and lasted until 60 min ($N = 3$). **b, c** Multiplex ELISA of cAMP secretion to assess the involvement of Syk, PKCδ, and PKA in the PCSK9-induced inflammation pathway ($N = 4$). **b** R406, rottlerin, and H892HCl were used to inhibit Syk, PKCδ, and PKA, respectively. Rottlerin blocked PCSK9-mediated cAMP induction, whereas R405 and H892HCl did not ($N = 3$). **c, d** Immunoblot analysis demonstrating that PCSK9-induced the phosphorylation of Syk, PKCδ, and AKT in THP-1 cells. The phosphorylation was attenuated in cells transfected with CAP1 siRNA ($N = 4$ for p-Syk and p-AKT; $N = 3$ for p-PKCδ). **e** Immunoblot analysis of $Ldlr^{-/-}$ BMDMs, demonstrating that PCSK9-induced phosphorylation of Syk, PKCδ, AKT, and NF-κB p65 was independent of LDLR ($N = 3$). **f** $Ldlr^{-/-}$ mice arteries were partially ligated and treated with AdV-CTRL or AdV-PCSK9, followed by a comparison of arteries exposed to S-flow and D-flow. Immunofluorescence staining for analyzing the expression of PCSK9 (red, upper panels; $N = 3$), p-Syk (gray, middle panels; $N = 4$), and p-PKCδ (green, bottom panels; $N = 4$). The expression of PCSK9 significantly increased in the group treated with AdV-PCSK9 compared with that in the control group. Additionally, p-Syk and p-PKCδ increased more significantly under D-flow of AdV-PCSK9-treated mice than in the control group. The scale bar represents 10 μm. The differences between the groups were compared using the unpaired $t$-test (two-tailed). All experiments are independently performed. Data are presented as mean values ± SD and SEM (in (**f**) only).

Biotechnology; sc-100917), or TLR4 (Abcam; ab22048) overnight at 4 °C. Interaction between PCSK9 and CAP1 or TLR4 was visualized using oligonucleotide-conjugated secondary antibodies (Sigma-Aldrich; DUO92102-1KT). Images were acquired using a Zeiss LSM 710 Confocal Laser Scanning Microscope (Carl Zeiss AG, Jena, Germany). Interactions were quantified by counting the number of dots per cell using the ImageJ (plug ITCN) software (National Institutes of Health). Differences between means were analyzed using $t$-test with the Prism 6 software (GraphPad Software). In different figures, each bar represents the mean obtained from the quantification of signals selected randomly in six different fields.

### Cell adhesion assay
THP-1 cells were individually subjected to knockdown procedures utilizing either CTRL siRNA or CAP1 siRNA, and the cells were allowed to incubate for a period of 72 h to ensure effective knockdown. Subsequently, THP-1 cells underwent a 6-h pre-treatment with rhPCSK9. For cellular visualization, FITC labeling was achieved using a PKH67 green fluorescent cell linker kit (Sigma-Aldrich, PKH67GL). Then, a monolayer of human umbilical vein endothelial cells (HUVECs) was prepared for co-culture. FITC-labeled THP-1 cells were cocultured with the HUVECs for 1 h, both in the presence and absence of rhPCSK9. To remove non-adherent cells, two successive washes were conducted using cell culture media. Following this, we used a fluorescence microscope to capture detailed images of the THP-1 cells adhering to the HUVEC monolayer. Fluorescence-positive cells were quantified by counting using the ImageJ software (National Institutes of Health). Differences between means were analyzed using $t$-test with the Prism 6 software (GraphPad Software).

### Surface fluorescence-activated cell sorting (FACS)
THP-1 cells were incubated with recombinant human PCSK9 (500, 1000, 2000 ng/mL) for 1 h. After incubation, the cells were fixed with 1% formaldehyde and washed with cold phosphate-buffered saline (PBS). Following centrifugation, the cells were stained with anti-human CAP1 antibody (Santa Cruz Biotechnology; sc-134637) and anti-rabbit IgG Alexa Fluor 488 (Invitrogen; A21206) secondary antibodies. After staining, the labeled cells were washed with FACS buffer (PBS supplemented with 0.5% bovine serum albumin [BSA]) and analyzed using the FACSCanto II Flow Cytometry System (BD Biosciences, San Jose, CA, USA).

### FACS analysis
To assess ox-LDL binding to the cell surface, THP-1 cells with WT CAP1 overexpressed or knocked down using siRNA were pre-incubated with recombinant human PCSK9 (2 μg/mL) for 30 min. Subsequently, 10 μg/mL of DiI-ox-LDL (Thermo Fisher Scientific; L34358) was added for 1 h and the cells were then fixed with 1% formaldehyde and washed with cold PBS. Following centrifugation, the cells were washed with

FACS buffer (PBS supplemented with 0.5% BSA) and analyzed using a FACSCanto II Flow Cytometry System (BD Biosciences).

To assess VLA-4 activation, THP-1 cells with CTRL or CAP1 siRNA were pre-incubated with PBS or recombinant human PCSK9 (2 μg/mL) for 30 min. Subsequently, the cells were washed twice with cold PBS, dispersed by passing through a 40-μm strainer (Falcon Cell Strainer, Corning), and resuspended in FACS buffer. After incubation with the FITC-tagged VLA-4 antibody (Merck Millipore, Burlington, MA, USA Merck; FCMAB389F) at 4 °C for 1 h, the cells were washed with cold PBS, and FACS analysis was performed. Flow cytometry analysis and sorting (BD FACSCanto II, LSR II, and FACS Aria III,; Franklin Lakes, NJ, USABD Biosciences) were performed using several antibodies specific for VLA-4, CD11b (BD Biosciences, 553310), and F4/80 (eE-Bioscience, San Diego, CA, USA; 17-4801-82). Data were analyzed using FlowJo version 10.0.5.

### Ox-LDL uptake assay
Bone marrow cells were cultured with 30 ng/mL M-CSF (PeproTech; 300-25-100) in RPMI 1640 supplemented with 10% FBS and antibiotics-antimycotics (both from Gibco) for 3–5 days to obtain BMDMs. Ox-LDL (Thermo Fisher Scientific; L34357) was added at a final concentration of 20 μg/mL to induce ox-LDL uptake in a time-dependent manner (0, 24, 48, and 72 h). Subsequently, BMDMs were fixed with 4% formalin for 10 min at room temperature, and then stained with Oil Red O solution (Abcam; ab150678) following the manufacturer's instructions. Lipid uptake was observed under an inverted light microscope (magnification, ×100 and ×40, Nikon; ECLPSE Ci-L) and the accumulated lipid droplets in the cells were stained red.

### Sample preparation
Immunoprecipitation (IP) samples were prepared as described previously with some modifications[38]. Briefly, elution buffer (2% sodium dodecyl sulfate, 5 mM Tris (2-carboxyethyl) phosphine, and 20 mM chloroacetamide in 50 mM ammonium bicarbonate) was added to the beads. The mixture was boiled for 15 min at 95 °C to elute the interaction partners. The eluted proteins were digested using filter-aided sample preparation, as described previously[39]. Briefly, the eluate was loaded onto a 30 K Amicon filter (Millipore, MA, USA). Buffer exchanges were performed with the UA solution (8 M urea in 0.1 M Tris [pH 8.5]) via centrifugation at $14,000 \times g$ for 15 min. Following an exchange of the buffer with 40 mM ammonium bicarbonate, protein digestion was performed overnight at 37 °C using trypsin/Lys-C Mix (Promega) at a 100:1 protein-to-protease ratio. The digestion-generated peptides were collected via centrifugation. After the filter units were washed with 40 mM ammonium bicarbonate, a second digestion was performed at 37 °C for 2 h with trypsin (enzyme-to-substrate ratio (w/w) of 1:1000). All resulting peptides were acidified with 10% trifluoroacetic acid and desalted using a homemade C18 SDB-RPS-StageTip column, as described previously[39]. Briefly, peptide samples were loaded onto in-house StageTips and sequentially washed

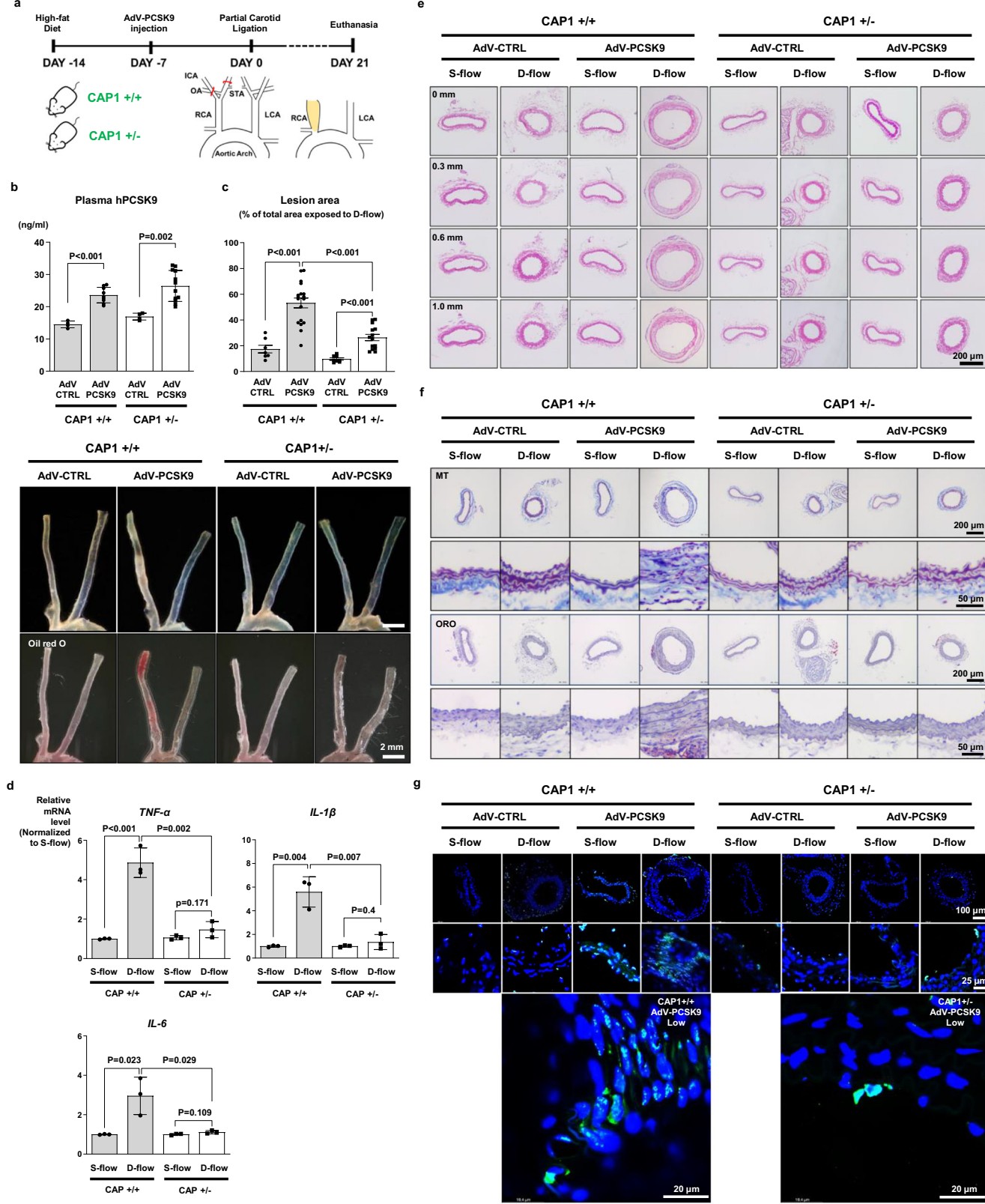

with 300 µl of 0.2% TFA. Peptides were eluted with 50 µl of 5% [v/v] ammonium hydroxide (NH4OH) in 80% [v/v] acetonitrile. The desalted samples were completely dried in a vacuum dryer and stored at −80 °C.

## LC-MS/MS analysis

LC-MS/MS analysis was performed using a Q Exactive Plus Hybrid Quadrupole-Orbitrap Mass Spectrometer (Thermo Fisher Scientific)

coupled to the Ultimate 3000 RSLC system (Thermo Fisher Scientific) via an EASY-Spray Source (Thermo Fisher Scientific), as described previously[40]. Prior to sample injection, the dried peptide samples were re-dissolved in solvent A (2% [v/v] acetonitrile and 0.1% [v/v] formic acid). The peptide samples were separated on a two-column system, consisting of a trap column (300 µm ID × 5 mm, C18, 5 µm) and an analytical column (75 µm ID × 50 cm, C18, 1.9 µm, 100 Å) with a 120 min

**Fig. 8 | CAP1 deficiency attenuated PCSK9-induced atherosclerosis in CAP1-heterozygous knockout mice. a** Experimental scheme demonstrating that PCSK9 aggravated atherosclerosis in *Cap1*[+/+] versus *Cap1*[+/−] mice. AdV-PCSK9 ($1 \times 10^{11}$ infectious units/mouse) was intravenously administered to mice on a high-fat diet. **b** Plasma hPCSK9 levels were measured using ELISA in *Cap1*[+/+] (AdV-CTRL $N = 4$; AdV-PCSK9 $N = 9$) and *Cap1*[+/−] (AdV-CTRL $N = 4$; AdV-PCSK9 $N = 12$) mice with or without PCSK9 overexpression. The plasma hPCSK9 concentration in the AdV-CTRL group was 12–14 ng/mL, whereas it was 30–50% higher in the AdV-PCSK9 group. **c** Oil Red O staining of carotid arteries after AdV-CTRL or AdV-PCSK9 injection into *Cap1*[+/+] (AdV-CTRL $N = 7$; AdV-PCSK9 $N = 18$) and *Cap1*[+/−] (AdV-CTRL $N = 8$; AdV-PCSK9 $N = 14$) mice. The atherosclerotic plaque area in the D-flow arteries increased after PCSK9 administration in *Cap1*[+/+] mice, which was prevented in *Cap1*[+/−] mice. The scale bar represents 2 mm. **d** qPCR analysis demonstrating pro-inflammatory molecules in S-flow or D-flow arteries from *Cap1*[+/+] versus *Cap1*[+/−] mice. Expression of TNF-α, IL-1β, and IL-6 was higher in D-flow than in S-flow arteries ($P < 0.001$, $P = 0.004$, and $P = 0.023$, respectively) in *Cap1*[+/+] mice with a high serum level of PCSK9. However, induction of inflammatory cytokines under D-flow was prevented in *Cap1*[+/−] mice ($P = 0.002$, 0.007, 0.029, respectively) ($N = 3$). **e** H&E staining of S-flow or D-flow arteries (from the aortic root at 0.3, 0.6, and 1 mm, respectively) from *Cap1*[+/+] versus *Cap1*[+/−] mice with AdV-CTRL or AdV-PCSK9. In the presence of a high serum level of PCSK9, significant atherosclerotic plaques developed under D-flow in *Cap1*[+/+] mice, which was prevented in *Cap1*[+/−] mice. The scale bar represents 200 μm. **f** Masson's trichrome and Oil red O staining. Each scale bar represents 200 μm, and 50 μm (magnified fields). **g** Immunofluorescence images stained with TUNEL (green). The bottom panel displays a magnified view, specifically highlighting the arteries exposed to low shear stress from the AdV-PCSK9 group. The scale bar represents 100 μm (top), 25 μm (mid) and 20 μm (enlarged). The differences between the groups were compared using the unpaired *t*-test (two-tailed). All experiments are independently performed. Data are presented as mean values ± SEM and SD (in (**b**, **d**) only).

gradient from 6% to 30% acetonitrile at 300 nl/min and were analyzed using mass spectrometry. The column temperature was maintained at 60 °C using an easy-spray column heater. Survey scans (350 to 1,650 m/z) were acquired at a resolution of 60,000 at 200 m/z. A top-15 method was used to select precursor ions with an isolation window of 2 m/z. MS/MS spectra were acquired at a higher energy C-trap dissociation with normalized collision energy of 28. The maximum ion injection time for the MS1 and the MS2 scans were 25 and 125 ms, respectively.

## Database search

MS spectra were processed using MaxQuant software version 1.6.1.0[41]. The MS/MS spectra were searched against the UniProt human protein sequence database (version 12.2014; 88,657 entries), including forward and reverse sequences and common contaminants. Primary searches were performed using a 6-ppm precursor ion tolerance for total protein level analysis. The MS/MS ion tolerance was set to 20 ppm. Cysteine I carbamidomethylation was set as a fixed modification. N-acetylation of protein and oxidation of methionine (M) were set as variable modifications. Enzyme specificity was set to full tryptic digestion. Peptides with a minimum length of six amino acids and up to two missed cleavages were considered as search parameters.

## Bioinformatics analysis

To identify proteins that interacted specifically with CAP1 and eliminate false interactions from the negative controls, SAINTexpress (https://saint-apms.sourceforge.net/Main.html) analysis was performed based on protein intensity, as described previously[42]. The probability scores of the bait and prey proteins were calculated as the average of the probabilities in individual replicates (AvgP). Proteins with AvgP ≥ 0.9 in one biological replicate or those that were detected in at least two of four biological replicates with AvgP ≥ 0.5 were likely interactors. GO enrichment was performed using DAVID version 6.8[43]. The *p* value of the enrichment of a pathway was computed using EASE score threshold of <0.1.

## Gene expression analysis

RNA was extracted from cells using TRIzol (Thermo Fisher Scientific; 15596026) according to the manufacturer's instructions. Reverse transcription-polymerase chain reaction (RT-PCR) was performed as described previously[8]. Briefly, cDNA was synthesized using a PrimeScript[t] 1st strand cDNA Synthesis Kit (Takara Bio, San Jose, CA, USA) and oligo-dT primer. Semi-quantitative PCR was performed using Maxime PCR PreMix (iNtRON Biotechnology, Gyeonggi-do, Republic of Korea) according to the manufacturer's instructions and real-time PCR was performed using Power SYBR Green PCR Master Mix (Applied Biosystems, Foster City, CA, USA) and an ABI Prism 7500 Sequence Detection System (Applied Biosystems). The relative gene expression levels were calculated using the $2^{-\Delta\Delta Ct}$ method[44] using GAPDH as a reference.

## Immunofluorescence

Cells were fixed in 4% paraformaldehyde (FUJIFILM Wako Pure Chemical Corporation, Osaka, Japan; #163-20145) at room temperature (-15–25 °C) for 15 min, washed thrice with cold PBS, permeabilized with 0.05% Triton X-100 in PBS, and blocked with 1% BSA in PBS at room temperature for 30 min. The cells were incubated overnight with primary antibodies at 4 °C. After washing, the samples were incubated for 1 h with secondary antibodies at -15–25 °C. Finally, the nuclei were stained with 4′–6-diamidino-2-phenylindole dihydrochloride (Sigma-Aldrich; D8417). Fluorescence confocal images were captured using Zeiss LSM 710 (Zeiss Microscopy, Oberkochen, Germany) and Leica TCS STED CW confocal microscope (Leica Microsystems, Wetzlar, Germany).

The tissues from mice carotid arteries were fixId in paraformaldehyde at 4 °C for 7 days and then trimmed. These tissues were embedded in Tissue-Tek O.C.T. Compound (Sakura Finetek Japan, Torrance, CA, USA) and rapidly frozen using dry ice. Frozen tissues were subsequently sectioned to a thickness of 8 μm. The sections were thawed at room temperature for 30 min. After rinsing thrice with TBST, the tissues were permeabilized with 0.05% Triton X-100 in PBS, and then blocked with 1% BSA in PBS at room temperature for 30 min. These steps were carried out in the same manner as previously described for cell immunostaining. The intensity of positive cells was quantified using images captured with a Leica STELLARIS 8 confocal microscope (Leica Microsystems). Image analysis was performed using ImageJ software (National Institutes of Health). Signals originating from autofluorescent laminar structures were excluded.

## Competitive ELISA assay

A 96-well microplate was coated with human Fc-CAP1 and was incubated with 160 μM of 6× His-PCSK9 (rhPCSK9-His, human Fc-CAP1 were kindly provided by Y-Biologics, Seoul, Republic of Korea). Additionally, either 10 or 50 μM of IgG, evolocumab, or hFc-CAP1 was added and incubated for 2 h at room temperature. Following incubation with anti-6× His secondary antibody conjugated with HRP (Invitrogen, PA1-23024), 3,3′,5,5′-tetramethylbenzidine substrate at 100 μl/well (Sigma-Aldrich; T0440) was added to generate a detectable signal using ELISA. The reaction was stopped by the addition of acidic stop solution (1 N $H_2SO_4$), and the plate was read on the microplate reader (GloMax Discover Microplate Reader, Promega) at 450 nm.

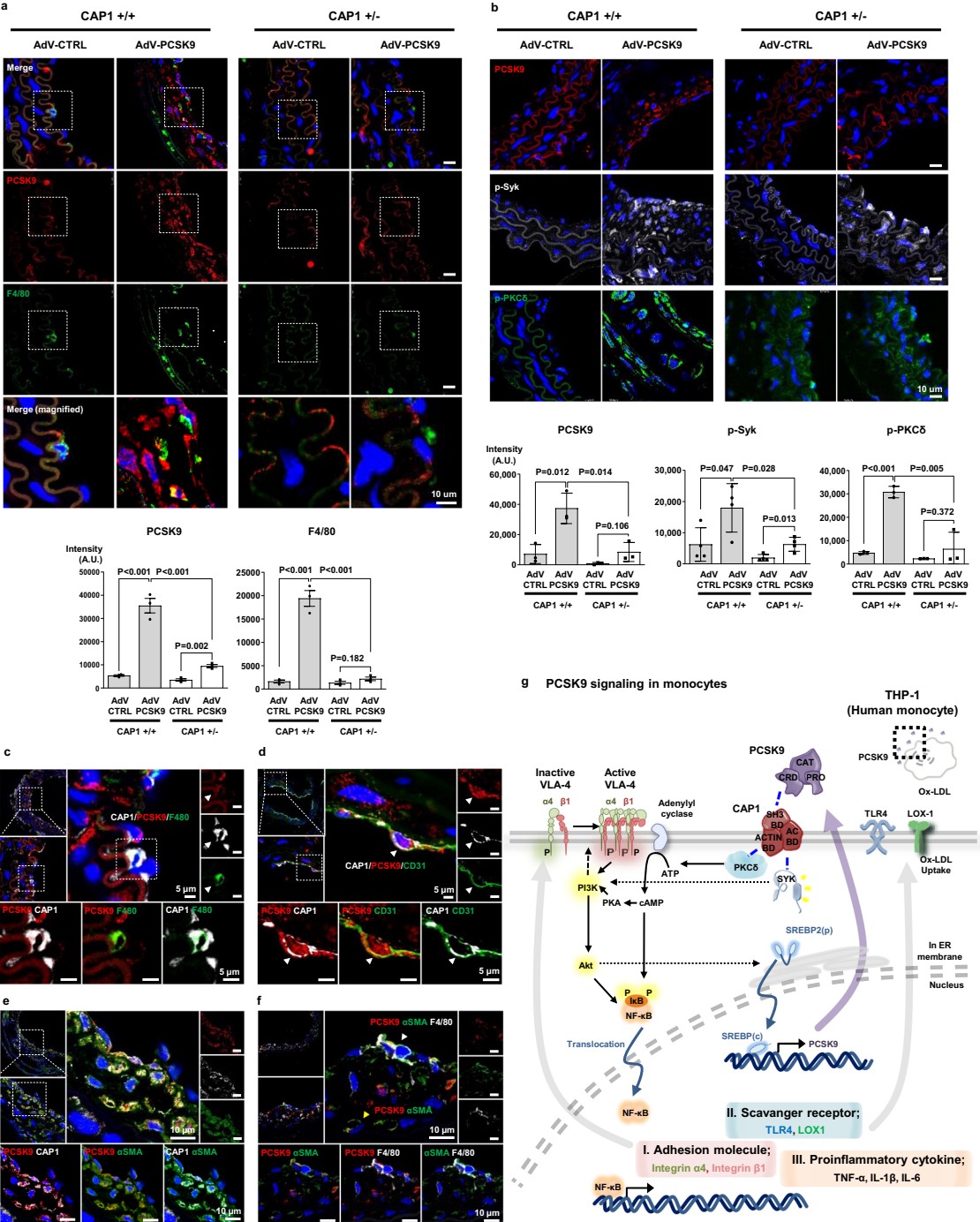

**Fig. 9 | PCSK9 colocalized with CAP1 in atherosclerotic plaques, and PCSK9-induced phosphorylation was inhibited in CAP1-heterozygous knockout mice.**
**a** Immunofluorescence staining for PCSK9 (red) and F4/80 (green) in arteries under D-flow in *Cap1*[+/+] and *Cap1*[+/−] mice injected with AdV-CTRL or AdV-PCSK9. PCSK9 and F4/8 expression significantly increased in *Cap1*[+/+] mice with AdV-PCSK9 compared with that in AdV-CTRL. However, in *Cap1*[+/−] mice, the increase in PCSK9 was less significant and F4/80 showed no significant change (*N* = 3). The scale bar represents 10 μm. **b** Partially ligated *Cap1*[+/+] and *Cap1*[+/−] mice with only arteries exposed to D-flow were compared after AdV-CTRL or AdV-PCSK9 injection. Immunofluorescence staining of PCSK9 (red, upper panels; *N* = 3), p-Syk (gray, middle panels; *N* = 4), and p-PKCδ (green, bottom panels; *N* = 3). AdV-PCSK9 increased PCSK9 expression in *Cap1*[+/+] mice, lesser than in *Cap1*[+/−] mice. Additionally, p-Syk and p-PKCδ increased more significantly in *Cap1*[+/+] mice than in *Cap1*[+/−] mice. The scale bar represents 10 μm. **c** Immunofluorescence images of arteries under D-flow from *Cap1*[+/+] mice injected with AdV-PCSK9 showing colocalization of PCSK9 (red), CAP1 (gray), and F4/80 (green). The scale bar represents 5 μm.

**d** Immunofluorescence images of arteries under D-flow from *Cap1*[+/+] mice injected with AdV-PCSK9 showing colocalization of PCSK9 (red), CAP1 (gray), and CD31 (green). The scale bar represents 5 μm. **e** Immunofluorescence images of arteries under D-flow from *Cap1*[+/+] mice injected with AdV-PCSK9 showing colocalization of PCSK9 (red), CAP1 (gray), and αSMA (green). The scale bar represents 10 μm. **f** Immunofluorescence staining for PCSK9 (red), αSMA (green), and F4/80 (gray) in arteries under D-flow from *Cap1*[+/+] mice injected with AdV-PCSK9. White arrow indicates PCSK9, αSMA, and F4/80 colocalization. The yellow arrow denotes PCSK9 and αSMA colocalization (top middle panel). The three bottom panels show the colocalization of PCSK9 and αSMA (bottom left), PCSK9 and F4/80 (bottom middle), and αSMA and F4/80 (bottom right), respectively. The scale bar represents 10 μm. **g** Schematic model showing PCSK9-mediated inflammation in monocytes mediated by CAP1 recruiting PKCδ and Syk and modulating PCSK9-mediated inflammatory signal transduction. Group differences were compared using the unpaired *t*-test (two-tailed). All data are presented as mean values ± SEM.

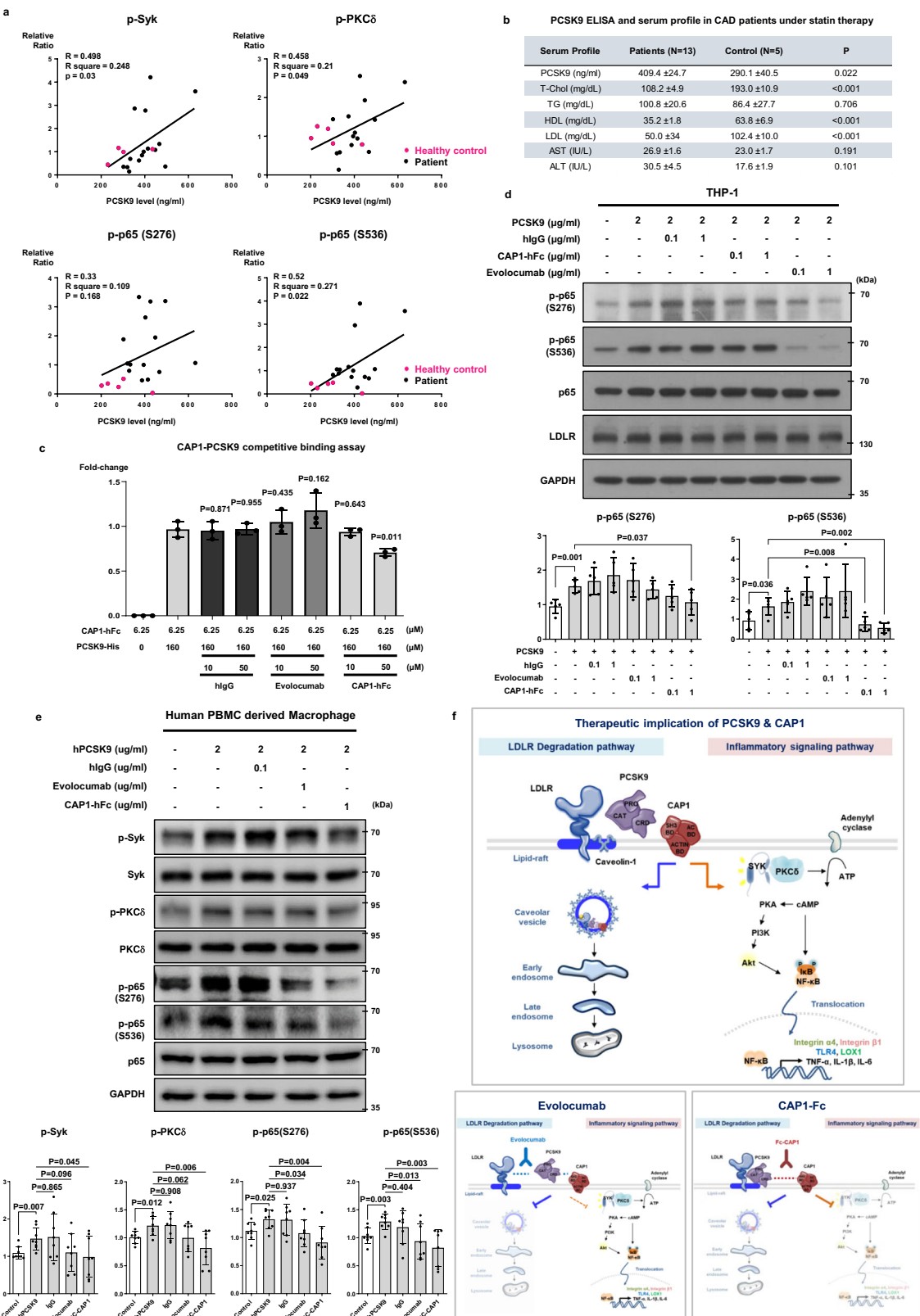

**b** PCSK9 ELISA and serum profile in CAD patients under statin therapy

| Serum Profile | Patients (N=13) | Control (N=5) | P |
|---|---|---|---|
| PCSK9 (ng/ml) | 409.4 ±24.7 | 290.1 ±40.5 | 0.022 |
| T-Chol (mg/dL) | 108.2 ±4.9 | 193.0 ±10.9 | <0.001 |
| TG (mg/dL) | 100.8 ±20.6 | 86.4 ±27.7 | 0.706 |
| HDL (mg/dL) | 35.2 ±1.8 | 63.8 ±6.9 | <0.001 |
| LDL (mg/dL) | 50.0 ±34 | 102.4 ±10.0 | <0.001 |
| AST (IU/L) | 26.9 ±1.6 | 23.0 ±1.7 | 0.191 |
| ALT (IU/L) | 30.5 ±4.5 | 17.6 ±1.9 | 0.101 |

## Ethics statement

This study was approved by the Institutional Review Board of Seoul National University Hospital (IRB no. H-2208-112-1351). This research was conducted in accordance with the Helsinki Declaration. Informed consent was obtained from all research participants, and ethical principles for the protection of personal information were strictly adhered to. Ethical guidelines for human experimentation were rigorously followed throughout the study. Anonymized and de-identified information was used for analyses.

## Isolation of human PBMCs and preparation of human PBMC-derived macrophages

Human PBMCs were isolated from blood samples obtained from CAD patients who were under appropriate medications including statin as

**Fig. 10 | PCSK9 levels in the serum of patients with coronary artery disease (CAD) correlated with Syk, PKC, and NF-κB phosphorylation, and PCSK9-mediated phosphorylation was blocked by CAP1-hFc. a** Simple linear regression analysis of Pearson correlation by distance. This figure illustrates the relationship between serum PCSK9 concentration and the quantified levels of phosphorylated proteins in a single individual. Serum PCSK9 concentration displayed a positive correlation with the phosphorylation of Syk, PKCδ, p65(S276), and p65(S536) in human peripheral blood mononuclear cells (PBMCs). The black line represents an asymptotic regression line fitted to the raw data ($N = 19$). **b** Average serum PCSK9 concentration and lipid profile results from 13 hyperlipidemia patients and five healthy donors. T-Chol total cholesterol, TG triglyceride, HDL high-density lipoprotein, LDL low-density lipoprotein, AST aspartate aminotransferase, ALT alanine aminotransferase. **c** Competitive ELISA binding assay showed that PCSK9-His bound to CAP1, and its interaction was competitively inhibited by CAP1-hFc, whereas evolocumab did not hinder the PCSK9-CAP1 interaction ($N = 3$). **d** Immunoblot analysis of NF-κB p65 signal activation after treatment with rhPCSK9 (2 μg/

mL) along with human IgG, evolocumab, or CAP1-hFc in THP-1 cells revealed that the phosphorylation of p65 induced by PCSK9 was notably reduced by CAP1-hFc ($N = 5$). **e** Immunoblot analysis demonstrated that PCSK9-induced phosphorylation of Syk, PKCδ, and NF-κB p65 in human PBMC-derived macrophages was attenuated by CAP1-hFc. After 1-h treatment of rhPCSK9 with human IgG, evolocumab, or CAP1-hFc, the PCSK9-induced phosphorylation of Syk, PKCδ, and NF-κB p65 was significantly decreased by CAP1-hFc ($N = 8$). **f** Schematic diagram depicting CAP1 as the binding partner of PCSK9, which mediates not only caveolae-dependent endocytosis and lysosomal degradation of LDLR, but also recruits Syk and PKCδ and modulates PCSK9-mediated inflammatory signal transduction. CAP1-hFc inhibits the binding of CAP1 and PCSK9, which sequentially block LDLR degradation and the inflammatory signaling pathway, whereas the PCSK9 inhibitor evolocumab can only block the LDLR degradation pathway. The differences between the groups were compared using the unpaired t-test (two-tailed). All experiments are independently performed and all data are presented as mean values ± SD.

---

well as from healthy volunteers under no medications using Ficoll-Paque gradient separation (Cytiva, Marlborough, MA, USA; 17-1440-02). Briefly, blood samples were mixed with PBS in a 1:2 ratio and gently inverted. Ficoll-Paque media were carefully loaded at the bottom of the blood-PBS mixture. After centrifugation at $2400 \times g$ at room temperature for 30 min, a thin band containing PBMCs formed. This PBMC-rich band was carefully collected and subjected to two washes with PBS. Isolated human PBMCs cultured with RPMI high-glucose medium supplemented with 10% FBS and 1X antibiotics-antimycotics (both from Gibco). To differentiate human PBMCs into human PBMC-derived macrophages, 30 ng/mL human M-CSF was added, and the cells were cultured for 1 week.

### Correlation between PCSK9 serum levels and Syk, PKC, and NF-κB phosphorylation

Blood samples were obtained from individuals diagnosed with CAD. Serum was separated from 1 mL of each patient's blood sample, and the concentration of human PCSK9 was measured using the Quantikine Human PCSK9 ELISA Kit (R&D Systems, Minneapolis, MN, USA; DPC900). Simultaneously, human PBMCs were isolated, and the relative ratio of phosphorylated proteins was determined through western blotting.

### Statistical analysis

Spearman's rank correlation coefficients were applied to determine the statistical significance of the relationships between serum PCSK9 levels and alterations in the relative ratios of phosphorylated protein expression levels. Prism 8 software (GraphPad Software) was employed for data analysis. The correlation coefficient (r) was used to quantify the strength and direction of the linear relationship between the variables. Additionally, the coefficient of determination ($r^2$) was derived from the square of the correlation coefficient, indicating the proportion of the variance in one variable that could be explained by the other variable in a linear relationship. All statistical tests were two-tailed Student's t-test, and p-values less than 0.05 were considered statistically significant. All experiments were independently performed at least three times except for the LC-MS/MS analysis in Fig. 6. Values are expressed as mean ± standard error of mean (SEM) or mean ± standard deviation (SD).

### Reporting summary

Further information on research design is available in the Nature Portfolio Reporting Summary linked to this article.

## Data availability

The mass spectrometry raw data generated in this study have been deposited in the PRIDE[45] database under the accession code

PXD047058. All other data supporting the findings of this study are available within the paper and its Supplementary Information. Source data are provided with this paper.

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

## Acknowledgements

We thank Hye Young Park for their technical assistance and for helpful discussions, and Jay Horton (UT Southwestern) for the AdV-PCSK9 construct. Recombinant human PCSK9-His, human Fc-CAP1 were kindly supported by Y-Biologics. This work was supported by grants from the Korea Health Industry Development Institute (KHIDI) (HI14C1277 to H.S.K.) funded by the Korea Government (MHW). The research was also supported by a grant from the National Research Foundation of Korea (NRF) funded by the Korea Government (MSIT) (NRF-2021R1A2C2094323 to H.D.J. and RS-2023-00228390 to H.S.K.) and by a grant from the Korea Drug Development Fund funded by the Ministry of Science and ICT, Ministry of Trade, Industry, and Energy, and Ministry of Health and Welfare (HN21C0524, Republic of Korea to H.D.J.).

## Author contributions

H.D.J., D.S. and S.K. conducted experiments and analyzed and interpreted the data. H.L., H.C.L., H.P., M.F., E.C., S.Choi., B.J.K., J.H.Y., G.N. and S.Cho. performed and helped in the experiments and analyzed the data. J.L. assisted with animal experiments. C.W.K. provided advice on animal models of carotid artery ligation. D.H. performed LC-MS/MS and assisted with bioinformatics analysis. H.S.K. and H.D.J. supervised research and edited the paper. All authors contributed to the manuscript and approved the submitted version of the manuscript.

## Competing interests

The authors declare no competing interests.
