## [Peer Review File · Nature Communications]

PCSK9 stimulates Syk, PKC δ , and NF- κ B, leading to atherosclerosis progression independently of LDL receptorEditorial Note: Parts of this Peer Review File have been redacted as indicated to remove third-party material where no permission to publish could be obtained.

REVIEWER COMMENTS

Reviewer #1 (Remarks to the Author):

The author identified a novel mechanism how PCSK9 induced inflammation and atherosclerosis directly or independently of LDLR. Moreover, these functions of PCSK9 were dependent on CAP1 and thus attenuated by CAP1 depletion. They further verified that PCSK9-CAP1 interaction induced inflammation via SYK and PKC δ signal pathway. This study has provided some interesting and novel findings. However, there were some defects and deficiencies in this study.

1. Statistical charts should be provided for all western blots. Some bands of western blots were unconvincing. For example, TLR4 expression in THP-1 should be similar in all CTRL siRNA treatment groups, but the results presented in this article were significantly different in Fig3e and Fig7d.

2. The author indicated that PCSK9 promoted the pro-inflammatory genes in THP-1, HepG2 and BMDM. Could PCSK9 activate the pro-inflammatory genes in HUVEC?

3. In Fig 3b, co-localization of PCSK9 and CAP1 were observed. But the author should show non-treatment and all PCSK9 treatment groups at the same time.

4. CAP1 expression should be inhibited significantly by CAP1 siRNA treatment, but the results presented in In Fig3f show that CAP1 expression level was still very high in CAP siRNA group, while in fig 4a, CAP1 expression can be inhibited completely by CAP1 SiRNA. These results indicated that CAP1 RNA interference system using by authors was very unstable.

5. Experimental groups were inconsistent especially in vivo. The author presented low shear stress and normal shear stress groups in Fig 8e. Such groups should also be shown in Figure

2c. Moreover, the author should investigate and discuss about the role of PCSK9 in different shear stress which is related to the progression of atherosclerosis.

6. HE staining should also be performed by cross sectional analysis in Fig 8e similar to Fig 2c.

7. All RNA interference experiments should have interference efficiency measurements. The author should synthesize at least three RNA interference chains for each target gene and detect interference efficiency of each RNA interference chain. The best one can be chosen for further research.

8. The author should discuss the relevant research progress and the significance of this study in the discussion section instead of repeating all the experimental results.

9. The background of heterozygous Cap1 knockout mice should be described in detail. Moreover, the author needs to compare Cap1 gene and CAP1 protein expression of heterozygous Cap1 knockout mice to Wild type mice. Furthermore, homozygous Cap1 knockout mice must be better than heterozygous mice.

Reviewer #2 (Remarks to the Author):

This study gives a completely new mechanistic insight by which PCSK9 contributes to atherosclerosis, beyond LDLR, by triggering inflammation upon binding to CAP1, a human resistin receptor. The authors gave a thorough mechanistic insight into this interaction, identifying two novel proteins in the signalling cascade of PCSK9/CAP1 complex, SYK and PKC δ .

NF- κ B and inflammatory genes were induced in monocytes and induced atherosclerosis in Ldlr $^{-/-}$ mice, as well as rhPCSK9 induced phosphorylation of SYK and PKC δ , which confirmed PCSK9-induced inflammation was triggered by its binding to CAP1, followed by downstream activation of SYK and PKC δ , and that this mechanism was independent of LDLR. The authors show that CAP1 is the major receptor of PCSK9.

This paper could have a major impact on future targeted therapy development for atherosclerosis and inflammation-related diseases.

The methodology is sound, and provided in enough detail in the methods for the work to be

reproduced.

Minor comments:

All P values should be stated as upper cases.

Line 119: P value should be at 3 decimal points

Line 140 "Resistin, pro-inflammatory cytokines (TNF- α , IL-1 β , IL-6), LPS and PCSK9 itself (p=0.005, p=0.0003, respectively) increased PCSK9 expression (Fig. 5d)."

It is not clear which P values are stated, there are only two, and more than two parameters in the sentence.

It would also be of notion to state in the discussion part if there are any clinical studies so far which analysed CAP1 in the context of atherosclerosis-related pathologies.

Abstract section:

Lines 25/26: „we treated PCSK9 in the monocyte and endothelial cells“

The sentence is not clear, should be more appropriately described.

Abbreviations lacking for AAV-PCSK9 a CAP1, TLR4

Reviewer #3 (Remarks to the Author):

This interesting study was carried out to establish the role of PCSK9 in monocyte activation and atherosclerosis via LDLR independent mechanism. The topic is timely and has very high clinical impact. The study is a logical extension of the same group's previous study that established that PCSK9-CAP1 binding is a key mediator of caveolae-dependent endocytosis and lysosomal degradation of LDLR (Jang et al, Eur Heart J 2020). Overall, the present manuscript is of potential interest and the experiments seem to have been carefully executed. Some approaches are sophisticated. This reviewer, however, found various concerns and suggested below solutions to further strengthen the manuscript.

[General concerns]

1. The authors did not address mechanisms in depth, leaving the manuscript somewhat superficial. This is a major weakness. For instance, in in vivo experiments, the authors did not examine possible mechanisms by which AAV-PCSK9 and CAP1 deficiency affected lesion

size of mouse carotid arteries.

2. The numbers of animals used in the experiment are small (Figure 2b, c and Figure 8). This raised a concern related to scientific rigor.

3. The use of THP-1 cells for most in vitro experiments is also a major concern. Cancer cell lines are very different from primary cells.

4. It is widely known that undetectable traces of endotoxin on a recombinant protein, PCSK9 in the present manuscript, can cause cell activation as evidenced in many previous studies. This possibility needs to be ruled out via more thorough examinations.

5. The authors examined the atherosclerotic lesion area and oil red O positive area of carotid arteries. This is insufficient. Other assays/parameters (e.g., cellular components, cell growth, apoptosis, necrotic core size, etc) would help to not only support the authors' theory, but also understand what PCSK9 really did to affect lesion development in this model.

6. Which cell types are a main source of PCSK9 in the atherosclerotic lesion of the carotid arteries in this model?

7. The authors decided that "immune system process" is the one to go after in the DAVID functional enriched analysis. The immune system process, however, is at the lowest ranking in GO Biological process (Supplementary Fig.3). Please provide a strong rationale behind this choice.

8. There are some disconnections between in vitro and in vivo data. This is also a major concern. How did AAV-PCSK9 affect SYK and PKC δ in vivo?

9. More data need to be quantified. For example, the authors showed that CAP1-deficiency decreased macrophage accumulation in the carotid arteries only qualitatively.

10. Are CAP1, SYK and PKC δ regulated in macrophages isolated from patients before and

after treatment with PCSK9 inhibitors?

11. CAP1 protein is ubiquitously expressed in several cell types including ECs, SMCs and immune cells in vascular tissues. The authors need to demonstrate how much effect monocyte-derived CAP1 exerts on the atherosclerotic plaque in vivo.

[More specific concerns]

12. Figure 1A and 4A: Total p65 was not quantified.

13. Figure 2: This reviewer wonders why authors chose a short timepoint (2 weeks) of HFD before carotid ligation.

14. Figure 2: It would be interesting to see the changes in the expression of adhesion and inflammatory markers in liver tissue, carotid artery and BMDMs of AAV-PCSK9 injected LDLR^{-/-} mice to have consistency in findings from in vitro experiments (Figure 1).

15. Figure 2E: Phosphorylation is a transient process and stimulation of 10-30 min is standard for measuring phosphorylation of signalling proteins. This reviewer wonders why authors chose 24h rhPCSK9 stimulation for measuring phosphorylation and whether time-course experiments were conducted. In addition, the quantification data is missing for this experiment.

16. Figure 3F: From immunoblot, it seems CAP1 siRNA was not efficient. The authors may want to present the mRNA levels of CAP1 after siRNA.

17. HUVEC-THP-1 adhesion assay (Figure 4F): Details regarding the assay and analysis is not clear.

18. Figure 5: The experimental design raised a concern. The authors used siRNA of CAP1 in THP-1 cells where LDLR was present.

19. Figure 6: The authors conducted elegant experiments to show the downstream molecules involved after PCSK9-CAP1 binding by pulldown followed by MS/MS. However, better explanations of selection of SYK/PKC δ (out of 2103 binding proteins to CAP1) as the signalling pathway need to be provided.
20. Pulldown and MS/MS assay are particularly interesting. Did the authors reiterates that the pulldown of 2103 proteins are the result of PCSK9-CAP1 interaction or these are just the binding partners of CAP1 in its native form?
21. Figure 8F: The overall mechanism presented in the schematic is too complex with several levels of signalling mechanism involved and little experimental evidence provided in the study.
22. In qPCR data throughout the manuscript, a control group/condition have no error bar. This reviewer does not believe one can use all the values in control samples as 1 in statistics.
23. Molecular weight markers are missing in immunoblots throughout the manuscript.
24. Abstract did not summarize the overall study well.
25. Introduction: Lines 45-48 need citations.
26. Statistical details are missing in the figure legends.

Comments from Reviewer #1 and Responses

The author identified a novel mechanism how PCSK9 induced inflammation and atherosclerosis directly or independently of LDLR. Moreover, these functions of PCSK9 were dependent on CAP1 and thus attenuated by CAP1 depletion. They further verified that PCSK9-CAP1 interaction induced inflammation via SYK and PKC δ signal pathway. This study has provided some interesting and novel findings. However, there were some defects and deficiencies in this study.

[Response]: We thank the reviewer for their insightful comments and suggestions. A point-by-point response to all comments has been provided below.

[Comment #1-1]: Statistical charts should be provided for all western blots. Some bands of western blots were unconvincing. For example, TLR4 expression in THP-1 should be similar in all CTRL siRNA treatment groups, but the results presented in this article were significantly different in Fig3e and Fig7d.

[Response #1-1]: We have described the number of biological replicates and the number of repeated studies. As western blotting is a semi-quantitative method, we tried our best to show quantitative data using a scatter plot with mean and standard error wherever possible. As the reviewer mentioned, TLR4 expression in THP-1 cells was not similar to that with CTRL siRNA. Therefore, we repeated the same experiments and replaced western blots to show a similar TLR4 expression band.

Changes in the Manuscript *Figure 3e and Figure 7d*

Fig. 3e Original data

Fig. 3e Revised data

Fig. 3e. Immunoblot analysis demonstrating knockdown of TLR4, LOX1, and CAP1 after 72 h of siRNA treatment (*upper left*: original data, *upper right*: updated data, *bottom*: quantification of updated western blotting data).

Fig. 7d Original data

Fig. 7d Revised data

Fig. 7d. Immunoblot analysis demonstrating knockdown of TLR4, LOX1, and CAP1 after 72 h of siRNA treatment. Subsequently, monocytes were treated with rhPCSK9, resulting in a significant attenuation of phosphorylation of SYK and PKC δ only in cells with CAP1 knockdown (*left*: original data, *right*: updated data).

[Comment #1-2]: The author indicated that PCSK9 promoted the pro-inflammatory genes in THP-1, HepG2 and BMDM. Could PCSK9 activate the pro-inflammatory genes in HUVEC?

[Response #1-2]: We investigated whether PCSK9 could activate pro-inflammatory genes in human umbilical vein endothelial cells (HUVECs). Similar to our findings in THP-1 cells, HepG2 cells, and bone marrow-derived macrophages (BMDMs), we treated HUVECs with rhPCSK9 at different concentrations: 0, 50, 200, and 2000 ng/ml for a duration of 12 h. We then examined the mRNA levels of pro-inflammatory cytokines such as TNF- α , interleukin (IL)-1 β , IL-6, and adhesion molecules, including ICAM-1, VCAM-1, E-selectin (SELE) using Real-Time qPCR. PCSK9 treatment significantly increased the mRNA levels of pro-inflammatory cytokines, specifically TNF- α , IL-1 β , and IL-6 in a dose-dependent manner (P<0.001, P=0.017, P<0.001 for TNF- α ; P=0.100, P=0.022, P=0.003 for IL-1 β ; P<0.001, P<0.001, P=0.002 for IL-6, at 50, 200, and 2000 ng/ml, respectively). Additionally, PCSK9 treatment induced a significant increase in the mRNA levels of adhesion molecules, such as ICAM-1, VCAM-1, and SELE (P<0.001, P=0.019, P<0.001 for ICAM-1; P<0.001, P=0.001, P=0.023 for VCAM-1; P<0.004, P<0.001, P=0.011 for SELE, at 50, 200, and 2000 ng/ml, respectively). These findings are presented *in Figure 1e of our manuscript.*

Changes in the Manuscript Page 4, line 86, Figure 1e

We have added the following text:

PCSK9 treatment in human ECs led to an increase in the mRNA levels of pro-inflammatory cytokines such as TNF- α , IL-1 β , IL-6 and induction of adhesion molecules, including VCAM1, ICAM1, and SELE (Fig. 1e).

Fig. 1e. qPCR analysis demonstrated that PCSK9 significantly increased the expression of pro-inflammatory cytokines, including TNF- α , IL-1 β , and IL-6, and the mRNA level of adhesion molecules (*ICAM-1*, *VCAM-1*, E-selectin [*SELE*]) in endothelial cells (N \geq 5).

[Comment #1-3]: In Fig 3b, co-localization of PCSK9 and CAP1 was observed. But the author should show non-treatment and all PCSK9 treatment groups at the same time.

[Response #1-3]: In monocytes, PCSK9 colocalized with CAP1 in the cell membrane, and the colocalization of PCSK9 and CAP1 further increased 60 min after PCSK9 treatment. We revised the figures and the text according to the reviewer's comment in Fig. 3b.

Changes in the Manuscript Page 6, line 128, Figure 3b

Through immunofluorescence, colocalization of PCSK9 and CAP1 was observed mainly in the cell membrane, and this colocalization further increased 60 min after rhPCSK9 treatment in THP-1 cells (Fig. 3b).

Fig. 3b. Immunofluorescent staining of THP-1 cells with CAP1 (green) and PCSK9 (red) demonstrating their colocalization (left panel) mainly to the membrane surface of monocytes (yellow). The colocalization between PCSK9 and CAP1 further increased 60 min after treatment with 2 µg/ml rhPCSK9 (right panel). Colocalization analysis within the membrane was performed using orthogonal views from different planes of confocal microscope images. Scale bar represents 5 µm.

[Comment #1-4]: CAP1 expression should be inhibited significantly by CAP1 siRNA treatment, but the results presented in Fig. 3f show that CAP1 expression level was still very high in CAP siRNA group, while in Fig. 4a, CAP1 expression can be inhibited completely by CAP1 SiRNA. These results indicated that CAP1 RNA interference system used by the authors was very unstable.

[Response #1-4]: Previously, we established an efficient CAP1 siRNA interference system and reported CAP1 as a receptor for resistin in humans (Lee, S. et al., *Cell Metab*, 2014, 19(3), 484–497) and as a binding partner of PCSK9 in LDLR degradation (Jang, H.D. et al., *Eur Heart J*, 2020, 41(2), 239–252) using THP-1 and HepG2 cells, respectively. In these studies, we used siRNA targeting CAP1 with the sequence of 5'-AAACCGAGTCCTCAAAGAGTA-3', which is identical to that used in our previous reports (Reviewer only Fig. 1-1). For the target knockdown, siRNA oligos were transfected using RNAiMax Lipofectamine (Invitrogen Life Technologies) according to the manufacturer's instructions.

[FIGURE REDACTED]

Reviewer only Fig. 1-1. Establishment of an efficient CAP1 siRNA interference system. Lee, S. et al., *Cell Metab*, 2014, 19(3), 484–497 (*left panel*) and Jang, H.D. et al., *European Heart J*, 2020, 41(2), 239–252 (*right panel*).

To maximize the knockdown efficiency of siRNA in THP-1 cells, we re-ordered CAP1 siRNA with the same sequence as Lee et al. and Jang et al., extended THP-1 cell starvation to overnight, and evaluated the extent of CAP1 protein knockdown in a time-dependent manner. We treated overnight starved THP-1 cells in basal RPMI with CAP1 siRNA and Lipofectamine RNAiMAX for 6 h and changed the media from starvation (0% FBS) to growth RPMI (10% FBS). After 24, 48, and 72 h of siRNA treatment, cells were harvested with RIPA buffer and western blotting was performed to compare CAP1 protein expression. We found that the optimum time required for CAP1 to be knocked down sufficiently was 72 h and **we provided this information as supplementary Figure 4.**

Supplementary Fig. 4. Immunoblot analysis demonstrating the effects of time elapsed after siRNA transfection on the degree of CAP1 or TLR4 knockdown (top). THP-1 cells were transfected with CAP1 or TLR4 siRNA and harvested at different time points: 0, 24, 48, and 72 hr. The representative figures of three independent experiments are shown and quantified using ImageJ software (bottom).

Therefore, we carried out all CAP1 and TLR4 knockout experiments in THP-1 cells using this protocol. As a result, CAP1 protein expression was significantly attenuated as demonstrated in Fig. 4a, b, e, and g.

Changes in the Manuscript Figure 4a, 4b, 4e and 4g

Fig. 4

Fig. 4a, 4b, 4e, and 4g. Immunoblot demonstrating that siRNA targeting CAP1 attenuated CAP1 protein expression in THP-1 and HUVEC cells.

[Comment #1-5]: Experimental groups were inconsistent, especially in vivo. The author presented low shear stress and normal shear stress groups in Fig 8e. Such groups should also be shown in Figure 2c. Moreover, the author should investigate and discuss the role of PCSK9 in different shear stress which is related to the progression of atherosclerosis.

[Response #1-5]: In response to your suggestions, we have aligned Figure 2c with Figure 8e to include both low and normal shear stress groups, ensuring a coherent presentation of our experimental design. To address this concern, additional mouse experiments were conducted, enhancing the consistency of our study.

In line with your recommendations, we have also enhanced the uniformity of our presentation by renaming the 'low shear stress (in right carotid artery [RCA])' and 'normal shear stress (in left carotid artery [LCA])' groups as 'disturbed flow (D-flow)' and 'stable steady laminar flow (S-flow)' respectively, improving clarity across the manuscript. To comprehensively explore arterial dynamics, we obtained sections at intervals of 0.3, 0.6, and 1 mm from the artery root, employing hematoxylin and eosin staining for detailed structural assessment. Notably, the AdV-PCSK9 injection group demonstrated a substantial increase in plaque volume and arterial thickness compared with the control AdV-CTRL group. Importantly, these effects were most pronounced within arteries exposed to disturbed flow (D-flow).

Changes in the Manuscript Page 5, line 104, Figure 2c

Furthermore, in cross-sectional analysis, the AdV-PCSK9 injection group showed greater plaque volume and arterial thickness than the control AdV-CTRL group (Fig. 2c).

Fig. 2c. Carotid artery cross-section staining shows atherosclerotic plaque development in partial ligation-induced atherosclerosis in *Ldlr*^{-/-} mice. Enlarged atherosclerotic plaques were observed in the arteries of AdV-PCSK9-treated mice under D-flow compared with those of AdV-CTRL, indicating the significant impact of PCSK9 on atherosclerosis. Hematoxylin and eosin staining of serial sections from the aortic root at 0.3, 0.6, and 1 mm. The scale bar represents 200 μ m.

The overexpression of PCSK9 (AdV-PCSK9) rapidly induced hyperlipidemia and led to the development of atherosclerosis within three months in C57BL/6 mice, which is significantly faster than other conventional methods. Partial carotid ligation induced disturbed flow and low shear stress in the artery, resulting in increased reactive oxygen species (ROS) and inflammation. Recently, Kumar, S. et al. (*Lab Invest*, 2017, 97(8), 935–945) combined these two approaches to develop accelerated atherosclerosis in C57 mice, which we used in our study. In a related study, Ding et al. (*Antioxid Redox Signal*, 2015, 22(9), 760–771) demonstrated that the low shear stress induced PCSK9 expression and ROS generation in vascular endothelial cells (ECs) and smooth muscle cells (SMCs). The interaction between PCSK9 and ROS may play a crucial role in the progression of atherosclerosis within arterial channels with low shear stress.

In our study, we aimed to shed light on the role of PCSK9 in monocyte-driven inflammatory gene expression and its potential interaction with CAP1. Utilizing a mouse model of atherosclerosis through partial carotid ligation surgery, we investigated PCSK9 and CAP1 expression across various cell types within the carotid artery, including SMCs, ECs, and infiltrated macrophages. Our findings directly linked PCSK9 to inflammatory gene expression in monocytes. Following AdV-PCSK9 treatment, PCSK9 expression prominently increased in arteries subjected to either steady (S-flow) or disturbed (D-flow) condition, as illustrated in Figure 7f.

Changes in the Manuscript Page 9, line 219, Figure 7f

In the $Ldlr^{-/-}$ mouse arteries, PCSK9 expression was greater in mice that received AdV-PCSK9 than in those that received AdV-CTRL ($P=0.001$). Elevated PCSK9 levels in the arterial tissue were associated with a significant increase in the phosphorylation of Syk and PKC δ in neointimal tissue, and this effect was further enhanced by partial ligation ($P=0.001$, $P=0.002$, respectively; Fig. 7f).

Fig. 7f. *Ldlr*^{-/-} mice arteries were partially ligated and treated with AdV-CTRL or AdV-PCSK9, followed by a comparison of arteries exposed to S-flow and D-flow. Immunofluorescence staining for analyzing the expression of PCSK9 (red, upper panels), p-Syk (gray, middle panels), and p-PKC δ (green, bottom panels). The expression of PCSK9 significantly increased in the group treated with AdV-PCSK9 compared with that in the control group ($N \geq 3$). Additionally, p-Syk and p-PKC δ increased more significantly under D-flow of AdV-PCSK9-treated mice than in the control group. The scale bar represents 10 μ m. The differences between the groups were compared using the unpaired *t*-test.

Additionally, we observed co-expression of PCSK9 and CAP1 within atherosclerotic plaques, evident in macrophages (Fig. 9c), ECs (Fig. 9d), and SMCs (Fig. 9e). Notably, SMCs within the atherosclerotic milieu exhibited characteristics akin to immune-like cells, consistent with previous research (Rosenfeld, M.E. et al., *Nat Med*, 2015, 21(6), 549–551.; Gomez, D. et al., *Cardiovasc Res*, 2012, 95(2), 156–164). PCSK9 expression was identified in both SMA⁺F4/80⁺ and SMA⁺F4/80⁻ cell populations, emphasizing its relevance across these cell types (Fig. 9f). In conclusion, these observations strongly indicate that both CAP1 and PCSK9

play pivotal roles in orchestrating vascular inflammation, fostering macrophage infiltration, and driving atherosclerosis progression. These findings significantly enhance our understanding of the intricate interplay between PCSK9 and CAP1 and their combined influence on atherosclerotic pathogenesis.

Changes in the Manuscript Page 10, line 253, 9c, 9d, 9e, and 9f

Next, to elucidate the cellular localization of CAP1 and PCSK9 within the atherosclerotic plaque, we conducted immunofluorescence staining of infiltrated macrophages, ECs, and SMCs, which constitute atherosclerotic plaques. We observed that PCSK9 and CAP1 were expressed in atherosclerotic plaques and confirmed their colocalization on macrophages (Fig. 9c), ECs (Fig. 9d), and SMCs (Fig. 9e). Notably, SMCs within the atherosclerotic milieu exhibited characteristics akin to immunocyte-like cells, consistent with previous research^{16,17}. PCSK9 expression was identified in both SMA+F4/80+ and SMA+F4/80- cell populations, emphasizing its relevance across these cell types (Fig. 9f).

Fig. 9c. Immunofluorescence images of arteries under D-flow from *Cap1*^{+/+} mice injected with AdV-PCSK9 showing colocalization of PCSK9 (red), CAP1 (gray), and F4/80 (green). The scale bar represents 5 μm.

Fig. 9d. Immunofluorescence images of arteries under D-flow from *Cap1*^{+/+} mice injected with AdV-PCSK9 showing colocalization of PCSK9 (red), CAP1 (gray), and CD31 (green). The scale bar represents 5 μ m.

Fig. 9e. Immunofluorescence images of arteries under D-flow from *Cap1*^{+/+} mice injected with AdV-PCSK9 showing colocalization of PCSK9 (red), CAP1 (gray), and α SMA (green). The scale bar represents 10 μ m.

Fig. 9f. Immunofluorescence staining for PCSK9 (red), α SMA (green), and F4/80 (gray) in arteries under D-flow from *Cap1^{+/+}* mice injected with AdV-PCSK9. White arrow indicates PCSK9, α SMA, and F4/80 colocalization. The yellow arrow denotes PCSK9 and α SMA colocalization (top middle panel). The three bottom panels show the colocalization of PCSK9 and α SMA (bottom left), PCSK9 and F4/80 (bottom middle), and α SMA and F4/80 (bottom right), respectively. The scale bar represents 10 μ m.

[Comment #1-6]: HE staining should also be performed by cross-sectional analysis in Fig 8e similar to Fig 2c.

[Response #1-6]: In line with the reviewer's suggestion, we performed a cross-sectional analysis in Figure 8e, which is similar to the approach employed in Figure 2c. Serial sections were obtained from the aortic root at 0.3, 0.6, and 1 mm intervals. In the cross-sectional histological analysis (Fig. 8e), compared with the AAV-CTRL injection group, AAV-PCSK9 injection led to an increase in plaque volume and arterial thickness, but these effects were attenuated in *Cap1^{+/-}* mice. By adopting an alternative approach, we could enhance the

visualization of atherosclerosis development, surpassing the limitations of solely presenting a single cross-sectional view.

Changes in the Manuscript Page 9, line 239, Figure 8e

In the cross-sectional histological analysis, compared with the AdV-CTRL injection group, AdV-PCSK9 injection led to an increase in plaque volume and arterial thickness, but these effects were attenuated in Cap1^{+/-} mice (Fig. 8e).

Fig. 8e. Hematoxylin and eosin staining of serial sections of arteries (from the aortic root at 0.3, 0.6, and 1 mm, respectively) under S-flow or D-flow from *Cap1*^{+/+} versus *Cap1*^{+/-} mice injected with AdV-CTRL or AdV-PCSK9. In the presence of a high serum level of PCSK9, significant atherosclerotic plaques developed under D-flow in *Cap1*^{+/+} mice, which was effectively prevented in *Cap1*^{+/-} mice. The scale bar represents 200 μm.

[Comment #1-7]: All RNA interference experiments should have interference efficiency measurements. The author should synthesize at least three RNA interference chains for each target gene and detect the interference efficiency of each RNA interference chain. The best one can be chosen for further research.

[Response #1-7]: In this study, we used siRNA constructs, specifically siCAP1 from the published paper by Yamazaki, K. et al. (*Lab Invest*, 2009, 89, 425–432.) as well as siTLR4 and siLOX1 from Dharmacon ON-TARGETplus SMARTpool siRNA. The CAP1 siRNA construct used in this study targets the 5'-aaaccgagtcctcaagagta-3' sequence of human CAP1. This siRNA construct used in our research elucidated the role of CAP1 in the following contexts, and it effectively knocked down CAP1 protein levels (Reviewer only Fig. 1-1):

- 1) The identification of CAP1 as the functional receptor for resistin. (Lee, S. et al., *Cell Metab*, 2014, 19, 484–497).
- 2) CAP1 as a new binding partner of PCSK9 and its role as a key mediator of caveolae-dependent endocytosis and lysosomal degradation of LDLR. (Jang, H.D. et al., *Eur Heart J*, 2020, 41, 239–252).

Yamazaki, K. et al. (*Lab Invest.*, 2009, 89(4), 425–432) conducted a comparative analysis of two distinct siRNA molecules, siCAP1A and siCAP1B. Their investigation demonstrated the effective knockdown of CAP1 in both CFPAC-1 and PANC-1 cell lines using siCAP1A. Additionally, Tan M. et al. (*Oncol Rep.*, 2013, 30, 1639–1644.) utilized the identical siCAP1A sequence from Yamazaki's work in their own research. The identical siCAP1A sequence was also applied in our research (Reviewer only Fig. 1-2).

[FIGURE REDACTED]

Reviewer only Fig. 1-2. Effects of CAP1 knockdown. Yamazaki, K. et al., *Lab Invest*, 2009, 89, 425–432 (*left panel*) and Tan, M. et al., *Oncol Rep*, 2013, 30, 1639–1644 (*right panel*).

In this study, we also used siRNA constructs targeting TLR4 and LOX1. The siRNA oligos were obtained from Dharmacon ON-TARGETplus SMARTpool siRNA, and each consisted of a mixture of four sequences (Reviewer only Fig. 1-3). For efficient knockdown of the target

genes, the siRNA oligos were transfected using RNAiMax Lipofectamine (Invitrogen Life Technologies) following the manufacturer's instructions.

TLR4	Target (5' - to 3')	OLR1 (LOX1)	Target (5' - to 3')
TLR4-1	CUAGCUUUCUAAAUCUUA	OLR1-1	GGAUUAGUAGUGACCAUUA
TLR4-2	CUCUCUACCUAAUAUUGA	OLR1-2	UGUCUGACCUCCU AACACA
TLR4-3	UUCUGGACUAUCAAGUUUA	OLR1-3	CAGAAGAAGGCAAACCUAA
TLR4-1	CCUAUAAGCUAAUAUCAUA	OLR1-4	GAGAAGUGCUUGUCUUUGG

Reviewer only Fig. 1-3. Sequence of siTLR4 and siLOX1.

[Comment #1-8]: The author should discuss the relevant research progress and the significance of this study in the discussion section instead of repeating all the experimental results.

[Response #1-8]: We have taken your suggestions into careful consideration and made corresponding revisions to enhance the quality of our manuscript. Instead of repeating the experiments, we compared our results with previous findings to delineate the PCSK9- and CAP1-mediated inflammatory mechanism, and we further emphasized the translational perspective of our research. Our findings will extend the clinical significance of PCSK9 from the control of LDL-C levels to the modulation of inflammatory signaling, provoking new analyses of the published landmark trials using evolocumab, alirocumab, or inclisiran as well as the designing of new clinical trials. We have extended our discussion to encompass the clinical relevance of CAP1 within the context of atherosclerosis-related pathologies. In addition, we have introduced a direct comparison experiment between Fc-CAP1 and evolocumab, an existing PCSK9 antibody, which highlighted the therapeutic potential of CAP1. Moreover, in line with your recommendation, we conducted a correlation study using blood

samples from patients with coronary artery disease (CAD) and healthy volunteers. This investigation aims to elucidate the relationship between PCSK9 levels and the magnitude of inflammation. We anticipate that these results will provide a solid foundation for the development of more comprehensive therapeutic strategies, targeting dyslipidemia, atherosclerotic cardiovascular diseases, and inflammatory diseases.

Notably, we have expanded our discussion to incorporate the pathology of CAP1 and extended our contemplation to practical therapeutic implications, including a comparative assessment with the rapidly advancing field of PCSK9 antibody research. In conclusion, we express our gratitude for the reviewer's valuable insights, which have guided us in strengthening both the depth and clinical relevance of our study. The revisions we have implemented align our work more effectively with the broader scientific landscape.

[Comment #1-9]: The background of heterozygous Cap1 knockout mice should be described in detail. Moreover, the author needs to compare Cap1 gene and CAP1 protein expression of heterozygous Cap1 knockout mice to Wild type mice. Furthermore, homozygous Cap1 knockout mice must be better than heterozygous mice.

[Response #1-9]: In our previous report (Jang, H.D. et al., *Eur Heart J*, 2020, 41, 239–252), we successfully engineered a CAP1 knockout mouse using precise TALEN-mediated targeting of CAP1 exon 3 (Reviewer only Fig. 1-4). This approach provided us with a valuable platform to elucidate the intricate *in vivo* functions of CAP1.

[FIGURE REDACTED]

Reviewer only Fig. 1-4. T7E1 assay and sequencing of CAP1 in mouse embryonic stem cells treated with TALEN targeting exon 3 of CAP1; Jang, H.D. et al., *Eur Heart J*, 2020, 41, 239–252.

We concluded that homozygous *Cap1* knockout mice may be better than heterozygous mice; however, *CAP1* homo-knockout mice ($CAP1^{-/-}$) were not found in the neonatal population when heterozygous *CAP1* knockout mice ($CAP1^{+/-}$) were crossed. At embryonic day 14.5, $CAP1^{-/-}$ null embryos were observed at the expected Mendelian ratio, but after embryonic day 16.5, $CAP1^{-/-}$ embryos were no longer observed (Reviewer only Fig. 1-5).

[FIGURE REDACTED]

Reviewer only Fig. 1-5. Genotyping of the offspring of $CAP1^{+/-}$ mice showing that $CAP1^{-/-}$ in mice is lethal on embryonic day 16.5; Jang, H.D., et al. *Eur Heart J*, 2020, 41, 239–252.

Owing to the lethality observed in *CAP1* knockout mice between embryonic days 14.5 and 16.5, only heterozygous knockout mice ($CAP1^{+/-}$) were utilized for subsequent experiments. In the analysis of organs from 8- to 10-week-old mice, there were no discernible morphological differences between $CAP1^{+/-}$ and $CAP1^{+/+}$ mice. However, both mRNA and protein levels of *CAP1* were reduced in $CAP1^{+/-}$ mice compared with those in $CAP1^{+/+}$ mice (Reviewer only Fig. 1-6 and 1-7).

Reviewer only Fig. 1-6. Gross morphologies of each organ did not differ between the CAP1^{+/+} and CAP1^{+/-} mice.

[FIGURE REDACTED]

Reviewer only Fig. 1-7. (a) Real-time PCR illustrating decreased mRNA levels of *CAP1* in each organ in CAP1^{+/-} mice (n=3 per group). (b) Immunoblot analysis demonstrating decreased protein levels of CAP1 in each organ in CAP1^{+/-} mice; Jang, H.D. et al., *Eur Heart J*, 2020, 41, 239–252.

Comments from Reviewer #2 and Responses

This study gives a completely new mechanistic insight by which PCSK9 contributes to atherosclerosis, beyond LDLR, by triggering inflammation upon binding to CAP1, a human resistin receptor. The authors gave a thorough mechanistic insight into this interaction, identifying two novel proteins in the signalling cascade of PCSK9/CAP1 complex, SYK and PKC δ . NF- κ B and inflammatory genes were induced in monocytes and induced atherosclerosis in Ldlr $^{-/-}$ mice, as well as rhPCSK9 induced phosphorylation of SYK and PKC δ , which confirmed PCSK9-induced inflammation was triggered by its binding to CAP1, followed by downstream activation of SYK and PKC δ , and that this mechanism was independent of LDLR. The authors show that CAP1 is the major receptor of PCSK9.

This paper could have a major impact on future targeted therapy development for atherosclerosis and inflammation-related diseases. The methodology is sound and provided in enough detail in the methods for the work to be reproduced.

[Response]: We thank the reviewers for the valuable insights provided. A point-by-point response has been provided below.

[Minor Comment #2-1]: All P values should be stated as upper cases.

[Response #2-1]: As the reviewer suggested, we have changed all the P values used in this paper to upper case.

[Comment #2-2]: Line 119: P value should be at three decimal points.

[Response #2-2]: As suggested, we changed all P values in our manuscript to three decimal points. However, we only added two decimal points for P values when the third decimal point was 0. For example, when P value was 0.0103, we wrote it as 0.01.

[Comment #2-3]: Line 140 “Resistin, pro-inflammatory cytokines (TNF- α , IL-1 β , IL-6), LPS and PCSK9 itself (p=0.005, p=0.0003, respectively) increased PCSK9 expression (Fig. 5d).” It is not clear which P values are stated, there are only two, and more than two parameters in the sentence.

[Response #2-3]: We paraphrased this confusing sentence to make it easier to understand.

Changes in the Manuscript Page 7, line 178

We deleted the section below:

Resistin, pro-inflammatory cytokines (TNF- α , IL-1 β , IL-6), LPS, and PCSK9 itself (p=0.005, p=0.0003, respectively) increased PCSK9 expression (Fig. 5d).

We have rewritten this text as follows:

PCSK9 expression was significantly induced by pro-inflammatory cytokines (TNF- α , IL-1 β , IL-6; $P < 0.001$, $P < 0.001$, $P < 0.001$, respectively) and LPS ($P = 0.002$; Fig. 5e). In addition, treatment with rhPCSK9 and another CAPI ligand, resistin, showed a significant increase in PCSK9 expression ($P < 0.001$ and $P < 0.001$, respectively), suggesting that PCSK9 is induced not only in the pro-inflammatory environment but also due to the autocrine effect of PCSK9 (Fig. 5e).

[Comment #2-4]: It would also be of notion to state in the discussion part if there are any clinical studies so far which analyzed CAP1 in the context of atherosclerosis-related pathologies.

[Response #2-4]: Thank you for your comment. Resistin, a cysteine-rich peptide hormone, triggers pathways that lead to vascular inflammation, lipid accumulation, and plaque vulnerability in atherosclerosis. Meanwhile, CAP1, the receptor for resistin, activates NF- κ B via the cAMP/PKA pathway (Jang, H.D. et al., *Eur Heart J*, 2020, 41, 239–252). The significance of CAP1 is highlighted by its elevated mRNA expression in CAD patients and its positive correlation with carotid intima-media thickness in patients with end-stage renal disease (ESRD) (Munjas, J. et al., *Eur J Clin Invest*, 2017, 47, 659–666; *Cardiorenal Med*, 2020, 10, 51–60). Specifically, Munjas, J. et al. (*Eur J Clin Invest*, 2017, 47, 659–666) investigated the association of CAP1 with CAD. They found that the CAP1 mRNA was significantly higher in peripheral blood mononuclear cells of patients with CAD than healthy controls (Reviewer only Fig. 2-1).

[FIGURE REDACTED]

Reviewer only Fig. 2-1. PBMC CAP1 mRNA in CG, significant CAD, and nonsignificant CAD group. Munjas, J. et al., *Eur J Clin Invest*, 2017, 47, 659–666.

Additionally, Munjas, J. et al. (*Cardiorenal Med*, 2020, 10, 51–60) showed not only that patients with ESRD have significantly higher CAP1 gene expression level in PBMCs, but also that carotid intima-media thickness is positively correlated with CAP1 (Reviewer only Fig. 2-2).

[FIGURE REDACTED]

Reviewer only Fig. 2-2. *CAP1* mRNA level is increased in ESRD (left panel) and CIMT is positively correlated with *CAP1* (right panel). Munjas, J. et al., *Cardiorenal Med*, 2020, 10, 51–60.

Moreover, there are several non-clinical studies such as those by Zhou, L. et al. (*Clinica Chimica Acta*, 2021, 512, 84–91) and Emily Punch et al. (*J Am Heart Assoc*, 2022, 37, 364–371) stating that resistin binding to *CAP1* can trigger various intracellular signal transduction pathways to induce vascular inflammation, lipid accumulation, and plaque vulnerability, which may lead to atherosclerosis. We added *CAP1*-related pathologies in the Discussion as follows:

Changes in the Manuscript (Discussion) Page 13, line 327

*Resistin, a peptide hormone rich in cysteine, initiates diverse pathways within cells to prompt vascular inflammation, the buildup of lipids, and heightened susceptibility of plaques. This positions resistin as a possible biomarker and treatment target for atherosclerosis¹⁹. We have previously reported that *CAP1*, the receptor for human resistin, leads to NF- κ B activation via the cAMP/PKA pathway⁸. Interestingly, *CAP1* mRNA expression is reported to be not only significantly increased in CAD patients^{21,22}, but also positively correlated with the carotid intima-media thickness in patients with end-stage renal disease²². We observed that *CAP1* directly binds to the C-terminal cysteine-rich domain of PCSK9, which is structurally similar to the globular C-terminal of the resistin trimer⁷. Therefore, we speculated that PCSK9 binding to *CAP1* may switch on pro-inflammatory signaling and further aggravate atherosclerosis independently of LDLR.*

[Comment #2-5]: Abstract section: Lines 25/26: we treated PCSK9 in the monocyte and endothelial cells “The sentence is not clear, should be more appropriately described.

Abbreviations lacking for AAV-PCSK9 a CAP1, TLR4

[Response #2-5]: We have revised this sentence to provide an accurate description and have included the full names for the following entities in the abstract in red: **proprotein convertase subtilisin/kexin type 9 (PCSK9), low-density lipoprotein receptor (LDLR), adenovirus (AdV)-PCSK9, adenylyl cyclase-associated protein 1 (CAP1), Toll like receptor 4 (TLR4) and peripheral blood mononuclear cells (PBMCs).**

Changes in the Manuscript Page 2, line 30, Abstract

We deleted the section below:

To study the direct inflammatory effect of human PCSK9 (hPCSK9) on atherosclerosis, **we treated PCSK9 in the monocyte and endothelial cells.** PCSK9 activates nuclear factor (NF)- κ B, pro-inflammatory cytokines, and adhesion molecules.

We have rewritten this bolded text as follows:

The direct inflammatory action of proprotein convertase subtilisin/kexin type 9 (PCSK9) was examined in vitro in monocytes and endothelial cells and an in vivo atherosclerosis animal model.

Comments from Reviewer #3 and Responses

This interesting study was carried out to establish the role of PCSK9 in monocyte activation and atherosclerosis via LDLR independent mechanism. The topic is timely and has very high clinical impact. The study is a logical extension of the same group's previous study that established that PCSK9-CAP1 binding is a key mediator of caveolae-dependent endocytosis and lysosomal degradation of LDLR (Jang et al, Eur Heart J 2020). Overall, the present manuscript is of potential interest and the experiments seem to have been carefully executed. Some approaches are sophisticated. This reviewer, however, found various concerns and suggested below solutions to further strengthen the manuscript.

[Response]: We thank the reviewer for their insightful comments. We have provided point-by-point responses below.

[Comment #3-1]: The authors did not address mechanisms in depth, leaving the manuscript somewhat superficial. This is a major weakness. For instance, in in vivo experiments, the authors did not examine possible mechanisms by which AAV-PCSK9 and CAP1 deficiency affected the lesion size of mouse carotid arteries.

[Response #3-1]: We appreciate the constructive feedback provided by the reviewer, which has significantly contributed to improving our research. In the first submission, we presented the gross image, H&E, qPCR, and macrophage infiltration to compare CAP1 hetero-knockout mice and WT mice, implying that CAP1 deficiency conferred protective effects against atherosclerosis. As the reviewer pointed out, although we demonstrated the mechanism in depth *in vitro*, it is somewhat superficial to assert that Adv-PCSK9 and CAP1 deficiency

affected the lesion size of mouse carotid arteries. To address this concern, we conducted additional mouse experiments and confirmed important mechanisms through immunostaining *in vivo*, which were only verified *in vitro* before. First, we demonstrated that PCSK9 activates pro-inflammatory genes directly, independently of LDLR, both *in vitro* and *in vivo*. Through staining PCSK9 and CAP1 in *Ldlr*^{-/-} arteries, we observed an elevation in their colocalization in AdV-PCSK9-treated mice, which was further augmented under D-flow conditions (Fig. 3g).

Changes in the Manuscript Page 6, line 139, Figure 3g

*Furthermore, we investigated the colocalization of PCSK9 and CAP1 in the context of atherosclerosis in an *Ldlr*^{-/-} mouse model using partial carotid ligation. Immunostaining in arterial tissues revealed that the colocalization of PCSK9 and CAP1 increased in AdV-PCSK9-treated mice and further enhanced under D-flow conditions by partial ligation (Fig. 3g).*

Fig. 3g. Immunofluorescence staining of CAP1 (green) and PCSK9 (red) in the arteries of AdV-CTRL or AdV-PCSK9-treated *Ldlr*^{-/-} mice under S-flow and D-flow. CAP1 and PCSK9 were colocalized in all groups. CAP1 and PCSK9 expression increased in the AdV-PCSK9 injection group compared with that in the AdV-CTRL group under D-flow. The scale bar represents 20 μm . The differences between the groups were compared using the unpaired *t*-test.

Next, we examined arteries from *Cap1*^{+/+} and *Cap1*^{+/-} mice exposed to D-flow to explore potential mechanisms underlying the impact of AAV-PCSK9 on mouse carotid artery lesion size under CAP1 deficiency. As anticipated, overexpression of PCSK9 in *Cap1*^{+/+} mice resulted in elevated Syk and PKC δ phosphorylation, an effect that was mitigated by *Cap1*^{+/-} (Fig. 9b). Finally, we updated the main figures and completed the schematic diagram.

Changes in the Manuscript Page 10, line 251, Figure 9b

Furthermore, elevated serum levels of PCSK9 led to a significant increase in Syk and PKC δ phosphorylation in vivo ($P=0.03$, $P<0.001$, respectively), both of which were markedly reduced in $Cap1^{+/-}$ mice ($P=0.007$, $P=0.006$, respectively) (Fig. 9b).

Fig. 9b. Partially ligated $Cap1^{+/+}$ and $Cap1^{+/-}$ mice with only arteries exposed to D-flow were compared after AdV-CTRL or AdV-PCSK9 injection. Immunofluorescence staining of PCSK9 (red, upper panels), p-Syk (gray, middle panels), and p-PKC δ (green, bottom panels). AdV-PCSK9 injection increased PCSK9 expression in $Cap1^{+/+}$ mice, which was lesser than that in $Cap1^{+/-}$ mice. Additionally, p-Syk and p-PKC δ increased more significantly in $Cap1^{+/+}$ mice than in $Cap1^{+/-}$ mice ($N=3$). The scale bar represents 10 μ m.

Changes in the Manuscript Page 10, line 261, Figure 9g

As summarized in Fig. 9g, PCSK9 binds to CAP1, leading to the activation of Syk and PKCδ and induction of inflammatory gene cascades, TLR4, and scavenger receptors on monocytes and adhesion molecules on ECs. These actions of PCSK9 were dependent on CAP1 and thus attenuated by CAP1 depletion. In vivo systemic injection of AdV-PCSK9 significantly induced atherosclerosis of the carotid artery exposed to D-flow, which was prevented in Cap^{+/-} mice. These results suggested that the PCSK9-CAP1-PKCδ/Syk pathway may be a viable target for developing new therapeutics for dyslipidemia, atherosclerotic cardiovascular, and inflammation-based diseases.

Fig. 9g. Schematic model showing PCSK9-mediated inflammation in monocytes mediated by CAP1 recruiting PKC δ and Syk and modulating PCSK9-mediated inflammatory signal transduction. The differences between the groups were compared using the unpaired t-test.

[Comment #3-2]: The numbers of animals used in the experiment are small (Figure 2b, c and Figure 8). This raised a concern related to scientific rigor.

[Response #3-2]: In response to your suggestion, we performed a new set of mouse experiments with an increased sample size to enhance the statistical power and reliability of our analyses. These adjustments allowed us to conduct additional analyses and gain further insights into our research question.

Specifically, in LDLR^{-/-} mice, we increased the sample size for the AdV-CRTL group from n=3 to n=8, and for the AdV-PCSK9 group, we expanded the sample size from n=10 to n=12.

Notably, our findings were consistent with our previous observations, revealing a significant increase in lesion area attributable to AdV-PCSK9. This effect was evident in both the quantitative analysis (Fig. 2b) and individual gross images (Reviewer only Fig. 3-1)

Changes in the Manuscript *Figure 2b and 8c*

Fig. 2b. Oil red O staining of whole carotid arteries shows atherosclerotic plaque in the partial ligation-induced carotid atherosclerosis in *Ldlr*^{-/-} mice. The lesion area was significantly broader in AdV-PCSK9-treated *Ldlr*^{-/-} mice (44.4% of total RCA) than in AdV-control mice (CTRL) (22.5%) (N ≥ 8).

Gross images of the arteries from LDLR ^{-/-} mouse

Reviewer only Fig. 3-1. Lesion area and gross findings of carotid arteries in *Ldlr*^{-/-} mice following AdV-CTRL or AdV-PCSK9 injection. The lesion area was assessed by measuring the plaque area in the right carotid artery (RCA) as a proportion of the total RCA area.

Furthermore, we increased the sample sizes for the corresponding groups: WT AdV-CRTL, WT AdV-PCSK9, CAP1^{+/-} AdV-CTRL, and CAP1^{+/-} AdV-PCSK9, from n=3, n=13, n=4, and n=12 to n=7, n=18, n=8, and n=14, respectively. Remarkably, in WT mice, AdV-PCSK9 injection resulted in a significant increase in lesion area, whereas this effect was attenuated in *Cap1*^{+/-} mice (Fig. 8c and Reviewer only Fig.3-2). These findings further support our previous observations and provide additional insights into the role of AdV-PCSK9 in lesion development.

Fig. 8c. Oil Red O staining of both carotid arteries after AdV-CTRL or AdV-PCSK9 was injected into *Cap1*^{+/+} and *Cap1*^{+/-} mice ($N \geq 7$). The atherosclerotic plaque area in the arteries under D-flow increased significantly after PCSK9 administration in *Cap1*^{+/+} mice, which was remarkably prevented in *Cap1*^{+/-} mice. The scale bar represents 2 mm.

Gross images of the arteries from CAP1 WT and CAP1 +/- mouse

Reviewer only Fig. 3-2 Lesion area and gross findings of carotid arteries in WT and *CAP1*^{+/-} mice after AdV-CTRL or AdV-PCSK9 injection. The lesion area was assessed by measuring the plaque area in the right carotid artery (RCA) as a proportion of the total RCA area.

[Comment #3-3]: The use of THP-1 cells for most in vitro experiments is also a major concern.

Cancer cell lines are very different from primary cells.

[Response #3-3]: We highly appreciate the valuable feedback regarding the utilization of THP-1 cells in the context of our research. Despite their cancerous origin, THP-1 cells are extensively utilized in immunological research because of their structural and functional resemblance to primary monocytes and macrophages (Daigneault, M. et al., *PLoS One*, 2010, 5, e8668.2010). These cells have shown value in elucidating mechanisms behind inflammatory diseases and gene expression modulation (Hjort, M.R. et al., *Inflammation*, 2003, 27, 137–145; Kramer, P.R. et al., *Mol Cell Endocrinol*, 2002, 279, 16–25).

We agree that there exist inherent distinctions between THP-1 and primary cells that may impact the significance of our findings; therefore, the inclusion of primary cells for our experiments will enrich our dataset. In response to the reviewer's counsel, we conducted experiments using human primary cells, yielding a substantial improvement in the manuscript's quality (Figure 10). To extend these findings to human primary cells, we obtained human peripheral blood mononuclear cells (PBMCs) through IRB approval. We created a descriptive statement explaining the purpose of this project and a patient consent form, which were submitted to the Cardiovascular Center and Biomedical Institute, Seoul National University Hospital, Seoul, Republic of Korea. Subsequently, we collected 50 cc of blood from 14 patients diagnosed with CAD and five healthy donors. Consistently, we observed the same PCSK9-induced inflammation phenomenon in both THP-1 and human primary cells.

Furthermore, our experiments on inflammatory pathways using human PBMCs revealed significant insights. In line with our findings in THP-1 cells, we found a notable correlation between serum concentration of PCSK9 and phosphorylation of Syk, PKC δ , p65 (S276), and p65 (S536) in human PBMCs (Fig. 10a).

Changes in the Manuscript Page 11, line 272, *Figure 10a*

To investigate the potential association between human serum PCSK9 levels and its effect on inflammatory signaling pathways, blood samples were collected from individuals diagnosed with CAD and healthy donors. CAD patients were all under strict statin treatment. Pearson's correlation and simple linear regression analyses were used to examine the relationship between PCSK9 concentration in the serum and phosphorylation of Syk, PKC δ , p65(S276), and p65(s536) in PBMCs. Interestingly, serum PCSK9 concentration showed a positive correlation with the quantified phosphorylation of Syk ($P=0.03$, $R=0.498$), PKC δ ($P=0.049$, $R=0.458$), p65 (S276, $P=0.168$, $R=0.33$), and p65 (S536, $P=0.022$, $R=0.52$) in matched PBMCs (Fig. 10a and Supplementary Fig. 3).

Fig. 10

Fig. 10a. Simple linear regression analysis of Pearson correlation by distance. This figure illustrates the relationship between serum PCSK9 concentration and the quantified levels of phosphorylated proteins in a single individual. Serum PCSK9 concentration displayed a positive correlation with the phosphorylation of Syk, PKC δ , p65(S276), and p65(S536) in human peripheral blood mononuclear cells (PBMCs). The black line represents an asymptotic regression line fitted to the raw data (N=19).

Our investigation expanded to include the differentiation of human PBMCs into macrophages. This allowed us to explore the complex landscape of signaling pathways in greater detail. In human macrophages differentiated from PBMCs, PCSK9 induced significant phosphorylation of Syk, PKC δ , and p65 (S276 and S536) ($P < 0.001$, $P < 0.002$, $P < 0.001$, and $P = 0.002$, respectively). Notably, the phosphorylation of Syk, PKC δ , and p65 was more effectively and significantly attenuated by CAP1-hFc than by evolocumab (Fig. 10e). As summarized in Figure 10f, based on these findings, we developed a schematic figure illustrating the therapeutic implications of PCSK and CAP1.

Changes in the Manuscript Page 12, line 302, Figure 10e and 10f

PCSK9 significantly induced the phosphorylation of Syk, PKC δ , and p65 (S276 and S536) ($P < 0.001$, $P < 0.001$, $P < 0.001$, and $P = 0.002$, respectively) in human PBMC-derived macrophages, which was more effectively and significantly attenuated by CAP1-hFc ($P < 0.001$, $P = 0.002$, $P < 0.001$, and $P = 0.013$, respectively) than by evolocumab ($P = 0.003$, $P = 0.038$, $P = 0.002$, and $P = 0.03$, respectively) (Fig. 10e).

Fig. 10e. Immunoblot analysis demonstrated that PCSK9-induced phosphorylation of Syk, PKC δ , and NF- κ B p65 in human PBMC-derived macrophages was attenuated by CAP1-hFc. After 1-h treatment of rhPCSK9 with human IgG, evolocumab, or CAP1-hFc, the PCSK9-induced phosphorylation of Syk, PKC δ , and NF- κ B p65 was significantly decreased by CAP1-hFc (N \geq 7).

Fig. 10f. Schematic diagram depicting CAP1 as the binding partner of PCSK9, which mediates not only caveolae-dependent endocytosis and lysosomal degradation of LDLR, but also recruits Syk and PKC δ and modulates PCSK9-mediated inflammatory signal transduction. CAP1-hFc inhibits the binding of CAP1 and PCSK9, which sequentially block LDLR degradation and the inflammatory signaling pathway, whereas the PCSK9 inhibitor evolocumab can only block the LDLR degradation pathway. The differences between the groups were compared using the unpaired t-test.

In conclusion, while acknowledging concerns about the differences between THP-1 cells and primary cells, their functional and practical benefits warrant their inclusion in our investigation. Validating our findings through experiments with human PBMCs enhances the credibility and applicability of our study's outcomes to broader pathophysiologic contexts, providing deeper insights into intricate correlations and cellular responses.

[Comment #3-4]: It is widely known that undetectable traces of endotoxin on a recombinant protein, PCSK9 in the present manuscript, can cause cell activation as evidenced in many previous studies. This possibility needs to be ruled out via more thorough examinations.

[Response #3-4]: In our study, we utilized recombinant human PCSK9 (rhPCSK9) protein obtained from the *Escherichia coli* expression system with a histidine (His) tag, which was provided by the Y-biologics. The purification process involved an Ni-NTA column, followed by buffer exchange with Dulbecco's PBS. Considering the potential for endotoxin contamination, we assessed endotoxin levels in various lots of rhPCSK9 using limulus amoebocyte lysate (LAL) analysis with the Endosafe nexgen-PTS system. The results indicated values of 0.168 endotoxin units (EU)/ml (0.0168 ng/μg) and 0.96 EU/ml (0.096 ng/μg) for the tested lots (Reviewer only Fig. 3-3).

SAMPLE DATA				SPIKE DATA				
Channel	Reaction Time	CV%	Sample Value	Channel	Reaction Time	CV%	Spike Value	Spike Recovery %
1	786	8.4 %	0.168 EU/mL	2	460	1.6 %	0.066 EU/mL	68 %
3	698			4	450			

SAMPLE DATA				SPIKE DATA				
Channel	Reaction Time	CV%	Sample Value	Channel	Reaction Time	CV%	Spike Value	Spike Recovery %
1	392	3.7 %	0.960 EU/mL	2	302	4.1 %	0.082 EU/mL	107 %
3	372			4	320			

Reviewer only Fig. 3-3. Measurement of endotoxin concentrations in the recombinant human PCSK9 proteins. The amount of LPS was measured using the LAL endotoxin test kit and calculated using the standard curve.

While acknowledging the variability of the endotoxin response across different cell types (Meng, F. et al., *J Exp Med*, 1997, 85, 1661–1670; Chaiwut, R. et al., *BMC Res Notes*, 2022, 15, 42; and Leister, K.P. et al., *Curr Chem Genomics*, 2011, 5, 21), it is noteworthy that numerous commercially available recombinant proteins are produced in *E. coli*, with most suppliers guaranteeing contamination levels **below 1 EU** (Schwarz, H. et al., *PLoS One*, 2014, 9, e113840).

In our study, we further investigated the inflammatory response by treating the THP-1 cell line with LPS (Supplementary Fig. 2). Our results demonstrated that 20 ng/ml LPS induced an inflammatory signal under both CTRL and CAP1 siRNA conditions, highlighting consistent LPS signaling independent of CAP1 expression.

Supplementary Fig. 2. Effects of siRNA against CAP1 on LPS-induced phosphorylation of p65, Akt, Erk, and p38. THP-1 cells were transfected with CAP1 siRNA or non-targeting control (CTRL siRNA) and cultured for 3 days after LPS treatment in a time-dependent manner (0, 10, 30, and 60 min). Immunoblot analysis demonstrating

that CAP1-deficient cells showed LPS-mediated pro-inflammatory signaling equivalent to control cells. The representative figures of three independent experiments are shown.

Conversely, the inflammatory signal induced by rhPCSK9 was attenuated in CAP1-deficient cells. This suggests that the observed reduction in the inflammatory signal can be attributed to the inflammatory role of rhPCSK9 itself, rather than endotoxin-mediated effects (Fig. 4a and 4b).

Considering the available references and our endotoxin assessments, we believe that the observed endotoxin levels in rhPCSK9 are within the acceptable range, suggesting minimal or negligible risk of immune activation.

Fig. 4

Fig. 4a and 4b (a) Immunoblot demonstrating that rhPCSK9 treatment (2 µg/ml for 20 min) resulted in NF-κB p65 phosphorylation, which was blocked in CAP1-deficient THP-1 cells, but not in TLR4-deficient THP-1 cells (N=3). (b) rhPCSK9 treatment (2 µg/ml, 48 h) elevated TNF-α, IL-1β, and IL-6 protein levels in THP-1 cells treated with CTRL and TLR4 siRNA, but not in CAP1-deficient THP-1 cells (N=3).

[Comment #3-5]: The authors examined the atherosclerotic lesion area and oil red O-positive area of carotid arteries. This is insufficient. Other assays/parameters (e.g., cellular components, cell growth, apoptosis, necrotic core size, etc) would help to not only support the authors' theory but also understand what PCSK9 really did to affect lesion development in this model.

[Response #3-5]: In response to this valuable feedback, we conducted further experimentation using murine models for a more exhaustive analysis of the atherosclerotic lesion area. We validated crucial mechanisms through *in vivo* immunostaining, corroborating previously established *in vitro* findings. Additionally, we have updated our main figures and finalized the schematic diagram.

Upon meticulous examination of the data, a noteworthy increase in the fibrotic area (as indicated by Masson's Trichrome [MT] staining) and lipid accumulation (visualized through oil red O staining) is apparent within the carotid artery following AdV-PCSK9 injection under conditions of D-flow in *Ldlr*^{-/-} mice (Fig. 2d and 2e). Furthermore, the number of terminal deoxynucleotidyl transferase dUTP nick end labeling (TUNEL)-positive cells was distinctly elevated in the arteries of *Ldlr*^{-/-} mice injected with AdV-PCSK9 under D-flow (Fig. 2f). Elevated levels of PCSK9 were observed in AdV-PCSK9-injected arteries obtained from *Ldlr*^{-/-} mice. This elevation is particularly pronounced under conditions of D-flow, as indicated by the P value of 0.067 (Fig. 2g). Notably, a significant increase in the infiltration of macrophages was also observed within the AdV-PCSK9-injected arteries exposed to D-flow, with a distinct contrast to the arteries subjected to S-flow conditions (P = 0.002). By conducting these additional experiments and analyses, we have expanded the scope of our investigation, which not only strengthens the support for our proposed theory but also provides deeper insights into the intricate effects of PCSK9 on lesion development in our experimental model.

Changes in the Manuscript Page 5, line 105, Figure 2d, 2e, 2f and 2g

*Systemic administration of AdV-PCSK9 in *Ldlr*^{-/-} mice substantially increased the area of fibrosis (Masson's trichrome stain) and lipid accumulation (Oil red O) in the arteries exposed to D-flow (Fig. 2d and 2e). Furthermore, AdV-PCSK9 increased the number of terminal deoxynucleotidyl transferase dUTP nick end labeling (TUNEL)-positive cells in the arteries exposed to D-flow in *Ldlr*^{-/-} mice (Fig. 2f). The overall PCSK9 expression was greater in the arteries injected with AdV-PCSK9 in *Ldlr*^{-/-} mice than in those injected with AdV-CTRL. This difference further increased in the ligated carotid artery exposed to D-flow (P=0.067; Fig. 2g).*

Additionally, compared with AdV-CTRL, AdV-PCSK9 administration significantly increased the infiltration of macrophages in the arteries exposed to D-flow ($P=0.002$; Fig. 2g).

Fig. 2d, 2e, and 2f. (d-f) Carotid artery cross-section staining shows atherosclerotic plaque development in partial ligation-induced atherosclerosis in *Ldlr*^{-/-} mice. Enlarged atherosclerotic plaques were observed in the arteries of AdV-PCSK9-treated mice under D-flow compared with those of AdV-CTRL mice, indicating the significant impact of PCSK9 on atherosclerosis. (d) Masson's trichrome staining. The scale bar represents 200 μ m, and scale bars of magnified fields represent 50 μ m. (e) Oil red O staining. The scale bar represents 200 μ m, and scale bars of magnified fields represent 50 μ m. (f) Immunofluorescence images stained with TUNEL (green). The scale bar represents 100 μ m, and scale bars of magnified fields represent 50 μ m.

Fig. 2g. Immunofluorescence images stained with F4/80 (green) and PCSK9 (red) in *Ldlr*^{-/-}, demonstrating significantly elevated PCSK9 expression in AdV-PCSK9-injected mice and increased F4/80 expression under D-flow. Each scale bar represents 20 μ m (N \geq 3)

[Comment #3-6]: Which cell types are a main source of PCSK9 in the atherosclerotic lesion of the carotid arteries in this model?

[Response #3-6]: According to the review paper by Xia, X. et al. (*Front Cardiovasc Med*, 2021, 8, 764038), PCSK9 is expressed in extrahepatic cells and tissues, such as the vascular SMCs, macrophages, ECs, the pancreas, kidneys, intestine, and central nervous system. Therefore, we stained the right carotid artery of AdV-PCSK9-injected wild-type mice with PCSK9, EC (CD31), SMC (α SMA), and macrophage (F4/80) antibodies. As a result, we observed co-expression of PCSK9 and CAP1 within atherosclerotic plaques, which were evident in macrophages (Fig. 9c), ECs (Fig. 9d), and SMCs (Fig. 9e).

Notably, SMCs within the atherosclerotic milieu exhibited characteristics akin to immune-like cells, consistent with previous research (Rosenfeld, M.E. et al., *Nat Med*, 2015, 21, 549–551; Gomez, D. et al., *Cardiovasc Res*, 2012, 95, 156–164). PCSK9 expression was identified in both SMA⁺F4/80⁺ and SMA⁺F4/80⁻ cell populations, emphasizing its relevance across these cell types (Fig. 9f).

In conclusion, these observations strongly indicate that both CAP1 and PCSK9 play pivotal roles in coordinating vascular inflammation, promoting macrophage infiltration, and advancing atherosclerosis. These findings significantly enhance our understanding of the intricate interplay between PCSK9, CAP1, and their combined influence on atherosclerotic pathogenesis.

Changes in the Manuscript Page 10, line 253, Figure 9c, 9d, 9e, and 9f

Next, to elucidate the cellular localization of CAP1 and PCSK9 within the atherosclerotic plaque, we conducted immunofluorescence staining of infiltrated macrophages, ECs, and SMCs, which constitute atherosclerotic plaques. We observed that PCSK9 and CAP1 were expressed in atherosclerotic plaques and confirmed their colocalization on macrophages (Fig. 9c), ECs (Fig. 9d), and SMCs (Fig. 9e). Notably, SMCs within the atherosclerotic milieu exhibited characteristics akin to immunocyte-like cells, consistent with previous research^{16,17}. PCSK9 expression was identified in both SMA⁺F4/80⁺ and SMA⁺F4/80⁻ cell populations, emphasizing its relevance across these cell types (Fig. 9f).

Fig. 9c. Immunofluorescence images of arteries under D-flow from *Cap1*^{+/+} mice injected with AdV-PCSK9 showing colocalization of PCSK9 (red), CAP1 (gray), and F4/80 (green). The scale bar represents 5 µm.

Fig. 9d. Immunofluorescence images of arteries under D-flow from *Cap1*^{+/+} mice injected with AdV-PCSK9 showing colocalization of PCSK9 (red), CAP1 (gray), and CD31 (green). The scale bar represents 5 µm.

Fig. 9e. Immunofluorescence images of arteries under D-flow from *Cap1*^{+/+} mice injected with AdV-PCSK9 showing colocalization of PCSK9 (red), CAP1 (gray), and αSMA (green). The scale bar represents 10 μm.

Fig. 9f. Immunofluorescence staining for PCSK9 (red), αSMA (green), and F4/80 (gray) in arteries under D-flow from *Cap1*^{+/+} mice injected with AdV-PCSK9. White arrow indicates PCSK9, αSMA, and F4/80 colocalization. The yellow arrow denotes PCSK9 and αSMA colocalization (top middle panel). The three bottom panels show the colocalization of PCSK9 and αSMA (bottom left), PCSK9 and F4/80 (bottom middle), and αSMA and F4/80 (bottom right), respectively. The scale bar represents 10 μm.

[Comment #3-7]: The authors decided that “immune system process” is the one to go after in the DAVID functional enriched analysis. The immune system process, however, is at the lowest ranking in GO Biological process (Supplementary Fig.3). Please provide a strong rationale behind this choice.

[Response #3-7]: In this study, we used hCAP1-tagged mFc to perform pulldown experiments and identified CAP1-binding proteins using mass spectrometry and the Significance Analysis of INTeractome (SAINT) algorithm (Fig. 6a). Among the 2,103 proteins pulled down by CAP1-mFc, we further analyzed 464 proteins with SAINT AvgP>0.6. The identification of these binding proteins was performed using the Database for Annotation, Visualization, and Integrated Discovery (DAVID) with an EASE score<0.1 for Gene Ontology (GO) term analysis (Fig. 6b). A threshold of EASE score<0.1 is conventionally employed within the DAVID framework to denote statistical significance. The outcomes yielded by this analysis are deemed to bear significance in accordance with scientific standards.

Changes in the Manuscript Page 8, line 197, Figure 6b

Among the 2,103 proteins that were pulled down by CAP1-mFc, 464 proteins with SAINT AvgP>0.6 were sorted and analyzed using the Database for Annotation, Visualization, and Integrated Discovery (DAVID) with an EASE score threshold of <0.1 for Gene Ontology (GO) term analysis. Following GO enrichment analysis, the enriched terms (level 1 of GO terms) in each of three categories (biological process, cellular component, and molecular function) were presented in Figure 6b.

Fig. 6b Revised data

Fig. 6b. Database for Annotation, Visualization, and Integrated Discovery (DAVID) functional enrichment analysis was performed on differentially bound proteins. The graph displays the top-ranked significantly enriched Gene Ontology (GO) terms in biological process, cellular component, and molecular function. All the adjusted statistically significant values of the terms were transformed to their negative 10-base logarithm. The analysis was conducted using an EASE score threshold of <0.1 .

[Comment #3-8]: There are some disconnections between *in vitro* and *in vivo* data. This is also a major concern. How did AAV-PCSK9 affect SYK and PKC δ *in vivo*?

[Response #3-8]: To address the disconnection between our *in vitro* and *in vivo* data, we would like to highlight that our study involved the systemic infusion of adenovirus (AdV)-PCSK9 via intravenous injection. Notably, AdV serotype 5, the selected viral vector, holds broad applicability both *in vitro* and *in vivo* and has demonstrated relevance in clinical studies for addressing genetic and acquired conditions, including cancer (Kalyuzhniy, O. et al., *Proc Natl Acad Sci U S A.* 2008, 105, 5483–5488). Upon intravenous administration, AdV serotype 5

exhibits a distinct tendency for rapid hepatic accumulation, followed by distribution to other organs (Liu, Q. et al., *Sci Rep*, 2017, 7, 3597).

Regarding the liver infection by AdV-PCSK9, it is important to emphasize that the PCSK9 protein maintained consistent and sustained expression, leading to its continual release from hepatic sources. This released PCSK9 exerted an autocrine effect within atherosclerotic plaques (Fig. 5g), significantly amplifying expression levels, particularly in monomacrophages, SMCs, and ECs. This heightened presence of PCSK9 then engaged these cells, triggering robust inflammatory responses via SYK, PKC, and NF- κ B pathway activation.

In our research methodology, we meticulously evaluated a murine atherosclerosis model induced by partially ligating the carotid artery. Employing precise immunostaining techniques, we effectively identified a noteworthy increase in PCSK9 expression within the neointimal region after AdV-PCSK9 administration (Fig. 9a). In contrast, in *Cap1*^{+/-} mice, PCSK9 expression and macrophage infiltration were markedly prevented (P<0.001, P<0.001, respectively) (Fig. 9a). Furthermore, elevated serum levels of PCSK9 and D-flow in arteries *in vivo* led to a significant increase in Syk and PKC δ phosphorylation (P=0.03, P<0.001, respectively), both of which were markedly inhibited in *Cap1*^{+/-} mice (P=0.007, P=0.006, respectively) (Fig. 9b).

In summary, our study sheds light on the intricate dynamics of AdV-PCSK9 administration, emphasizing its consistent impact on PCSK9 expression, macrophage infiltration, and downstream inflammatory responses. These insights contribute to our understanding of atherosclerosis progression and open avenues for exploring novel therapeutic interventions targeting these pathways.

Changes in the Manuscript Figure 5g, 9a and 9b

Fig. 5g. Immunoblot analysis of signal activation induced by PCSK9 in THP-1 cells transfected with CTRL or CAP1 siRNA demonstrating that PCSK9-induced SREBP-2 expression was attenuated when CAP1 was knocked down (N=3).

Fig. 9a. Immunofluorescence staining for PCSK9 (red) and F4/80 (green) in arteries under D-flow in *Cap1*^{+/+} and *Cap1*^{+/-} mice injected with AdV-CTRL or AdV-PCSK9. PCSK9 (red) and F4/80 (green) expression significantly increased in *Cap1*^{+/+} mice with AdV-PCSK9 compared with that in AdV-CTRL. However, in *Cap1*^{+/-} mice, the increase in PCSK9 was less significant and F4/80 showed no significant change (N=3). The scale bar represents 10 μ m.

Fig. 9b. Partially ligated *Cap1*^{+/+} and *Cap1*^{+/-} mice with only arteries exposed to D-flow were compared after AdV-CTRL or AdV-PCSK9 injection. Immunofluorescence staining of PCSK9 (red, upper panels), p-Syk (gray, middle panels), and p-PKCδ (green, bottom panels). AdV-PCSK9 injection increased PCSK9 expression in *Cap1*^{+/+} mice, which was lesser than that in *Cap1*^{+/-} mice. Additionally, p-Syk and p-PKCδ increased more significantly in *Cap1*^{+/+} mice than in *Cap1*^{+/-} mice (N=3). The scale bar represents 10 μm.

[Comment #3-9]: More data need to be quantified. For example, the authors showed that CAP1-deficiency decreased macrophage accumulation in the carotid arteries only qualitatively.

[Response #3-9]: We quantified immunofluorescence and Oil red O images using ImageJ (National Institutes of Health, Bethesda, MD, USA) into bar graphs to show significant differences. Specifically, we showed the significant increase of PCSK9, F4/80 expression (Fig. 2g), phosphorylated SYK, and PKC δ (Fig. 7f) in Adv-PCSK-injected LDLR knockout mice.

Changes in the Manuscript Page 5, line 109, Figure 2g

The overall PCSK9 expression was greater in the arteries injected with Adv-PCSK9 in $Ldlr^{-/-}$ mice than in those injected with Adv-CTRL. This difference further increased in the ligated carotid artery exposed to D-flow ($P=0.067$; Fig. 2g). Additionally, compared with Adv-CTRL, Adv-PCSK9 administration significantly increased the infiltration of macrophages in the arteries exposed to D-flow ($P=0.002$; Fig. 2g).

Changes in the Manuscript Page 9, line 219, Figure 7f

In the $Ldlr^{-/-}$ mouse arteries, PCSK9 expression was greater in mice that received Adv-PCSK9 than in those that received Adv-CTRL ($P=0.001$). Elevated PCSK9 levels in the arterial tissue were associated with a significant increase in the phosphorylation of Syk and PKC δ in neointimal tissue, and this effect was further enhanced by partial ligation ($P=0.001$, $P=0.002$, respectively; Fig. 7f).

Figure 2g. Immunofluorescence images stained with F4/80 (green) and PCSK9 (red) in *Ldlr^{-/-}*, demonstrating significantly elevated PCSK9 expression in AdV-PCSK9-injected mice and increased F4/80 expression under D-flow. Each scale bar represents 20 μ m (N \geq 3).

Additionally, we quantified the extent of lipid accumulation of lipid-laden BMDMs from *Cap1*^{+/+} and *Cap1*^{+/-} mice and showed significant inhibition of ox-LDL uptake in CAP1-deficient BMDMs (Fig. 5d). We also quantified immunofluorescence images of *Cap1*^{+/+} and *Cap1*^{+/-} mice stained with PCSK9, F4/80 (Fig. 9a), and phosphorylated SYK and PKC δ (Fig. 9b) under D-flow.

Changes in the Manuscript Page 7, line 168, Figure 5d

Bone marrow was extracted from WT and CAP1 heterozygous knockout mice (Cap1^{+/-} mice) and the cells were differentiated into BMDMs using monocyte-colony stimulating factor (M-CSF; 50 ng/ml). After ox-LDL treatment (20 μ g/ml), lipid aggregation and transformation into lipid-laden macrophages increased significantly in a time-dependent manner in Cap1^{+/+} BMDMs ($P < 0.001$), whereas lipid aggregation and transformation were inhibited in Cap1^{+/-} BMDMs ($P = 0.554$) (Fig. 5d).

Figure 5d. Oil Red O staining of BMDMs from *Cap1*^{+/+} and *Cap1*^{+/-} mice after differentiation into macrophages, followed by treatment with ox-LDL for 0, 24, 48, and 72 h. Ox-LDL treatment induced lipid formation and accumulation in *Cap1*^{+/+} BMDMs (upper panel) because of ox-LDL uptake, which was significantly inhibited in CAP1-deficient BMDMs. The scale bar represents 20 μ m (N=3).

Changes in the Manuscript Page 10, line 247, Figure 9a and 9b

To elucidate the underlying mechanisms, we performed analyses on ligated arteries exposed to D-flow. Adv-PCSK9 injection led to a significant increase in PCSK9 levels in the neointima, with infiltrated macrophages showing a significant colocalization with PCSK9 (P<0.001, P<0.001, respectively). In contrast, in Cap1^{+/-} mice, both PCSK9 levels and infiltrated macrophages were markedly reduced (P<0.001, P<0.001, respectively) (Fig. 9a). Furthermore, elevated serum levels of PCSK9 led to a significant increase in Syk and PKC δ phosphorylation in vivo (P=0.03, P<0.001, respectively), both of which were markedly reduced in Cap1^{+/-} mice (P=0.007, P=0.006, respectively) (Fig. 9b).

Figure 9a. Immunofluorescence staining for PCSK9 (red) and F4/80 (green) in arteries under D-flow in *Cap1*^{+/+} and *Cap1*^{+/-} mice injected with AdV-CTRL or AdV-PCSK9. PCSK9 (red) and F4/80 (green) expression significantly increased in *Cap1*^{+/+} mice with AdV-PCSK9 compared with that in AdV-CTRL. However, in *Cap1*^{+/-} mice, the increase in PCSK9 was less significant and F4/80 showed no significant change (N=3). The scale bar represents 10 μ m.

Figure 9b. Partially ligated *Cap1*^{+/+} and *Cap1*^{+/-} mice with only arteries exposed to D-flow were compared after AdV-CTRL or AdV-PCSK9 injection. Immunofluorescence staining of PCSK9 (red, upper panels), p-Syk (gray, middle panels), and p-PKCδ (green, bottom panels). AdV-PCSK9 injection increased PCSK9 expression in *Cap1*^{+/+} mice, which was lesser than that in *Cap1*^{+/-} mice. Additionally, p-Syk and p-PKCδ increased more significantly in *Cap1*^{+/+} mice than in *Cap1*^{+/-} mice (N=3). The scale bar represents 10 μm.

[**Comment #3-10**]: Are CAP1, SYK, and PKCδ regulated in macrophages isolated from patients before and after treatment with PCSK9 inhibitors?

[Response #3-10]: Thank you for your comment. Unfortunately, it was impossible to compare CAP1, SYK, and PKC δ before and after treatment with PCSK9 inhibitors as patients using evolocumab were already under medication. It was impossible to recruit healthy volunteers and obtain blood cells before and after several injections of evolocumab. Therefore, we observed the correlation of serum PCSK9 concentrations with SYK, PKC δ , and p65 phosphorylation in human PBMCs (Fig. 10a). In addition, in human PBMC-derived macrophages, we observed that CAP1-hFc attenuated phosphorylation of SYK, PKC δ , and p65 more effectively and significantly than evolocumab (Fig. 10e).

Changes in the Manuscript Page 11, line 272, Figure 10a

To investigate the potential association between human serum PCSK9 levels and its effect on inflammatory signaling pathways, blood samples were collected from individuals diagnosed with CAD and healthy donors. CAD patients were all under strict statin treatment. Pearson's correlation and simple linear regression analyses were used to examine the relationship between PCSK9 concentration in the serum and phosphorylation of Syk, PKC δ , p65(S276), and p65(S536) in PBMCs. Interestingly, serum PCSK9 concentration showed a positive correlation with the quantified phosphorylation of Syk ($P=0.03$, $R=0.498$), PKC δ ($P=0.049$, $R=0.458$), p65 (S276, $P=0.168$, $R=0.33$), and p65 (S536, $P=0.022$, $R=0.52$) in matched PBMCs (Fig. 10a and Supplementary Fig. 3). In Figure 10a, the data from CAD patients (indicated by black dots) exhibited a predominant distribution on the upper-right side of the slope, because of their higher PCSK9 concentration (409.4 ng/ml) and higher phosphorylation of signaling proteins compared with healthy donors whose distribution was on the lower-left side (indicated by pink dots, 290.1 ng/ml).

Fig. 10a. Simple linear regression analysis of Pearson correlation by distance. This figure illustrates the relationship between serum PCSK9 concentration and the quantified levels of phosphorylated proteins in a single individual. Serum PCSK9 concentration displayed a positive correlation with the phosphorylation of Syk, PKC δ , p65(S276), and p65(S536) in human peripheral blood mononuclear cells (PBMCs). The black line represents an asymptotic regression line fitted to the raw data (N=19).

Changes in the Manuscript Page 12, line 302, Figure 10e

PCSK9 significantly induced the phosphorylation of Syk, PKC δ , and p65 (S276 and S536) ($P < 0.001$, $P < 0.001$, $P < 0.001$, and $P = 0.002$, respectively) in human PBMC-derived macrophages, which was more effectively and significantly attenuated by CAPI-hFc ($P < 0.001$, $P = 0.002$, $P < 0.001$, and $P = 0.013$, respectively) than by evolocumab ($P = 0.003$, $P = 0.038$, $P = 0.002$, and $P = 0.03$, respectively) (Fig. 10e).

e

Fig. 10e. Immunoblot analysis demonstrated that PCSK9-induced phosphorylation of Syk, PKC δ , and NF- κ B p65 in human PBMC-derived macrophages was attenuated by CAP1-hFc. After 1-h treatment of rhPCSK9 with human IgG, evolocumab, or CAP1-hFc, the PCSK9-induced phosphorylation of Syk, PKC δ , and NF- κ B p65 was significantly decreased by CAP1-hFc (N \geq 7).

[Comment #3-11]: CAP1 protein is ubiquitously expressed in several cell types including ECs, SMCs, and immune cells in vascular tissues. The authors need to demonstrate how much effect monocyte-derived CAP1 exerts on the atherosclerotic plaque in vivo.

[Response #3-11]: We would like to express our gratitude for your valuable comment. We have previously reported that CAP1, the receptor for human resistin, leads to NF- κ B activation via the cAMP/PKA pathway (Lee, S. et al., *Cell Metab*, 2014, 19, 484–497). Additionally, the CAP1 protein is ubiquitously expressed, and its expression levels vary across different tissues (Jang, H.D. et al., *Eur Heart J*, 2020, 41, 239–252) (Reviewer only Fig. 3-4).

[FIGURE REDACTED]

Reviewer only Fig. 3-4. (a) Real-time PCR illustrating decreased mRNA levels of *CAP1* in each organ in *CAP1*^{+/-} mice (n=3 per group). (b) Immunoblot analysis demonstrating decreased protein levels of CAP1 in each organ in *CAP1*^{+/-} mice (Lee, S. et al., *Cell Metab*, 2014, 19, 484–497).

Interestingly, CAP1 mRNA expression was reported to be not only significantly increased in the CAD patients (Munjias, J. et al., *Eur J Clin Invest*, 2017, 47, 659–666), but also positively correlated with the carotid intima-media thickness in patients with end-stage renal disease (Munjias, J. et al., *Cardiorenal Med*, 2020, 10, 51–60).

In our current study, we examined the carotid artery of the *Ldlr*^{-/-} partial carotid ligation mouse model using CAP1 and PCSK9 antibodies for staining. Our results demonstrated an increase in CAP1 and PCSK9 expression in arteries under D-flow, which was further enhanced by AdV-PCSK9 treatment. Moreover, as atherosclerosis progressed, we observed an elevation in CAP1 levels (AdV-PCSK9, D-flow) (Fig. 3g). This suggests that CAP1 may play a crucial role in atherosclerosis induction, possibly through the PCSK9-CAP1 axis.

Furthermore, we investigated the colocalization of PCSK9 and CAP1 in the context of atherosclerosis in an *Ldlr*^{-/-} mouse model using partial carotid ligation. Immunostaining in arterial tissues revealed that the colocalization of PCSK9 and CAP1 increased in AdV-PCSK9-treated mice and further enhanced under D-flow conditions by partial ligation (Fig. 3g).

Fig. 3g. Immunofluorescence staining of CAP1 (green) and PCSK9 (red) in the arteries of AdV-CTRL or AdV-PCSK9-treated *Ldlr*^{-/-} mice under S-flow and D-flow. CAP1 and PCSK9 were colocalized in all groups. CAP1 and PCSK9 expression increased in the AdV-PCSK9 injection group compared with that in the AdV-CTRL group under D-flow. The scale bar represents 20 μm .

Next, to elucidate the cellular localization of CAP1 and PCSK9 within the atherosclerotic region, we conducted immunofluorescence staining of infiltrated macrophages, ECs, and SMCs that constitute atherosclerotic plaques. We observed that CAP1 and PCSK9 were expressed in atherosclerotic plaques, and PCSK9 colocalized with CAP1 in macrophages (Fig. 9c), ECs (Fig. 9d), and SMCs (Fig. 9e). Notably, SMCs within the atherosclerotic milieu exhibited characteristics akin to immune-like cells, consistent with previous research^{16,17}. PCSK9 expression was identified in both SMA⁺F4/80⁺ and SMA⁺F4/80⁻ cell populations, emphasizing its relevance across these cell types (Fig. 9f).

These observations suggest that PCSK9 along with CAP1 are critical factors orchestrating vascular inflammation, macrophage infiltration, and the progression of atherosclerosis. Furthermore, these observations serve to bolster the validity of our *in vitro* data, providing a robust foundation for our proposed mechanisms.

Fig. 9c and 9d. (c) Immunofluorescence images of arteries under D-flow from *Cap1*^{+/+} mice injected with AdV-PCSK9 showing colocalization of PCSK9 (red), CAP1 (gray), and F4/80 (green). The scale bar represents 5 μ m. (d) Immunofluorescence images of arteries under D-flow from *Cap1*^{+/+} mice injected with AdV-PCSK9 showing colocalization of PCSK9 (red), CAP1 (gray), and CD31 (green). The scale bar represents 5 μ m.

Fig. 9e and 9f. (e) Immunofluorescence images of arteries under D-flow from *Cap1*^{+/+} mice injected with AdV-PCSK9 showing colocalization of PCSK9 (red), CAP1 (gray), and αSMA (green). The scale bar represents 10 μm. (f) Immunofluorescence staining for PCSK9 (red), αSMA (green), and F4/80 (gray) in arteries under D-flow from *Cap1*^{+/+} mice injected with AdV-PCSK9. White arrow indicates PCSK9, αSMA, and F4/80 colocalization.

The yellow arrow denotes PCSK9 and α SMA colocalization (top middle panel). The three bottom panels show the colocalization of PCSK9 and α SMA (bottom left), PCSK9 and F4/80 (bottom middle), and α SMA and F4/80 (bottom right), respectively. The scale bar represents 10 μ m.

Changes in the Manuscript Page 10, line 256, Figure 9c, 9d, 9e and 9f

We observed that PCSK9 and CAP1 were expressed in atherosclerotic plaques and confirmed their colocalization on macrophages (Fig. 9c), ECs (Fig. 9d), and SMCs (Fig. 9e). Notably, SMCs within the atherosclerotic milieu exhibited characteristics akin to immunocyte-like cells, consistent with previous research^{16,17}. PCSK9 expression was identified in both SMA⁺F4/80⁺ and SMA⁺F4/80⁻ cell populations, emphasizing its relevance across these cell types (Fig. 9f).

[Comment #3-12]: Figure 1A and 4A: Total p65 was not quantified.

[Response #3-12]: Thank you for your suggestion. We added the total for p65 and revised the figures according to the reviewer's comment.

Changes in the Manuscript Figure 1a and 4a

Fig. 1a Original data

Fig. 1a Revised data

Fig. 1a. Immunoblot analysis demonstrated that proprotein convertase subtilisin/kexin type-9 (PCSK9) activated and phosphorylated nuclear factor (NF)- κ B p65 in THP-1 and human umbilical vein endothelial cells (HUVECs) in a dose-dependent manner (0, 50, 200, and 2,000 ng/ml) ($N \geq 3$).

Fig. 4a Original data

Fig. 4a Revised data

Figure 4a. Immunoblot demonstrating that rhPCSK9 treatment (2 µg/ml for 20 min) resulted in NF-κB p65 phosphorylation, which was blocked in CAP1-deficient THP-1 cells, but not in TLR4-deficient THP-1 cells (N=3).

[Comment #3-13]: Figure 2: This reviewer wonders why the authors chose a short time point (2 weeks) of HFD before carotid ligation.

[Response #3-13]: We appreciate the reviewer's comment and would like to provide a rationale for our choice of a short time point (2 weeks) of the high-fat diet (HFD) before carotid ligation in our study.

In a recent study by Kumar, S. et al. (*Lab Invest*, 2017, 97, 935–945), the authors presented a partial carotid ligation model using AAV-PCSK9 and HFD composition of 16% fat and 1.25% cholesterol (Research Diets, cat. #D12336, New Brunswick, NJ, USA). Briefly, they administered HFD and AAV-PCSK9. After 1 week, partial carotid ligation was performed, and the model was then observed for a period of 21 days.

In our study, instead of cholesterol, we used an HFD composition of 60% fat (Research Diets). To compensate for the absence of cholesterol, we extended the duration of the diet to 2

weeks or 3 weeks before carotid ligation. We observed that, even a 2-week HFD was enough to observe the lesion area in the RCA. Furthermore, we found that there was no significant difference in the lesion area between the group on the 2-week diet and the group on the 3-week diet (Reviewer only Fig. 3-5). We believe that this time frame adequately captured the desired effects of an HFD in our carotid ligation model and allowed us to evaluate the specific outcomes of interest in a timely manner.

Reviewer only Fig. 3-5. (a) Experimental scheme of partial carotid ligation. (b) Representative gross images and hematoxylin and eosin (H&E) staining of the 14-day high-fat diet (HFD) group. (c) Representative gross images of the 23-day HFD group. Lesion area quantification was performed using ImageJ software (National Institutes of Health, Bethesda, MD, USA).

[Comment #3-14]: Figure 2: It would be interesting to see the changes in the expression of adhesion and inflammatory markers in liver tissue, carotid artery and BMDMs of AAV-PCSK9 injected LDLR^{-/-} mice to have consistency in findings from in vitro experiments (Figure 1).

[Response #3-14]: We appreciate your comment. In response, we have conducted additional analyses of adhesion and inflammatory markers in different tissues of AdV-PCSK9-injected *Ldlr*^{-/-} mice. We examined the expression of inflammatory cytokines, including TNF- α , IL-1 β , and IL-6, in *Ldlr*^{-/-} mice arteries and liver tissues. In *Ldlr*^{-/-} mice arteries, we observed a significant increase in inflammatory cytokines after injection of AdV-PCSK9 (Fig. 2h). Similarly, in the *Ldlr*^{-/-} mice livers, we observed a significant increase in the mRNA levels of TNF- α and IL-1 β after injection of AdV-PCSK9, whereas IL-6 did not show a significant difference, which is consistent with our *in vitro* findings (Reviewer only Fig. 3-6). Moreover, albumin, a potential indicator of liver function damage, was found to be decreased in AdV-PCSK9-injected mice livers (Reviewer only Fig. 3-6).

Fig. 2h

Figure 2h. qPCR analysis of the same samples of carotid arteries from the *Ldlr*^{-/-} mice showing that the expression of inflammatory cytokines (TNF- α , IL-1 β , and IL-6) was significantly higher in AdV-PCSK9 than in AdV-CTRL.

Reviewer only Fig. 3-6. qPCR analysis demonstrating that AAV-PCSK9 significantly increased inflammatory cytokines (TNF- α , IL-1 β , and IL-6) and decreased albumin (ALB) expression in *Ldlr*^{-/-} mice livers.

Furthermore, in the liver, AdV-PCSK9 administration led to the upregulation of PCSK9 as well as adhesion molecules including integrin- α 4, integrin- β 1, VCAM-1, and ICAM-1 (Supplementary Fig. 1a). These results are consistent with our *in vitro* findings (Fig. 1), suggesting consistent effects of AdV-PCSK9 treatment on adhesion and inflammatory markers in different tissue environments.

However, *Ldlr*^{-/-} BMDMs did not exhibit significant PCSK9 overexpression upon AdV-PCSK9 treatment, and the protein expression of integrin- α 4 and integrin- β 1 did not increase significantly (Supplementary Fig. 1b).

Overall, additional analyses in different tissues further support the consistency of our findings, both *in vitro* and *in vivo*. We hope that these results contribute to a more comprehensive understanding of the effects of AdV-PCSK9 treatment on adhesion and inflammatory markers in *Ldlr*^{-/-} mice.

Supplementary Fig. 1. Expression of adhesion molecules in the animal experiment showing that PCSK9 aggravated atherosclerosis in *Ldlr*^{-/-} mice (Figure 2). AdV-PCSK9 at 1×10¹¹ infectious units/mouse was administered via the tail vein to mice on a high-fat diet. (a) Immunoblot analysis of the liver samples from *Ldlr*^{-/-} mice showing that the expression of adhesion molecules (integrin-α4, integrin-β1, VCAM-1, and ICAM-1) along with PCSK9 was significantly higher in AdV-PCSK9 mice than in AdV-CTRL mice. (b) Immunoblot analysis of BMDMs from *Ldlr*^{-/-} mice showed no significant changes in the expression of adhesion molecules (integrin-α4, integrin-β1).

Changes in the Manuscript Page 5, line 114, Fig. 2h and Supplementary Fig. 1

*The expression of inflammatory cytokines (TNF-α, IL-1β, and IL-6) in the carotid artery was significantly higher in AdV-PCSK9-injected mice than in AdV-CTRL-injected mice (Fig. 2h). These observations in the *Ldlr*^{-/-} mice suggested that PCSK9 induced NF-κB-mediated inflammation and atherosclerosis directly and not via LDLR.*

*We examined the main source of PCSK9 after systemic administration of AdV-PCSK9 in the *Ldlr*^{-/-} mice. As AdV mainly infects the liver¹¹, we observed that the liver was the main source*

of PCSK9 after AdV-PCSK9 administration, exhibiting overexpression of adhesion molecules (integrin- α 4, integrin- β 1, VCAM-1, and ICAM-1) compared with AdV-CTRL (Supplementary Fig. 1a). In contrast, bone marrow cells such as BMDMs were not effectively infected with AdV-PCSK9 after systemic administration (Supplementary Fig. 1b).

[Comment #3-15]: Figure 2E: Phosphorylation is a transient process and stimulation of 10-30 min is standard for measuring phosphorylation of signalling proteins. This reviewer wonders why the authors chose 24h rhPCSK9 stimulation for measuring phosphorylation and whether time-course experiments were conducted. In addition, the quantification data is missing for this experiment.

[Response #3-15]: We sincerely appreciate your valuable feedback and insightful suggestions, which have significantly improved our study. By adjusting the rhPCSK9 treatment time to 40 min, we were able to achieve more effective and informative results.

Initially, a 24-h treatment of PCSK9 was conducted to observe the degradation effect of LDLR. However, after careful consideration of the importance of emphasizing signal reduction in LDLR knockout, we concur that utilizing a shorter time point of 40 min is indeed more appropriate and informative for this figure. Consequently, we have made the necessary revisions to the data, and the updated figure is presented below: Fig. 2e is changed to Fig. 1h.

Changes in the Manuscript Fig. 1h

Fig. 2e Original data

Fig. 1h Revised data

Figure 1h. Immunoblot analysis demonstrating that PCSK9 treatment (2 µg/ml) for 40 min activated and phosphorylated NF-κB in BMDMs from *Ldlr*^{-/-} mice and BL6 control mice (N≥3).s

[Comment #3-16]: Figure 3F: From immunoblot, it seems CAP1 siRNA was not efficient. The authors may want to present the mRNA levels of CAP1 after siRNA.

[Response #3-16]: Thank you for your comment. To maximize the knockdown efficiency of siRNA in THP-1 cells, we re-ordered CAP1 siRNA with the same sequence as Lee et al. and Jang et al., extended THP-1 cell starvation to overnight, and evaluated the extent of CAP1 protein knockdown in a time-dependent manner. We treated overnight starved THP-1 cells in basal RPMI with CAP1 siRNA and Lipofectamine RNAiMAX for 6 h and changed the media from starvation (0% FBS) to growth RPMI (10% FBS). After 24, 48, and 72 h of siRNA treatment, cells were harvested with RIPA buffer and western blotting was performed to compare CAP1 protein expression. We found that the optimum time required for expression of CAP1 to be knocked down sufficiently was 72 h and we provided this information as Supplementary Fig. 4.

Changes in the Manuscript Supplementary Fig. 4 and Fig. 3f

Supplementary Fig. 4. Immunoblot analysis demonstrating the effects of time elapsed after siRNA transfection on the degree of CAP1 or TLR4 knockdown (top). THP-1 cells were transfected with CAP1 or TLR4 siRNA and harvested at different time points: 0, 24, 48, and 72 h. The representative figures of three independent experiments are shown and quantified using ImageJ software (bottom).

Therefore, we carried out all the CAP1 knockout experiments of THP-1 cells using this protocol. As a result, CAP1 protein expression was significantly attenuated, as demonstrated in Fig. 3f.

Fig. 3f Original data

Fig. 3f Revised data

Fig. 3f. Duolink Proximity Ligation Assay for detecting the interaction between PCSK9 and TLR4 with CTRL or CAP1 siRNA. The extent of the interaction between PCSK9 and TLR4 was quantified by counting the red dots (N=3). The scale bar represents 5 μ m.

[Comment #3-17]: HUVEC-THP-1 adhesion assay (Figure 4F): Details regarding the assay and analysis is not clear.

[Response #3-17]: Regarding the HUVEC-THP-1 adhesion assay (Fig. 4f), we would like to provide a more comprehensive explanation of our methodology.

First, we individually subjected THP-1 cells to knockdown using either CTRL siRNA or CAP1 siRNA for a period of 72 h. Subsequently, THP-1 cells were pre-treated with PCSK9 for 6 h. To visualize the cells, we labeled them with FITC using a PKH67 green fluorescent cell linker kit (Versatile PKH67 Green Fluorescent Membrane Kit [PKH67GL], Sigma-Aldrich, St. Louis, MO, USA).

Next, we prepared a monolayer of HUVECs and cocultured them with the FITC-labeled THP-1 cells for 1 h, both in the presence and absence of rhPCSK9. To ensure the removal of non-adherent cells, we performed two washes with cell culture media. Subsequently, we utilized a fluorescence microscope to capture images of the THP-1 cells that adhered to the

HUVEC monolayer. Our analysis involved quantifying the number of fluorescence-positive cells to derive meaningful results.

In response to your feedback, we have included a schematic representation of the experimental procedure in Fig. 4h and have provided a more detailed description of the methodology in the Methods section. Thank you for bringing this to our attention, and we believe these additions will enhance the clarity and transparency of our work.

Changes in the Manuscript Page 20, line 527, Figure 4h and Method

Cell adhesion assay

THP-1 cells were individually subjected to knockdown procedures utilizing either CTRL siRNA or CAPI siRNA, and the cells were allowed to incubate for a period of 72 hours to ensure effective knockdown. Subsequently, THP-1 cells underwent a 6-hour pre-treatment with rhPCSK9. For cellular visualization, FITC labeling was achieved using a PKH67 green fluorescent cell linker kit (Sigma-Aldrich, PKH67GL). Then, a monolayer of HUVEC cells was prepared for co-culture. FITC-labeled THP-1 cells were cocultured with the HUVEC cells for 1 hour, both in the presence and absence of rhPCSK9. To remove non-adherent cells, two successive washes were conducted using cell culture media. Following this, we used a fluorescence microscope to capture detailed images of the THP-1 cells adhering to the HUVEC monolayer. Fluorescence-positive cells were quantified by counting using the ImageJ software (National Institutes of Health). Differences between means were analyzed using t-test with the Prism 6 software (GraphPad Software).

Fig. 4h. Cell adhesion assay of THP-1 and HUVECs with rhPCSK9 treatment demonstrating that adhesion to HUVECs was enhanced by PCSK9 in THP-1 cells with CTRL siRNA, which was blocked in CAP1-deficient THP-1 cells. Representative images of fluorescently labeled adherent THP-1 cells (upper-left panel), and fluorescence-positive cells were counted to quantify cell adhesion (upper-right panel) (N=4). The schematic figure illustrating the adhesion assay (bottom panel). The differences between the groups were compared using the unpaired *t*-test.

[Comment #3-18]: Figure 5: The experimental design raised a concern. The authors used siRNA of CAP1 in THP-1 cells where LDLR was present.

[Response #3-18]: In Fig. 5a, we showed that rhPCSK9 treatment enhanced ox-LDL uptake in THP-1 cells, which was reduced in CAP1-deficient monocytes. We previously reported (Jang, H.D. et al., *Eur Heart J*, 2020, 41, 239–252) that LDLR expression increases in CAP1-knocked down HepG2 cells and the livers of *Cap1*^{+/-} mice. Therefore, we speculated that LDLR will be present and its protein expression will further be increased in CAP1-deficient monocytes.

During the development of atherosclerosis, LDL undergoes oxidation to form ox-LDL, leading to foam cell formation. Importantly, the receptor of ox-LDL changes from LDLR to SR-A1, CD36, LOX-1, and TLR4 (Reviewer only Fig. 3-7).

[FIGURE REDACTED]

Reviewer only Fig. 3-7. Difference between oxidized LDL (ox-LDL) and minimally modified LDL (MM-LDL). LDL is thought to be modified in a stepwise manner during the generation of ox-LDL. In the initial phase of modification, the lipid components (light blue circles) react with oxidation reagents, resulting in radical chain reactions that produce many types of lipid oxidation products (red, brown, yellow, or dark blue circle). Then, the lipid oxidation products react with the apolipoprotein B (apoB) protein (solid dark blue line) to generate adducts and cross-links. Radicals can attack the apoB protein directly, resulting in oxidative changes in amino acid side chains and cleavage of the peptide bonds (orange/gray line). MM-LDL may contain lipid oxidation products without extensive protein modification because it binds to LDL receptor rather than scavenger receptors. As modification on the apoB protein proceeds, its mobility in the agarose gel changes greatly, and it loses the affinity to the LDL receptor, and, in turn, becomes a ligand of scavenger receptors (Itabe, H. et al., *J Lipids*, 2011, 2011, 418313).

We hypothesized that the expression of scavenger receptors is attenuated under CAP1-deficient conditions as CAP1 siRNA-treated monocytes exhibited a reduction in ox-LDL uptake. As a result, CAP1-knocked down THP-1 cells showed decreased mRNA expression of *TLR4*, *LOX1*, *CD36*, and *SRA1* in Fig. 5b, which supports our hypothesis.

Fig. 5

Fig. 5a and 5b. (a) For the ox-LDL assay, THP-1 cells were transfected with CTRL or CAP1 siRNA and then treated with or without rhPCSK9. ox-LDL uptake decreased in CAP1-deficient THP-1 cells in response to PCSK9 (2 µg/ml). (b) qPCR analysis demonstrating the mRNA levels of several scavenger receptors in response to PCSK9 treatment in THP-1 cells transfected with CTRL or CAP1 siRNA. The mRNA levels of *LOX1*, *CD36*, *SRA1*, and *TLR4* increased significantly with rhPCSK9 (2 µg/ml) treatment in monocytes with CAP1, but not in THP-1 cells with CAP1 knockdown (N=4).

In response to the reviewer's valuable suggestion, we conducted additional experiments outlined as follows: we treated BMDMs from *Cap1*^{+/+} and *Cap1*^{+/-} mice with ox-LDL for 24, 48, and 72 h and stained the tissues with Oil Red O to observe lipid accumulation. As a result,

lipid accumulation significantly increased in *Cap1*^{+/+} BMDMs in a time-dependent manner, whereas it was significantly attenuated in *Cap1*^{+/-} BMDMs (Fig. 5d).

Fig. 5d. Oil Red O staining of BMDMs from *Cap1*^{+/+} and *Cap1*^{+/-} mice after differentiation into macrophages, after treatment with ox-LDL for 0, 24, 48, and 72 h. Ox-LDL treatment induced lipid formation and accumulation in *Cap1*^{+/+} BMDMs (upper panel) because of ox-LDL uptake, which was significantly inhibited in CAP1-deficient BMDMs. The scale bar represents 20 μ m (N=3).

Changes in the Manuscript Page 7, line 168, Figure 5d

Bone marrow was extracted from WT and CAP1 heterozygous knockout mice (Cap1^{+/-} mice) and the cells were differentiated into BMDMs using monocyte-colony stimulating factor (M-CSF; 50 ng/ml). After ox-LDL treatment (20 μ g/ml), lipid aggregation and transformation into

lipid-laden macrophages increased significantly in a time-dependent manner in Cap1^{+/+} BMDMs (P<0.001), whereas lipid aggregation and transformation were inhibited in Cap1^{+/-} BMDMs (P=0.554) (Fig. 5d).

[Comment #3-19]: Figure 6: The authors conducted elegant experiments to show the downstream molecules involved after PCSK9-CAP1 binding by pulldown followed by MS/MS. However, better explanations of the selection of SYK/PKC δ (out of 2103 binding proteins to CAP1) as the signalling pathway need to be provided.

[Response #3-19]: In our endeavors to elucidate the signaling pathways resulting from PCSK9-CAP1 binding, we employed a comprehensive approach involving pulldown followed by MS/MS analysis. To address the selection of SYK and PKC δ as key components of the signaling pathway, we employed the following process:

Initially, CAP1-binding proteins were identified using the SAINT algorithm (Fig. 6a). Among the vast array of 2,103 proteins pulled down by CAP1-mFc, we performed sorting and analysis of 464 proteins exhibiting SAINT AvgP>0.6. Further analysis was conducted using the Database for Annotation, Visualization, and Integrated Discovery (DAVID) with GO enrichment analysis, employing an EASE score threshold of <0.1. A threshold of EASE score <0.1 is conventionally employed within the DAVID framework to denote statistical significance. The outcomes obtained with this analysis are deemed to bear significance in accordance with scientific standards.

Following GO enrichment analysis, three categories (biological process, cellular component, and molecular function) were detected, and the enriched terms (level 1 of GO terms) in each category are shown in Figure 6b. We then conducted reanalysis using the specific gene list.

Our specific focus in the analysis was on the 'Immune system process (GO.0002376)' since CAP1 is known to function as a receptor in inflammatory signaling pathways. Notably, the GO terms associated with the 'immune system process' were ranked high and exhibited statistical significance in the enriched list (Fig. 6b).

Changes in the Manuscript Figure 6b

Fig. 6b Revised data

Fig. 6b. Database for Annotation, Visualization, and Integrated Discovery (DAVID) functional enrichment analysis was performed on differentially bound proteins. The graph displays the top-ranked significantly enriched Gene Ontology (GO) terms in biological process, cellular component, and molecular function. All the adjusted statistically significant values of the terms were transformed to their negative 10-base logarithm. The analysis was conducted using an EASE score threshold of <0.1.

With a meticulous approach, we ranked the SAINT AvgP values and identified a list of nine proteins with AvgP=1 within this category. Notably, among these, SYK and PKC δ emerged as particularly promising candidates because of their kinase activities (Fig. 6c).

We acknowledge the reviewer's valuable feedback, and with this expanded explanation, we aim to provide a clearer understanding of our rationale for selecting SYK and PKC δ as the pivotal signaling pathway components.

Changes in the Manuscript Page 8, line 202, Figure 6c

We focused on the 'Immune system process (GO.0002376)' category and ranked the SAINT AvgP in descending order, obtaining a list of nine proteins with AvgP=1. Among these, Syk and PKC δ were selected as potential binding candidates for CAPI because of their kinase activities (Fig. 6c).

Fig. 6c Revised data

[Immune system process (GO.0002376)				
PreyGene	Protein name	Avg Intensity	AvgP	FoldChange
SYK	spleen associated tyrosine kinase(SYK)	1.072	1	3.651
PRKCD	protein kinase C delta(PRKCD)	0.722	1	3.804
PSME3	proteasome activator subunit 3(PSME3)	1.688	1	4.204
RPS3	ribosomal protein S3(RPS3)	7.327	1	3.62
RAB32	RAB32, member RAS oncogene family(RAB32)	1.176	1	4.28
ILF3	interleukin enhancer binding factor 3(ILF3)	2.209	1	4.568
RPL13A	ribosomal protein L 13a(RPL13A)	2.483	1	4.498
DHX15	DEAH-box helicase 15(DHX15)	0.876	1	4.64
EXOSC6	exosome component 6(EXOSC6)	1.518	1	4.297

Fig. 6c. List of nine proteins that bound to CAPI with the most significant values of SAINT AvgP=1 in the immune system process (GO.0002376).

[Comment #3-20]: Pulldown and MS/MS assay are particularly interesting. Did the authors reiterate that the pulldown of 2103 proteins is the result of PCSK9-CAP1 interaction or these are just the binding partners of CAP1 in its native form?

[Response #3-20]: In our study, we employed a pulldown approach, utilizing proteins originating from human CAP1, wherein the carboxyl-terminal end was coupled with the Fc region of mouse immunoglobulin (referred to as CAP1-mFc).

It is worth noting that PCSK9 is predominantly expressed across diverse human tissues, including the liver, intestine, kidney, and central nervous system (Norata, G.D. et al., *Annu Rev Pharmacol Toxicol*, 2014, 54, 273–293) and actively participates in the proteolytic maturation of proteins intended for secretion (Seidah, N.G. et al., *Methods Mol Biol*, 2011, 23–57).

In our investigation, we uncovered the interaction of extracellular PCSK9 with cell membrane-associated CAP1, resulting in the subsequent internalization and activation of downstream signaling pathways. Acknowledging the intricacies of this interplay, our research was fundamentally focused on comprehending the binding partners of CAP1, rather than exclusively focusing on binding partners of CAP1 that arise solely from PCSK9-CAP1 interactions. We extend our gratitude for your valuable feedback, and we believe that this elaboration will foster an enhanced understanding of the guiding perspective behind our research.

[Comment #3-21]: Figure 8F: The overall mechanism presented in the schematic is too complex with several levels of signalling mechanism involved and little experimental evidence provided in the study.

[Response #3-21]: In this study, we found that PCSK9 has a direct pro-inflammatory action independently of LDLR. PCSK9 triggered the activation of NF- κ B and secretion of pro-inflammatory cytokines (TNF- α , IL-1 β , and IL-6) from monocytes as well as the expression of adhesion molecules and scavenger receptors on monocytes and ECs.

[1] Firstly, PCSK9 binds to CAP1, which results in the phosphorylation of Syk and PKC δ (Fig. 7). Phosphorylated Syk directly activates PI3K (Chen, F. et al., *Cancer Cell*, 2013, 23, 826–838), whereas PKC δ initiates the production of cyclic AMP (cAMP), subsequently leading to the activation of PI3K (Verónica, G.M. et al., *Biochem Pharmacol*, 2017, 145, 94–101) by the activated protein kinase A (PKA) (Fig. 7c). Next, PI3K phosphorylates AKT, which subsequently activates NF- κ B. Such signaling pathways were similarly observed in our previous study (Lee, S. et al., *Cell Metab*, 2014) where resistin, which has a similar structure to PCSK9, was used to induce inflammation through the CAP1 receptor via cAMP/PKA/NF- κ B axis.

[2] Secondly, PI3K promotes the clustering of VLA-4, which enhances its binding capacity. Although the impact of PI3K is not direct, it assumes a role via the PI3K \rightarrow PIP3 \rightarrow PKC β \rightarrow Talin-1 phosphorylation pathway (Polcik, L. et al., *Cells*, 2022, 11, 2235) to attach to the NPKY motif, prompting a change in shape that results in VLA-4 activation (Gahmberg, C.G. et al., *Trends Biochemical Sci*, 2022, 47, 265–278). Following VLA-4 activation, a sequence unfolds where it initiates PI3K-PIP3-AKT activation (Polcik, L. et al., *Cells*, 2022, 11, 2235).

[3] Thirdly, PCSK9-mediated AKT activates SREBPs through several mechanisms, one of which involves phosphorylation of SREBP cleavage-activating protein (SCAP), which is a key regulator of SREBP activation (Fig. 5g). When SREBPs are activated and translocate to the nucleus, they bind to the SRE motifs in the promoter region of the *PCSK9* gene. This binding

leads to the upregulation of *PCSK9* transcription (Figure 3e, 3f). Once synthesized, PCSK9 is transported to the endoplasmic reticulum and then secreted into the bloodstream.

[4] Lastly, PCSK9-mediated AKT activates NF- κ B, which leads to the transcriptional activation of scavenger receptor genes, increasing synthesis of scavenger receptors on the cell surface such as monocytes (Fig. 5). These receptors are then able to bind and internalize extracellular molecules, contributing to processes such as lipid uptake and immune response. Among the target genes regulated by NF- κ B are those encoding various pro-inflammatory cytokines, such as TNF- α , IL-1 β , IL-6, and IL-8, among others (Fig. 1). The transcriptional activation of these genes leads to the synthesis of the respective cytokines.

Changes in the Manuscript Page 16, line 418, Figure 9g

As summarized in Fig. 9g, PCSK9 binds to CAPI, leading to the activation of Syk and PKC δ and induction of inflammatory gene cascades, TLR4, and scavenger receptors on monomacrophages and adhesion molecules on ECs. These functions of PCSK9 were dependent on CAPI and thus attenuated by CAPI depletion. Systemic injection of AdV-PCSK9 significantly induced atherosclerosis of the carotid artery exposed to D-flow in vivo, which was prevented in Cap^{+/-} mice. Finally, the human serum level of PCSK9 correlated well with the degree of phosphorylation of these signaling proteins in PBMCs. These results suggest that the PCSK9-CAPI-Syk/PKC δ pathway may be a viable target for developing new therapeutics for dyslipidemia, atherosclerotic cardiovascular, and inflammation-based diseases.

g

PCSK9 signaling in monocytes

Fig. 9g. Schematic model showing PCSK9-mediated inflammation in monocytes mediated by CAP1 recruiting PKC δ and Syk and modulating PCSK9-mediated inflammatory signal transduction.

[Comment #3-22]: In qPCR data throughout the manuscript, a control group/condition has no error bar. This reviewer does not believe one can use all the values in control samples as 1 in statistics.

[Response #3-22]: We appreciate the reviewer's attention to the qPCR data in our manuscript. We performed at least three independent experiments and quantified every datum relative to each control group to produce quantified bar graphs. This approach allowed us to establish a standardized baseline where the control for each individual experiment was set to 1.

[Comment #3-23]: Molecular weight markers are missing in immunoblots throughout the manuscript.

[Response #3-23]: We appreciate the reviewer's observation regarding the absence of molecular weight markers in our immunoblot figures. In response to your suggestion, we have now included molecular weight markers in all immunoblot figures, following the example of Fig. 1a.

Fig. 1a. Immunoblot analysis demonstrated that proprotein convertase subtilisin/kexin type-9 (PCSK9) activated and phosphorylated nuclear factor (NF)- κ B p65 in THP-1 and human umbilical vein endothelial cells (HUVECs) in a dose-dependent manner (0, 50, 200, and 2,000 ng/ml) ($N \geq 3$).

[Comment #3-24]: Abstract did not summarize the overall study well.

[Response #3-24]: We have summarized the overall study end of the Abstract and accordingly revised the statements as shown below.

Changes in the Manuscript Page 2, line 30, Abstract

We have rewritten the abstract as follows:

The direct inflammatory action of proprotein convertase subtilisin/kexin type 9 (PCSK9) was examined in vitro in monocytes and endothelial cells and an in vivo atherosclerosis animal model. Systemic administration of adenovirus (AdV)-PCSK9 aggravated atherosclerosis in low-density lipoprotein receptor (LDLR)^{-/-} mice, implying inflammation was caused by PCSK9 independently from LDLR. Adenylyl cyclase-associated protein 1 (CAP1) was the main binding partner of PCSK9 and indispensable for the inflammatory action of PCSK9, including induction of cytokines, Toll like receptor 4 (TLR4), and scavenger receptors, enhancing the uptake of oxidized LDL. We found spleen tyrosine kinase (Syk) and protein kinase C delta (PKCδ) to be the key mediators of inflammation after PCSK9-CAP1 binding. Cap1^{+/-} mice were protected from AdV-PCSK9-induced atherosclerosis. Serum PCSK9 concentration positively correlated with Syk, PKCδ, and p65 phosphorylation in peripheral blood mononuclear cells (PBMCs) from patients with coronary atherosclerosis. The CAP1-fragment crystallizable region (CAP1-Fc) more effectively attenuated PCSK9-mediated Syk, PKCδ, and p65 phosphorylation in PBMCs than the PCSK9 inhibitor, evolocumab.

[Comment #3-25]: Introduction: Lines 45-48 need citations.

[Response #3-25]: Thank you for your comment. As per your suggestion, we have incorporated the following reference into the manuscript:

Reference 6. Walley, K.R., et al. PCSK9 is a critical regulator of the innate immune response and septic shock outcome. *Sci. Transl. Med.* **6**, 258ra143 (2014).

Changes in the Manuscript Page 3, line 51, Introduction

We have rewritten this text as follows:

Pcsk9-knockout mice display a decreased inflammatory response to lipopolysaccharide (LPS) and pharmacological inhibition of PCSK9 improves survival and inflammation in murine polymicrobial peritonitis⁶. Furthermore, Pcsk9 loss-of-function genetic variants in septic shock patients are associated with improved survival, whereas gain-of-function mutants show decreased survivability⁶.

[Comment #3-26]: Statistical details are missing in the figure legends.

[Response #3-26]: We have filled in all the statistical details in the figure legends used in this paper. All experiments were repeated at least three times and comparisons between groups were made using unpaired t-test or one-way analysis of variance (ANOVA) (Fig. 1a, 1b, 1c, 1d, 1e, 1f, 1h, 1i, Fig. 2b, 2g, 2h, Fig. 3a, 3e, 3f, Fig. 4a, 4b, 4c, 4d, 4e, 4g, 4h, Fig. 5b, 5c, 5d, 5e, 5f, 5g, Fig. 7a, 7b, 7c, 7d, 7e, 7f, Fig. 8b, 8c, 8d, Fig. 9a, 9b, Fig. 10a, 10b, 10c, 10d, 10e and Supplementary Fig. 4).

REVIEWERS' COMMENTS

Reviewer #1 (Remarks to the Author):

The authors provided a careful and serious response to review's comments. They also supplemented the experimental data as required. Now, the article can be considered for acceptance.

Reviewer #2 (Remarks to the Author):

The authors have properly addressed all the raised questions and remarks. As already stated, this study gives completely new mechanistic insight by which PCSK9 contributes to atherosclerosis, beyond LDLR, and could have a major impact on future targeted therapy development for atherosclerosis and inflammation-related diseases. Therefore, this reviewer recommends this article for publication.

Reviewer #3 (Remarks to the Author):

The authors have addressed most concerns raised by this reviewer by performing additional experiments. There are, however, the following residual issues.

[#3-3]

The schematic diagram (Figure 10f) does not represent current findings. Evolocumab also statistically inhibits PCSK9-induced p-p65 expression.

[#3-4]

As demonstrated in this revision, there was endotoxin contamination in the present study. However, they showed that rhPCSK9 induced p-p65 expression was not decreased by TLR4 siRNA in Fig.4, suggesting the pro-inflammatory effects of PCSK9 itself rather than endotoxin contamination.

[#3-6]:

This question was not satisfactorily addressed. Evidence suggests that various cells in

vascular wall (e.g., macrophages, ECs, SMCs) express PCSK9. The data presented in Figure 9 are thus reasonable and expected. But, relative contributions of these cells to PCSK9 biology or expression levels in the vasculature remain controversial and remain unaddressed in the context of atherosclerosis in the present study. At least two published studies suggested that SMCs rather than macrophages or ECs, are the main source of PCSK9 in atherosclerotic lesion. The expression levels of PCSK9 in these cell types need to be further investigated.

[#3-7 & #3-19]

The authors' response to this reviewer's major concern regarding a lack of logical transition from unbiased screening and validation is not satisfactory. We understood how the authors focused on "immune system process" and downstream SYK, and PKC δ signalling. However, as this reviewer pointed out, "immune system process" was the lowest ranking GO Biological process. After the authors identified the enriched terms in an unbiased manner, "immune system process" was manually chosen since CAP1 was known to function as a receptor in inflammatory signaling pathways. This disconnection between unbiased analysis and targeted validation experiments remains a major weakness of the manuscript.

[#3-7]

Color legend in the revised Figure 6B is incorrect.

[#3-11]

In response to this comment authors provided a new set of co-localization data suggesting that PCSK9 and CAP1 are colocalization on macrophages (Fig. 9c), ECs (Fig. 9d), and SMCs (Fig. 9e). SMCs within the atherosclerotic milieu exhibited characteristics akin to immunocyte-like cells, consistent with previous research Ref16,17. PCSK9 expression was identified in both SMA+F4/80+ and SMA+F4/80- cell populations, emphasizing its relevance across these cell types (Fig. 9f). But this dilutes the effect of CAP-1-PCSK9 interaction on monocytes leading to the activation of cytokines in monocytes and the progression of atherosclerosis.

[#3-15]

To address this reviewer's concern regarding a very late time-point for phosphorylation data in Figure 2E: the authors have conducted a new set of experiments with a 40 min timepoint which makes more sense. The new data is still missing total p65 expression and the data quantification.

Comments from Reviewer #3 and Responses

Reviewer #3 (Remarks to the Author):

The authors have addressed most concerns raised by this reviewer by performing additional experiments. There are, however, the following residual issues.

[Response]: We thank the reviewer for their insightful comments and suggestions. A point-by-point response to all comments has been provided below.

[Comment #3-3]: The schematic diagram (Figure 10f) does not represent current findings. Evolocumab also statistically inhibits PCSK9-induced p-p65 expression.

[Response #3-3]: In Figure 10c, a competitive ELISA binding assay, the binding of PCSK9-His to CAP1 was competitively inhibited by CAP1-hFc, but evolocumab did not directly hinder the PCSK9-CAP1 interaction. On the other hand, in Figure 10d and 10e, a western blot analysis of inflammatory signaling in monocyte and human PBMC-derived macrophages revealed that evolocumab displays a weaker inhibition of PCSK9-mediated NF- κ B activation, in comparison to the stronger inhibition by CAP1-hFc. Consistently, the phosphorylation of Syk or PKC δ , was significantly inhibited by CAP1-Fc but only weakly affected by evolocumab.

Considering the overall data, it is evident that CAP1-hFc effectively disrupts the direct protein-protein interaction between PCSK9 and CAP1, whereas evolocumab does not hinder the direct interaction. However, in the presence of other proteins besides these two, we cannot conclusively rule out the possibility of evolocumab binding to PCSK9 and subsequently

impeding the interaction between PCSK9 and CAP1. In addition, we also speculate that when PCSK9 binds with the bulky evolocumab antibody, steric hindrance not only inhibits the binding of LDL-R but also weakly inhibits the binding of CAP1. Therefore, we revised the schematic diagram of evolocumab by illustrating some distance between PCSK9 and CAP1 and then weak inhibition on the ‘inflammatory signalling pathway’.

Changes in the Manuscript Figure 10f

1st Revised data

2nd Revised data

f

[Comment #3-4]: As demonstrated in this revision, there was endotoxin contamination in the present study. However, they showed that rhPCSK9 induced p-p65 expression was not decreased by TLR4 siRNA in Fig.4, suggesting the pro-inflammatory effects of PCSK9 itself rather than endotoxin contamination.

[Response #3-4]: Thank you for the insightful interpretation. It has provided valuable information on the endotoxin levels, which we believe will be beneficial for future research endeavors.

[Comment #3-6]: This question was not satisfactorily addressed. Evidence suggests that various cells in vascular wall (e.g., macrophages, ECs, SMCs) express PCSK9. The data presented in Figure 9 are thus reasonable and expected. But, relative contributions of these cells to PCSK9 biology or expression levels in the vasculature remain controversial and remain unaddressed in the context of atherosclerosis in the present study. At least two published studies suggested that SMCs rather than macrophages or ECs, are the main source of PCSK9 in atherosclerotic lesion. The expression levels of PCSK9 in these cell types need to be further investigated.

[Response #3-6]: Through this revision process, we have gained a deeper understanding of the expression of PCSK9 not only in inflammatory cells but also in SMCs and ECs. We are grateful for your suggestion, which has prompted us to consider this matter more thoroughly. As you've proposed, we plan to conduct further research into the roles of PCSK9 in SMCs and ECs beyond monocytes in the progression of atherosclerosis.

[Comment #3-7 & #3-19]: The authors' response to this reviewer's major concern regarding a lack of logical transition from unbiased screening and validation is not satisfactory. We understood how the authors focused on "immune system process" and downstream SYK, and PKC δ signalling. However, as this reviewer pointed out, "immune system process" was the lowest ranking GO Biological process. After the authors identified the enriched terms in an

unbiased manner, “immune system process” was manually chosen since CAP1 was known to function as a receptor in inflammatory signaling pathways. This disconnection between unbiased analysis and targeted validation experiments remains a major weakness of the manuscript.

[Response #3-7 & #3-19]: In response to the reviewer's concern regarding the logical transition from unbiased screening to validation, we would like to emphasize our group's previous work. Previously, we found out that PCSK9, whose CRD domain is structurally similar to the resistin trimer, leads to the LDLR degradation pathway by binding to CAP1 (Jang, H.D. et al., *Eur Heart J*, 2020, 41, 239–252). We also identified that CAP1 is the receptor for Resistin leading to inflammation in human (Lee, S. et al., *Cell Metab*, 2014, 19, 484–497). Therefore, we speculated that PCSK9 might activate inflammatory signals through CAP1, directly or independently of LDLR. Then, we proved that PCSK9 triggered the activation of nuclear factor (NF)- κ B and secretion of pro-inflammatory cytokines (TNF- α , IL-1 β , and IL-6) from monocytes as well as the expression of adhesion molecules on monocytes and endothelial cells. In addition, we needed to identify the downstream molecules of the PCSK9-CAP1 inflammatory signaling mechanism, which is the purpose of conducting proteomics, based on PCSK9's direct activation of pro-inflammatory genes independently of LDLR in vitro and in vivo (Figure 1 and 2), and the requirement of CAP1 for PCSK9-mediated inflammation (Figure 3, 4, and 5).

While our screening was comprehensive, the selection of "immune system process" among the enriched terms was guided by our specific interest in understanding the inflammatory signaling mechanism of PCSK9-CAP1. Furthermore, we exclusively considered genes only with an AvgP (interaction score) exceeding 0.6 and an EASE score below 0.1 in analyzing GO

terms, which makes all terms to be significant and meaningful. Granted that choosing the lowest term is biased, however, we claim that even though "immune system process" ranked the lowest among the GO terms, it is not necessarily less important than those in higher ranks, as differences in p-values may be attributed to its various functions of CAP1.

Additionally, the higher-ranking terms, such as "metabolic process" and "cellular component organization or biogenesis," are currently under investigation in other projects within our group, indicating the diverse functions of CAP1. Such multifaceted biological functions suggest that CAP1 may play a role in various biological pathways, as reported in the literature (Rust, M.B. et al., *Front Cell Dev Biol*, 2020, 8, 586631). Lastly, I would like to briefly explain about domains of CAP1 protein, as it seems to be a reason why metabolic process was ranked the highest. CAP1 is composed of three main domains, which are adenylyl cyclase binding domain, SH3 binding domain and actin binding domain (Lee, S. et al., *Cell Metab*, 2014, 19, 484–497).

[FIGURE REDACTED]

Reviewer only Fig. 3-1. Schematic diagram of CAP1 domain. (Lee, S. et al., *Cell Metab*, 2014, 19, 484–497).

CAP1 is found in mammals, but it is believed that its cyclase regulatory component has been lost over the course of evolution. However, recent study revealed that CAP1 binds and activates mammalian cyclase, enabling to modulate cAMP synthesis (Zhang, X. et al., *Proc Natl Acad Sci U S A*, 2021, 118, e2024576118). cAMP mediates a wide range of intracellular processes and metabolic pathways such as regulation of hormone signaling, energy homeostasis, lipid metabolism, glucose homeostasis, regulation of enzyme activity by activating PKA, thermogenesis, cellular proliferation/growth and etc. Therefore, as CAP1 occupies a domain that activates cAMP, the highest rank 'metabolic process' may be an inevitable outcome.

Changes in the Manuscript Page 5, line 146

PCSK9, whose CRD domain is structurally similar to the resistin trimer, activates the LDLR degradation pathway by binding to CAPI⁷. We also established that CAPI serves as the receptor for resistin, leading to inflammation in humans⁸. Consequently, we focused on ~

[Comment #3-7] :Color legend in the revised Figure 6B is incorrect.

[Response #3-7]: Thank you for the helpful feedback. It seems that there was an error in the color legend during the copy-paste process. We have corrected the color legend in the revised version

Changes in the Manuscript Figure 6b

Fig. 6b 1st Revised data

2nd Revised data

[Comment #3-11]: In response to this comment authors provided a new set of co-localization data suggesting that PCSK9 and CAP1 are colocalization on macrophages (Fig. 9c), ECs (Fig. 9d), and SMCs (Fig. 9e). SMCs within the atherosclerotic milieu exhibited characteristics akin to immunocyte-like cells, consistent with previous research Ref16, 17. PCSK9 expression was identified in both SMA+F4/80+ and SMA+F4/80- cell populations, emphasizing its relevance across these cell types (Fig. 9f). But this dilutes the effect of CAP-1-PCSK9 interaction on monocytes leading to the activation of cytokines in monocytes and the progression of atherosclerosis.

[Response #3-11]: Regarding SMCs (vascular smooth muscle cells), they represent the most abundant cell type in vessels. In response to vascular injury, SMCs undergo a transition to a dedifferentiated type accompanied by morphological changes from a spindle shape to an irregular shape, referred to as synthetic SMCs. Certain synthetic SMCs exhibit macrophage-

like characteristics, as indicated by similar surface markers and function to macrophages. It has been estimated that these chimeric cells account for approximately $40\pm 6\%$ of all CD68-positive cells in human atherosclerotic lesions (Allahverdian, S. et al., *Circulation*, 2014, 129, 1551-1559). Although SMCs with macrophage-like characteristics contribute to the development of atherosclerosis, such contribution is lower than that of macrophages and the levels of inflammatory factors in these cells are also inferior to those in monocyte-derived macrophages. (Vengrenyuk, Y. et al., *Arterioscler Thromb Vasc Biol*, 2015, 35, 535-546). In summary, while certain SMCs can assume immune cell-like properties in atherosclerotic lesions, the overall contribution of SMCs to pro-inflammatory cytokine secretion is lower than that of monocytes. Therefore, while SMCs may contribute to the inflammatory process, it is undeniable that the monocyte-driven PCSK9 system remains the predominant factor in atherosclerosis.

Changes in the Manuscript Page 9, line 274

In addition, plaque macrophages express inflammatory cytokines that can enhance PCSK9 expression in ECs, VSMCs, and macrophages via SREBP-2, which enhances PCSK9 transcription²⁷. Although SMCs with macrophage-like characteristics contribute to the development of atherosclerosis, their contribution is lower than that of macrophages, and the levels of inflammatory factors in these cells are also inferior to those in monocyte-derived macrophages²⁸. Thus, based on the positive feedback loop between PCSK9 and cytokines involving CAP1 and SREBP-2, we expect PCSK9 to play a significant role in inflammatory vascular diseases.

[Comment #3-15]: To address this reviewer's concern regarding a very late time-point for phosphorylation data in Figure 2E: the authors have conducted a new set of experiments with a 40 min timepoint which makes more sense. The new data is still missing total p65 expression and the data quantification.

[Response #3-15]: We have adjusted the rhPCSK9 treatment time to 40 minutes, leading to more insightful and effective results. In the revised manuscript, we included quantitative images along with the immunoblot of p65, which seems to have been inadvertently omitted while emphasizing the 40-minute treatment time of rhPCSK9 in the comment. An independent experiment was conducted, and quantification was performed by normalizing to total p65. We confirmed a significant increase in p65 phosphorylation induced by PCSK9 in *Ldlr*^{-/-} and BL6 BMDM

Fig. 2e Original data

h Revised data